# Decoding molecular programs in melanoma brain metastases

Josefine Radke [1,2,3,4,18] ✉, Elisa Schumann[3,4,18], Julia Onken[3,5], Randi Koll[3,4], Güliz Acker[2,5], Bohdan Bodnar[5], Carolin Senger[6], Sascha Tierling[7], Markus Möbs [8], Peter Vajkoczy[5], Anna Vidal [9], Sandra Högler[10], Petra Kodajova[10], Dana Westphal[11,12,13,14], Friedegund Meier[11,12,13,14,15], Frank Heppner [2,3,4], Susanne Kreuzer-Redmer[16], Florian Grebien [9], Karsten Jürchott[17] & Torben Redmer [9,10] ✉

Melanoma brain metastases (MBM) variably respond to therapeutic interventions; thus determining patient's prognosis. However, the mechanisms that govern therapy response are poorly understood. Here, we use a multi-OMICS approach and targeted sequencing (TargetSeq) to unravel the programs that potentially control the development of progressive intracranial disease. Molecularly, the expression of E-cadherin (Ecad) or NGFR, the BRAF mutation state and level of immune cell infiltration subdivides tumors into proliferative/pigmented and invasive/stem-like/therapy-resistant irrespective of the intracranial location. The analysis of MAPK inhibitor-naive and refractory MBM reveals switching from Ecad-associated into NGFR-associated programs during progression. NGFR-associated programs control cell migration and proliferation via downstream transcription factors such as SOX4. Moreover, global methylome profiling uncovers 46 differentially methylated regions that discriminate BRAF$^{mut}$ and wildtype MBM. In summary, we propose that the expression of Ecad and NGFR sub- classifies MBM and suggest that the Ecad-to-NGFR phenotype switch is a rate-limiting process which potentially indicates drug-response and intracranial progression states in melanoma patients.

The development of brain metastases is frequent in melanoma, lung and breast cancer[1,2]. Despite much progress and remarkable response in a subset of patients[3], small molecule or immune checkpoint inhibitors (ICi) blocking oncogenic BRAF (BRAFi) or interfering with the PD-L1/PD1 axis to restore T cell activation are insufficient strategies to achieve a long-lasting prevention of intracranial relapse and progression[4,5]. The latter is determined by the emergence of multiple brain metastases and therefore associated with poor prognosis[3,6]. MBM develop in 20–40% of melanoma patients[3,7] during the course of disease, causing a median overall survival of 8.9 months[8] after the detection of MBM. The time from initial diagnosis of primary tumors to the detection of MBM ranges from 1–10 years, suggesting a slow evolutionary process of MBM from circulating tumor cells[9] which cross the blood brain barrier (BBB). Once tumor cells entered the brain, they initially remain in a dormant state and likely only a minority of micrometastases successfully develop symptomatic macrometastases following adaptation to the brain microenvironment[10–12]. Concordantly, micrometastases are observed in >90% of melanoma patients *post mortem*[13,14].

The loss of therapeutic control leads to intracranial progression, that in turn is the consequence of molecularly and genetically distinct subclones that variably respond to therapeutic interventions. Particularly, intrinsically resistant tumor cells harbor NRAS$^{Q61K/L}$, MEK1$^{P124}$ or RAC1$^{P29S}$ mutations or acquired secondary, resistance-conferring[15,16] mutations in BRAF (BRAF$^{L514K}$) as well as molecular circuits controlling minimal-residual disease (MRD)[3,17–24] which serve as important driving

forces of progression[25-28] that proceeds within ~6–11 months[29-31]. Recently, the presence of a neural crest stem cell (NCSC)-state of melanoma cells was associated with the maintenance of MRD. The NCSC-state in turn is controlled by an NGFR (nerve growth factor receptor)-associated network[23,24,32-35].

NGFR is crucial to maintain basic properties of melanoma cells such as survival, migration and stemness and is associated with drug resistance, metastasis and cellular plasticity[22,23,32,33,36-38]. Particularly, the latter non-genetic process, enabling the switching of melanoma cells within different phenotypical states, controls growth and invasiveness via modification of levels of NGFR expression[39]. Likely, phenotype switching is strongly affected by environmental cues such as inflammatory processes that foster dedifferentiation and enrichment of NGFR+ melanoma cells[23,40]. Intracranially, the enrichment of NGFR+ cells is potentially triggered by pro-inflammatory cytokines such as TGFβ that is provided by microenvironmental cells such as astrocytes or microglia[20,21,41,42]. Hence, inflammation-triggered mechanisms might drive the progression of MBM and may promote the emergence of migratory and drug-resistant cancer stem-like tumor cells (CSCs)[20,23,43-45].

On the other hand, NGFR controls migratory programs of melanocytes[46] that are connected with keratinocytes via E-cadherin (Ecad)-mediated adhesive junctions in the skin. The downregulation of Ecad is tightly controlled by a set of transcription factors mediating the transition of epithelial to mesenchymal (EMT) states and is a prerequisite for melanocyte migration and malignant transformation[47]. However, the expression of Ecad is essential to establish stemness and is restored in primary melanoma and organ-specific metastases[48-50].

The tracking of cellular subclones giving rise to MBM is rarely possible in human patients. Here we provide evidence that the expression of Ecad and NGFR defines at least two different molecular stages of MBM development and progression. Molecular subsets may differentially respond to therapy, thus determining the routes of intracranial disease.

## Results

### Therapeutic interventions promote the development NGFR+ MBM

Several lines of evidence suggest that therapeutic interventions such as BRAFi enhance the emergence of therapy-resistant cellular subclones, potentially driving relapse and progression at multiple extracranial and intracranial sites[51-55] (Fig. 1a). We investigated the levels of NGFR in MBM that developed and/or progressed in patients who received combinatorial therapies such as BRAFi/MEKi or ICi and XRT (SRS, WBRT). In line with our previous findings, we observed high expression of NGFR (70–100%) throughout the entire tumors in a subset of MBM (Fig. 1b, left panels and Supplementary Fig. 1a) irrespective of the intracranial location (Supplementary Data 1) and BRAF/NRAS mutation status as determined by TargetSeq during routine diagnostic work-up. Whole transcriptome data of MBM prior and after BRAFi/MEKi therapy are not available, therefore we investigated a set of patient-matched, drug-naïve (pre-relapse) and dabrafenib/trametinib (GSE77940[56]) treated and relapsed (post-relapse) tumors (n = 12). NGFR was significantly (2.5fold, p = 0.004) increased in five of six post-treatment patients, (Fig. 1b, right panel). Furthermore, we observed decreased levels in E-cadherin (Ecad, p = 0.013) and PLXNC1 (Fig. 1c, left panel) but gain in expression of invasion/migration-associated genes such as TSPAN13, TWIST1, LOXL2 and LOXL3 probably fostering relapse and progression[38,51,57]. GSEA (gene-set enrichment analysis) revealed a higher representation of signatures indicating an NGFR-driven or invasive (Hoek signature[58]) tumor cell state (Fig. 1c center and right panels, Supplementary Fig. 1b, left panel). Potentially, BRAFi promote phenotype switching through EMT (Supplementary Fig. 1b, center panel) or select for undifferentiated neural crest (NC)/NCSC-like cells (Supplementary Fig. 1b, right panel) accompanied by loss of

expression of DCT (dopachrome tautomerase; p = 0.040) and trending gain in AXL expression (Supplementary Fig. 1c). We hypothesized that intracranial progression is controlled by NGFR-driven programs fostering the emergence of NGFR+ micrometastases and investigated matched pre- and post-relapse MBM. NGFR+ cells infiltrated the brain tumor environment (BTE) (Fig. 1d, left panels) and formed micrometastases in close proximity to MBM/BTE transition sites (Fig. 1d, right panels). This phenotype was even more prominent in a MBM of a patient who was completely refractory to BRAFi/MEKi or ICi-based therapies (Supplementary Fig 1d). The developing human brain exhibits spatial differences of the cellular composition[59]. Hence, the response of tumors to environmental cues is likely governed by the cellular composition of the BTE and secreted soluble factors, that might control progression stages of MBM and primary brain tumors[60]. To gain insight into programs that define molecular subsets potentially determining the progressive state of MBM, we collected cryopreserved MBM (n = 16; Supplementary Data 1) from different intracranial sites (Fig. 2a) including longitudinal metastases (Pat8) and patient-matched tumors (patients 23, 24). The initial TargetSeq provided information on the mutation status of hot spot regions of 50 cancer-associated genes and revealed that MBM (n = 29) either harbored mutations in BRAF or NRAS with variant allele frequencies (VAF) of 0.91–0.26 (BRAF^V600) or 0.87–0.40 (NRAS^Q61/G13), in line with previous observations[3,61]. TargetSeq identified genetic aberrations in 11 genes among them expected drivers of melanoma progression such as CDKN2A (Supplementary Fig. 2a, b). Transcriptome profiling of MBM and normal brain controls (Cortex, Pons, Cerebellum/Cereb; BC; Supplementary Data 2) revealed a separation regarding molecular features and the content of admixed brain parenchyma, irrespective of the intracranial region or genetic state (presence of BRAF or NRAS mutations) of tumors (Fig. 2b). In a second step, we determined the DEGs (differentially regulated genes) among BC and MBM and identified a pan-gene signature including CDH1 (Ecad), PMEL and SOX4 and FOXD3 potentially controlling MBM-specific features (Fig. 2c and Supplementary Data 2). As neuronal cells express NGFR[62] our survey failed to identify the subset of NGFR+ tumors but observed a strong homogeneity of matched, synchronously resected metastases of patient 23 and a clear separation of metachronous, BRAFi/MEKi-naïve (M1, 2018) and drug-resistant (M4, 2020) subclones of Pat8.

### E-cadherin and NGFR expression define molecular subgroups of MBM

The malignant transformation of melanocytes to melanoma accompanies the downregulation of Ecad implying a low level of Ecad expression in metastases; however our previous exploration suggests that Ecad expression is maintained by certain circumstances even in distant metastases. We observed that 56.3% (9/16) of MBM featured Ecad expression among them tumors of patients who exhibited meningeosis melanomatosa (Pt.8,11) at late-disease stages and concordant pairs of pre- vs. post-relapse (Pat19) and extracranial (spinal) vs. intracranial (Pat6) MBM (Ecad^high, Supplementary Fig. 3a). In contrast, we observed NGFR expression in Ecad^low tumors (Supplementary Fig. 3b). Levels of Ecad were comparable among extracranial and intracranial metastases as confirmed by comparison of Ecad levels of intracranial (n = 79, MBM) and extracranial metastases (n = 59, EM; p = 0.6144) (Supplementary Fig. 3c, left panel). Likely, brain metastatic tumor cells exhibit a rapid EMT-MET capacity and metastases reacquire Ecad expression soon after colonization of distant organs. We surveyed the TCGA-SKCM data set comprising primary melanoma (PT) and EM and observed a significantly lower but not strongly decreased level of Ecad in EM than PT (Supplementary Fig. 3c, right panel). Ecad-mediated cell adhesion probably promotes different molecular traits in tumor cells as NGFR+ MBM that may exhibit Ncad (N-cadherin, CDH2) mediated cell adhesion. To ascertain the molecular features of Ecad+ and NGFR+ MBM, we ranked tumors regarding the expression

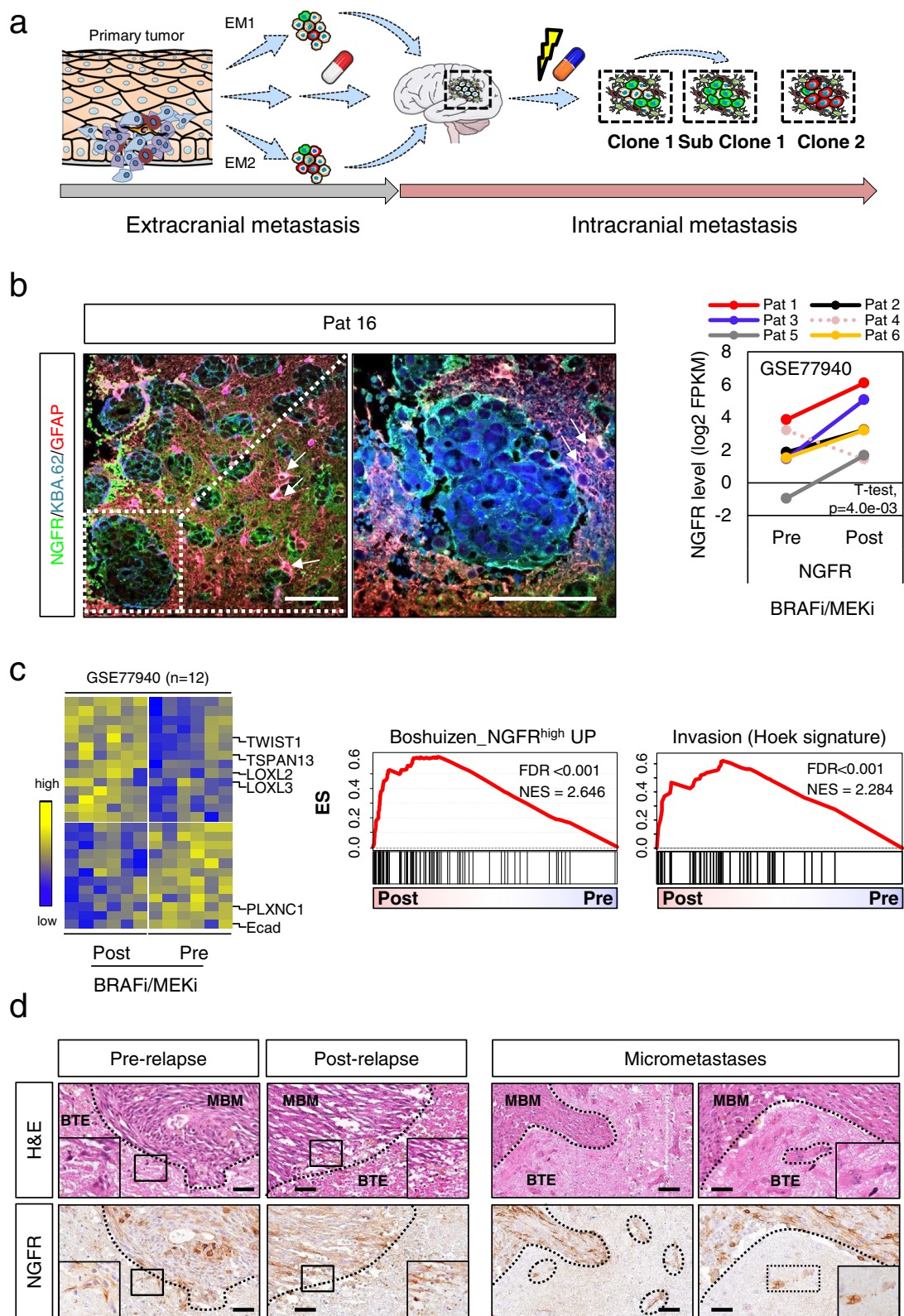

levels of both (Supplementary Fig. 3d) and determined genes that correlated with Ecad or *NGFR* expression. The comparison of Ecad$^{high}$ vs Ecad$^{low}$ and NGFR$^{high}$ vs NGFR$^{low}$ MBM identified ~2400 DEGs ($p \leq 0.05$) (Supplementary Data 3) and revealed an inverse association of Ecad and NGFR correlating genes (Supplementary Fig. 3e, left panel). Moreover, we found a correlation of the BRAF/NRAS mutation status of MBM, suggesting increased expression of *NGFR* ($p = 0.014$) in

*BRAF*$^{wt}$/*NRAS*$^{mut}$ and higher levels of Ecad in *BRAF*$^{mut}$ ($p = 0.013$) MBM (Supplementary Fig. 3e, center and right panels). Gene signatures clearly separated the fast majority of MBM with IHC-proven Ecad$^{+}$ or NGFR$^{+}$ phenotypes into distinct molecular subsets (Fig. 3a, b) and revealed no clear correlation with the degree of immune cell infiltration as judged by the presence of CD3$^{+}$ T cells (Fig. 3b, lower panels). Moreover, gene signatures uncovered intermediate state tumors (Pt.1,

**Fig. 1 | MAPKi treatment modulates the phenotype of melanoma. a** Schematic representation of routes of the metastatic cascade establishing extracranial (EM1, EM2, EMn, etc.) and intracranial metastases. Therapeutic interventions such as radiation (flash) and BRAFi/MEKi/ICi (pill) likely foster extracranial and intracranial metastasis and subclonal evolution. **b** Left panels: Immunofluorescence (IF) for NGFR (green), GFAP (labeling reactive and normal astrocytes) and KBA.62 (general marker for melanoma cells) of a MBM (Pat16) that progressed under combinatorial treatment and WBRT demonstrating the high infiltration of the brain tumor environment (BTE) and presence of reactive astrocytes (white arrows). DAPI served as nuclear dye. Right panel: NGFR expression levels of patient-matched melanoma ($n = 12$), pre- ($n = 6$) and post-BRAFi/MEKi ($n = 6$) therapy (study GSE77940), one tumor has been excluded (dotted line) in each group. Statistically significant differences were tested by a paired two-tailed t-test. **c** Left panel: depiction of top differentially regulated genes (DEGs) such as mediators of invasion/migration, *TSPAN13* (Transmembrane 4 Superfamily Member 13), *LOXL2* and *LOXL3* (Lysyl Oxidase Like 2/3) among pre- and post-BRAFi/MEKi melanoma. Center and right panels: GSEA of samples analyzed in **b** and **c** revealed enrichment of anti-PD-1 resistant/ NGFR^high and invasiveness-related phenotypes. FDR indicates the significance of enrichment, ES enrichment score, NES normalized enrichment score. 10,000 permutations were performed. **d** IHC of matched pre- and post-relapse MBM revealed NGFR$^+$ tumor cells showing infiltration of the BTE and formation and micrometastases. Hematoxylin and eosin (H&E) staining show discrimination of tumor (melanoma brain metastases, MBM) and BTE. Scale bars indicate 50 μm. Source data are provided as a Source Data file.

8/M4 and 22) that comprised co-existing Ecad$^+$ and NGFR$^+$ cells. Our survey identified Ecad, *GJB1, PMEL, TSPAN10* and *CDK2* or *NEGR1, SOX11, NGFR, EHD3* and *EDN3* among the top DEGs in Ecad^high or Ecad^low tumors (Supplementary Data 3). Hence Ecad$^+$ and NGFR$^+$ tumors likely present molecularly distinct subsets that might differentially respond to therapeutic interventions. GSEA unraveled the molecular features of Ecad$^+$ and NGFR$^+$ tumors (Supplementary Data 4) and revealed a proliferative phenotype of Ecad^pos tumor cells (Fig. 3c, left panel) and enrichment of a tumor-intrinsic NGFR-signature that was derived from a set of melanoma cell lines which had spontaneously acquired resistance to T cells in Ecad^low/NGFR^high tumors (Supplementary Fig. 4a). The signature potentially predicts anti-PD-1 therapy resistance, and increased immune exclusion. Moreover, Ecad^low/NGFR^high MBM exhibited an invasive phenotype (Hoek_Invasive, NES = 2.318, FDR < 0.001) among other core enrichments (Fig. 3c, right panel and Supplementary Data 4).

Generally, primary and secondary brain tumors are immunologically cold (non-inflamed) tumors[63] and efficiently evade immune surveillance. We observed that the presence of TILs (tumor infiltrating lymphocytes) (CD3D$^+$, CD8A$^+$ T cells) in a subset of our cohort ($n = 8$; 50%) served as a classifier that clearly distinguished favorable[64] TIL^high and TIL^low subgroups (CD3D, $p = 2.8e{-}07$; CD8A, $p = 2.6e{-}04$) of MBM (Fig. 3d). The latter association of survival and presence of CD3$^+$ and CD8A$^+$ T cells was validated in a cohort of MBM ($n = 80$; study EGAS00001003672, Supplementary Fig. 4b). Moreover, TIL^high tumors featured increased inflammatory responses (Fig. 3e, left panel). In line with this, a deconvolution using ESTIMATE and CIBERSORT tools revealed a clear separation of tumors showing absence of $M_0$ but presence of pro-inflammatory $M_1$ and anti-inflammatory $M_2$ macrophages, potentially validating the presence of a supervised inflammatory phenotype (Fig. 3e, right panel) in TIL^high tumors.

Hence, although the emergence of MBM is generally associated with poor prognosis, the different phenotypes might determine the degree of aggressiveness of intracranial tumors and their capability in the formation of multiple brain metastases and response to therapeutic interventions.

## The progression of MBM accompanies an E-cadherin-to-NGFR phenotype switch

Non-genetic processes such as cellular plasticity and phenotype switching are driving forces of tumor heterogeneity and likely determine drug response and tumor relapse[65]. To gain insights into phenotypical and molecular changes that occurred alongside progression, we investigated spatially separated, longitudinal metastases of Pat8 that were collected before BRAFi/MEKi therapy (M1) or which have developed and progressed under therapy (M2, M3, M4) (Fig. 4a). We observed a high level of NGFR expression in M3 and M4 but a low level in pigmented subclones M1 and M2 (Supplementary Fig. 5a). At the time of M4 resection, the patient exhibited a very aggressive disease stage that was accompanied by *meningeosis melanomatosa*, the penetration of (HMB45 positive) melanoma cells into the CSF[66] (Supplementary Fig. 5b). Presumably, acquisition of a NGFR$^+$ phenotype

marks a final step of MBM progression, likely presenting a stepwise and slowly proceeding than a rapid process. We examined the levels of Ecad and NGFR in a lymph node metastasis (LN-MET) and concordant MBM (M1, M4) and validated a co-occurrence of melanoma cells that featured distinct (~35% NGFR$^+$; ~50% Ecad$^+$) and overlapping (~10% NGFR$^+$/Ecad$^+$) phenotypes (Fig. 4b, left panels and Supplementary Fig. 5c). In line with previous observations we found a distinct expression of both markers in M1 and M4 (Fig. 4b, right panels). NGFR$^+$/Ecad$^+$ cells that potentially reflected the plasticity-driven Ecad-to-NGFR transition were not evident in M4. KBA.62 was used as a general marker of melanoma cells that enabled detection of stem-like NGFR$^+$ and non-stem-like melanoma cells of primary and metastatic melanoma[67].

Next, we investigated the representation of Ecad and *NGFR* gene signatures in pre- and post-BRAFi/MEKi treated melanoma and observed a significant upregulation of NGFR-core genes such as *EHD3* ($p = 0.042$). Mean expression values of NGFR- but not Ecad-signature genes significantly ($p = 6.2e{-}03$) separated pre- and post-treatment melanoma (Fig. 4c, left panel). Moreover, the signature-based deconvolution revealed a classification of pigmented/Ecad-core$^+$ and non-pigmented (amelanotic) but invasive/NCSC-like/NGFR-core$^+$ MBM. The latter subset exhibited a low level of *MITF*-target genes (Fig. 4c, center panel). Our previous studies suggest that the expression of *NGFR* is crucial for the maintenance of an NCSC-like phenotype, potentially fostering MRD in NGFR$^+$ MBM. GSEA and expression analysis revealed that NGFR$^+$ tumors indeed expressed high levels of MRD-associated genes (Fig. 4c, right panel and Supplementary Fig. 5d, left panel), particularly the MRD-associated gene *EHD3* was upregulated in post-treatment melanoma and was significantly more highly expressed in NGFR^high MBM (Fig. 4d, left panel). Moreover, expression of *EHD3* and *NGFR* were significantly correlated (Supplementary Fig. 5d, center and right panels) and might work in concert. In addition, we observed a significant correlation between the promoter of invasiveness *LOXL3* and expression of *NGFR* in primary (PT) and metastatic (EM) melanoma (Supplementary Fig. 5e). However, *LOXL3* expression did not significantly discriminate our MBM cohort. In summary, NGFR defines molecular features such as NCSC-like stemness and a MRD/resistant phenotype. Single sample GSEA (ssGSEA) indeed revealed a dependency of tumors from NGFR (p75^NTR)-mediated signaling and demonstrated a correlation of *NGFR* expression with brain metastasis in breast cancer[68] (Fig. 4d, right panel, $p < 0.05$), suggesting that NGFR^high melanoma are probably prone to brain metastasis.

## Ecad expression sensitized melanoma cells to dabrafenib

The complex interplay of cells that make up the microenvironment alongside tumor cells strongly determines tumor progression. We established MBM-derived in vitro models (BMCs) that exhibited different genetic backgrounds to gain insights into the environmental dependencies of tumor cell properties. The derivation of tumor cells and adaptation to the in vitro cell culture conditions was accompanied by the loss of environmental cells such as astrocytes and microglia (Fig. 5a). Expression profiling of the initial tumor (Pat8/M1) and the

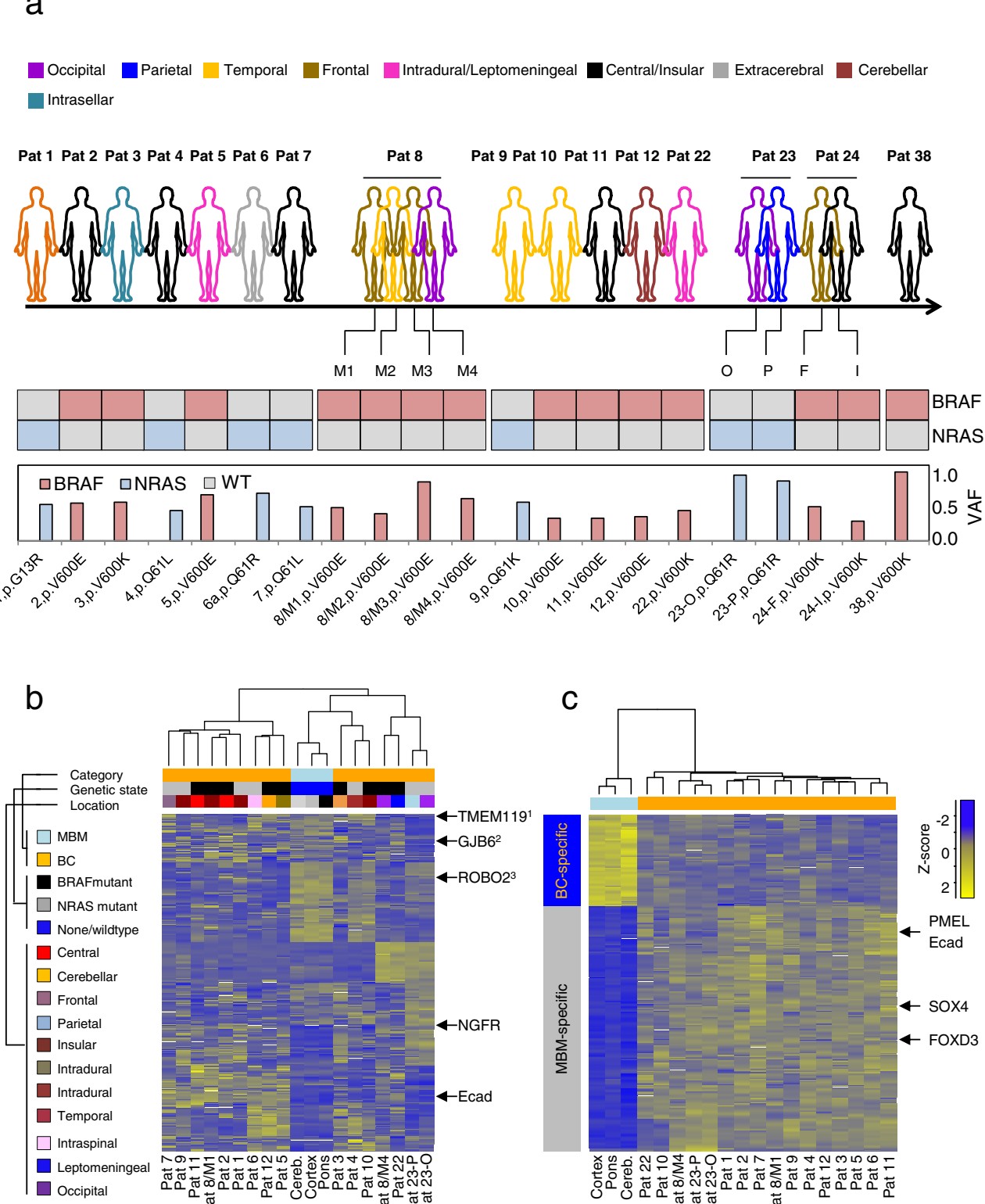

**Fig. 2 | Characterization of melanoma brain metastases (MBM) revealed different molecular subtypes. a** Representation of the core study cohort, including MBM that have been profiled by RNAseq, 805k global methylome analysis and/or TargetSeq. Concordance (Pt.8, 23, 24), intracranial site of tumors and the type and variant frequency (VAF) of BRAF/NRAS mutations is shown. **b** Unsupervised analysis of the top1000 variably expressed genes among all MBM (*n* = 16), compared with brain controls (BC, *n* = 3) revealed clustering independent from intracranial sites of discordant MBM or genetic state. Concordant MBM (Pat 23) show a convergent clustering. Legends provide information about the category (MBM, BC) or genetic state (BRAF/NRAS mutant) or intracranial site. Expression of Ecad or *NGFR* or markers of BTE cells such as microglia[1] (*TMEM119*), astrocytes[2] (*GJB2*), and neurons[3] (*ROBO2*) among samples is indicated. **c** Heat map illustration of DEGs among BC and MBM, top-expressed, MBM-related genes are indicated. In **b** and **c** only significantly regulated genes (Bonferroni corrected, *p* < 0.05) are shown.

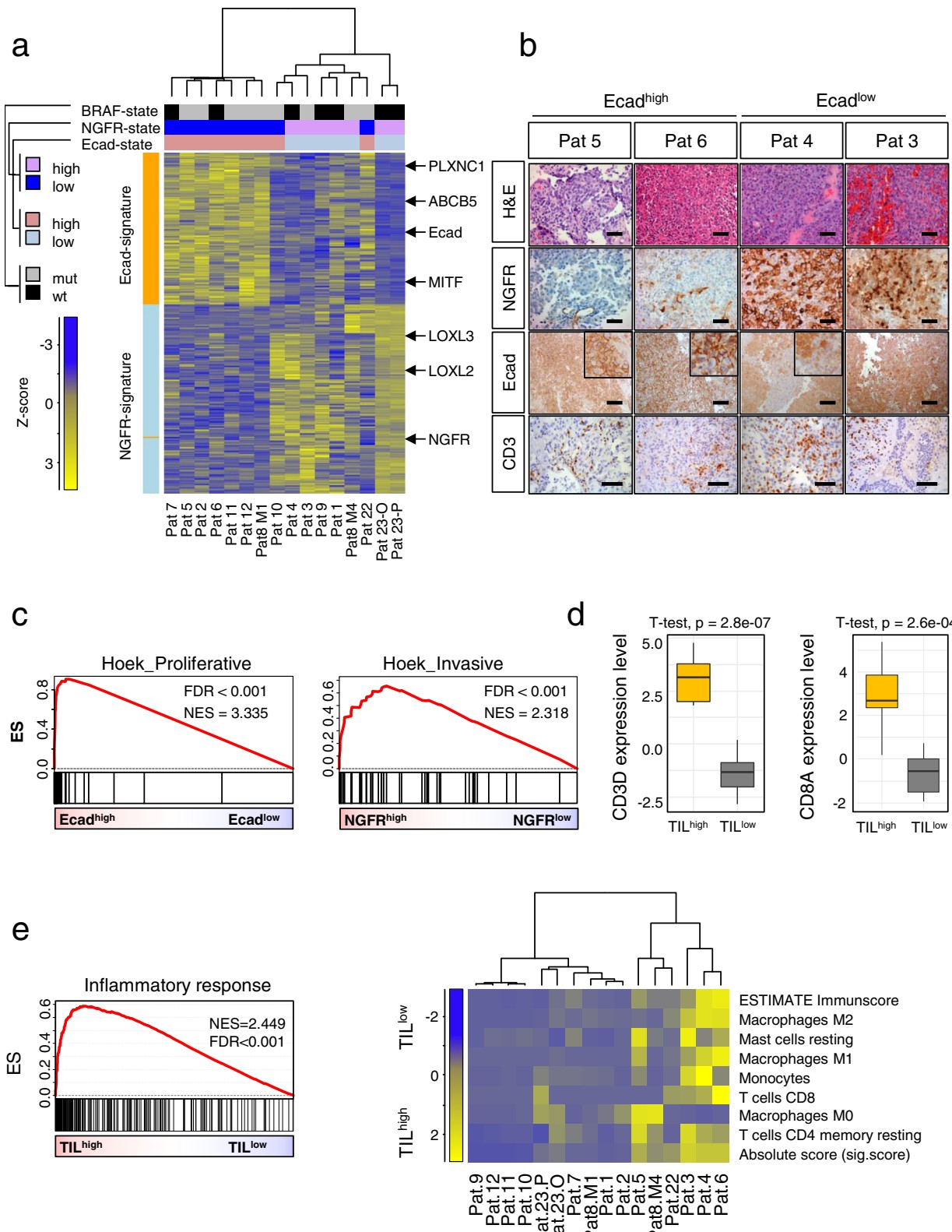

stably established, early-passage (p1, 1°) cell line (BMC1-M1) revealed a high concordance of M1 and BMC1-M1 ($R = 0.89$, $p < 2.2e-16$) and identified 992 DEGs that were likely controlled in a BTE-dependent manner (Supplementary Data 5). Particularly, we observed a loss of Ecad expression and gain in expression of *NGFR*, *FOSL1*, *EHD3* and *LOXL2* during prolonged culturing ($p \geq 3$, 2°) (Fig. 5b, left panel). Moreover, qPCR revealed that all except BMC1-M4 cells featured a low

mRNA expression of Ecad whereas levels of *NGFR* of most BMCs were comparable, with the exception of BMC2 cells (Fig. 5b, center and right panels). Concordantly, we found that all MBM-derived cell lines were enriched in NGFR[+] cells and comprised only a minor Ecad[+] subset (Fig. 5c, d and Supplementary Fig. 6a). AXL or MET, additional potential drivers of metastasis[69] showed low expression (Supplementary Fig. 6b, left panels). BMCs comprised NGFR[+] adherently growing

**Fig. 3 | Expression of Ecad and NGFR defines molecular subgroups of MBM.**
**a** Separation of MBM regarding the presence of Ecad- or NGFR-associated genes as identified in this study. Intermediate-state tumors (Pt.1, Pat8/M4, 22) feature expression of both signatures, at least partly. Molecular subgroups (BRAF^mut vs. wt) and Ecad/NGFR states are color coded. Heat map presents a supervised, euclidean, ward.D clustering. *PLXNC1* (Plexin C1), *ABCB5* (ATP Binding Cassette Subfamily B Member 5) and *MITF* (Melanocyte Inducing Transcription Factor) or *LOXL2*/*LOXL3* (Lysyl Oxidase Like 2/3) among others served as markers of Ecad^high or NGFR^high states, respectively. **b** IHC of selected MBM for Ecad, NGFR and CD3 validated proper expression of investigated markers and assigned molecular subtypes. Hematoxylin&Eosin (H&E) staining depicts morphological differences of tumors cells. **c** GSEA of Ecad^high and NGFR^high subsets showing enrichment of proliferative and invasive phenotypes. FDR indicates the significance of enrichment, ES enrichment score, NES normalized enrichment score. 10,000 permutations were performed. **d** Box plots depicting the levels of *CD3D* and *CD8A* of MBM defined as TIL^high and TIL^low, ranked by expression levels of *CD3D*, $n = 16$ biologically independent tumors were investigated. **e** Left panel: GSEA of TIL^high and TIL^low MBM demonstrating a high inflammatory response in TIL^high MBM. Right panel: Characterization of TIL^high and TIL^low MBM by CIBERSORT and ESTIMATE clearly discriminate tumors regarding levels of T cells (*CD4*, *CD8*), macrophages, monocytes and mast cells. Significance was determined by a unpaired two-tailed, *t*-test. Box and whisker plots show median (center line), the upper and lower quartiles (the box), and the range of the data (the whiskers), including outliers (**d**). Source data are provided as a Source Data file.

and suspension cells (observed in 3/5 BMCs; 60%) (Supplementary Fig. 6b, right panels) a feature that was probably associated with levels of NGFR expression (Supplementary Fig. 6c) as previously shown[39].

Minimal-residual disease (MRD) is established by a rare subset of drug-resistant tumor cells, consequentially leading to tumor relapse. BMC1-M4 cells were derived from a tumor that developed under BRAFi/MEKi therapy and was resected shortly after nivolumab/ipilimumab treatment; hence expected to feature a resistant phenotype. However, BMC1-M4 but not BMC1-M1 cells were sensitive to dabrafenib as reflected by distinct IC$_{50}$ values (Fig. 5e). Clonal and subconal evolution of M1 and M4 tumors might be responsible for a unique reaction to therapeutic drugs and in vitro cell culture processes might have selected for certain genetic subclones. We performed TargetSeq of longitudinal tumors, and associated BMCs and CSF of Pat8 as tumors and tumor-derived BMCs of Pt.35 and 27 for investigation of hotspot regions of 560 cancer-related genes with a mean coverage of 760x (range 290x-1,505x), (Supplementary Data 6). We identified 18 ground-state mutations that were commonly found in all specimens of Pat8, particularly BRAF^V600E and RAC1^P29S, well known genetic drivers of cancer progression (Supplementary Fig. 7a, b, Supplementary Data 6). Mutations in *RAC1* present early UV-radiation-induced aberrations, potentially driving BRAFi-resistance and cell migration[27,70]. In addition, we identified likely deleterious but functionally uncharacterized mutations in *CARD11*, *MYC* (CARD11^D56N; COSV62717671 and MYC^N26S; COSV52371145), and *NOTCH3* that have been associated with cancer[71] and/or were predicted as probably damaging (Supplementary Fig. 7c). Apart from shared mutations, we also detected mutations which were exclusively found in either of the longitudinal tumors. Likely, individual mutations such as in the MAPK-pathway (MAP3K1^E224X) hallmark late progression states and potentially fostered the emergence of highly aggressive subclones. The minor NOTCH3^S1128P subclone in Pat8/M4 (AF = 0.03) was maintained by in vitro cell culture conditions (BMC1-M4, AF = 0.46; Supplementary Fig. 7d), suggesting a potential role for cell survival.

Overall, these results suggest that the general composition of genetic subclones of longitudinal tumors was comparable and well represented by in vitro models. As subclonal evolution was likely not accountable for the different response of BMC1-M1 and BMC1-M4 cells to dabrafenib, we asked next, whether the overexpression of Ecad might be sufficient to modulate dabrafenib-sensitivity. BMC1-M4 and A375 cells were lentivirally transduced with plasmids for a constitutive expression of GFP or Ecad-GFP. Immunofluorescence imaging validated the absence of Ecad in GFP transduced cells and demonstrated the expression and proper membrane localization of Ecad (Fig. 5f, upper panels) in Ecad-GFP cells. Moreover, qPCR validated the overexpression of Ecad and modulation of expression of *NGFR* and of *TSPAN10*, in BMC1-M1, BMC1-M4 and A375 cells, at least partly (Fig. 5f, lower panels). However, a significant effect such as decrease of NGFR ($p = 1.8e−02$) and increase of *TSPAN10* (3.1e−04) was only observed in A375 cells, and was may be due to the duration of high expression of Ecad in the different cell types. We performed dabrafenib titration and tracked only GFP+ cells, hence excluded Ecad negative cells from

analyses. Indeed, Ecad-OE cells exhibited higher sensitivity to dabrafenib in a range of 1–30 nM, as reflected by a clear reduction of the IC$_{50}$ as observed in BMC1-M4 and A375 cells. However, the proliferation of Ecad overexpressing and control cells was comparable (Fig. 5g, Supplementary Fig. 8a, b).

Considering the high plasticity of melanoma cells and our previous data, Ecad and *NGFR* are likely interconnected. To address whether Ecad+ evolved from NGFR+ cells or vice versa (Supplementary Fig. 8c, upper scheme), we established a transcriptional dual-reporter system facilitating the tracing of the Ecad+ subset via the Ecad promoter-controlled expression of RFP. Furthermore, we monitored the NGFR subset via a 3´-UTR-GFP miRNA-reporter that indirectly enabled the detection of *NGFR*-mRNA stability (Supplementary Fig. 8c, lower scheme)[34,38,72]. The additional constitutive expression of iRFP enabled the general labeling of reporter cells independent from phenotype switching processes (Supplementary Fig. 8d, upper panels). As NGFR^GFP+ cells are generally sustained in vitro[72], we traced FACS-enriched RFP/iRFP cells (Supplementary Fig. 8d, lower panels). The initial (100%) Ecad+ fraction was decreased by $38.9 \pm 13.6\%$ ($p \leq 0.001$) 2 days after the FACS-based isolation (Supplementary Fig. 8e), suggesting that Ecad+ subsets are unstable and not sustained by standard 2D in vitro conditions. Nevertheless, live cell-imaging revealed rare (<0.1%) derivation of Ecad^RFP+ cells from NGFR^GFP+ (Supplementary Fig. 8f and Supplementary Movies 1, 2), or double negative cells.

In summary, our data suggest that NGFR and Ecad control molecular programs that determine cellular properties. However, cellular plasticity and environmental cues likely govern the spatiotemporal evolution and maintenance of these cellular subsets.

## A gene signature discriminates progressive and non-progressive MBM

It is likely that multiple MBM consequentially emerge from the seeding of a common founder clone and hence present temporally, but not necessarily genetically, distinct subclones. Whether the subclone M4 directly emerged from M1 or a related clone or subclone is unknown (Fig. 6a). Certainly, M4 represents a tumor that developed from a therapy resistant subclone. The comparative analysis of transcriptomes of M1 and M4 revealed 1063 differentially expressed genes that should represent the molecular features of intracranial progression. We defined a core signature of 389 genes (Supplementary Data 7) and investigated their abundance in all MBM. We observed subclustering and indeed identified additional tumors (Pat23-O, P; Pat22) that likely featured a progressive phenotype, among them NGFR+ tumors that exhibited a low level of Ecad expression (Fig. 6b). However, Pat22 (NGFR^low, leptomeningeal metastasis) and Pat8/M4 probably shared common features of progression, independent of NGFR expression. The progressive gene signature significantly ($p = 1.4e−04$) discriminated non-progressive and progressive MBM (Fig. 6c, upper panel). Moreover, *EHD3* and *LOXL2* were among the signature genes, showing a significantly increased level in Ecad^low MBM ($p = 0.013$ and $p = 0.026$) of an independent data set (Fig. 6c, center and bottom panels).

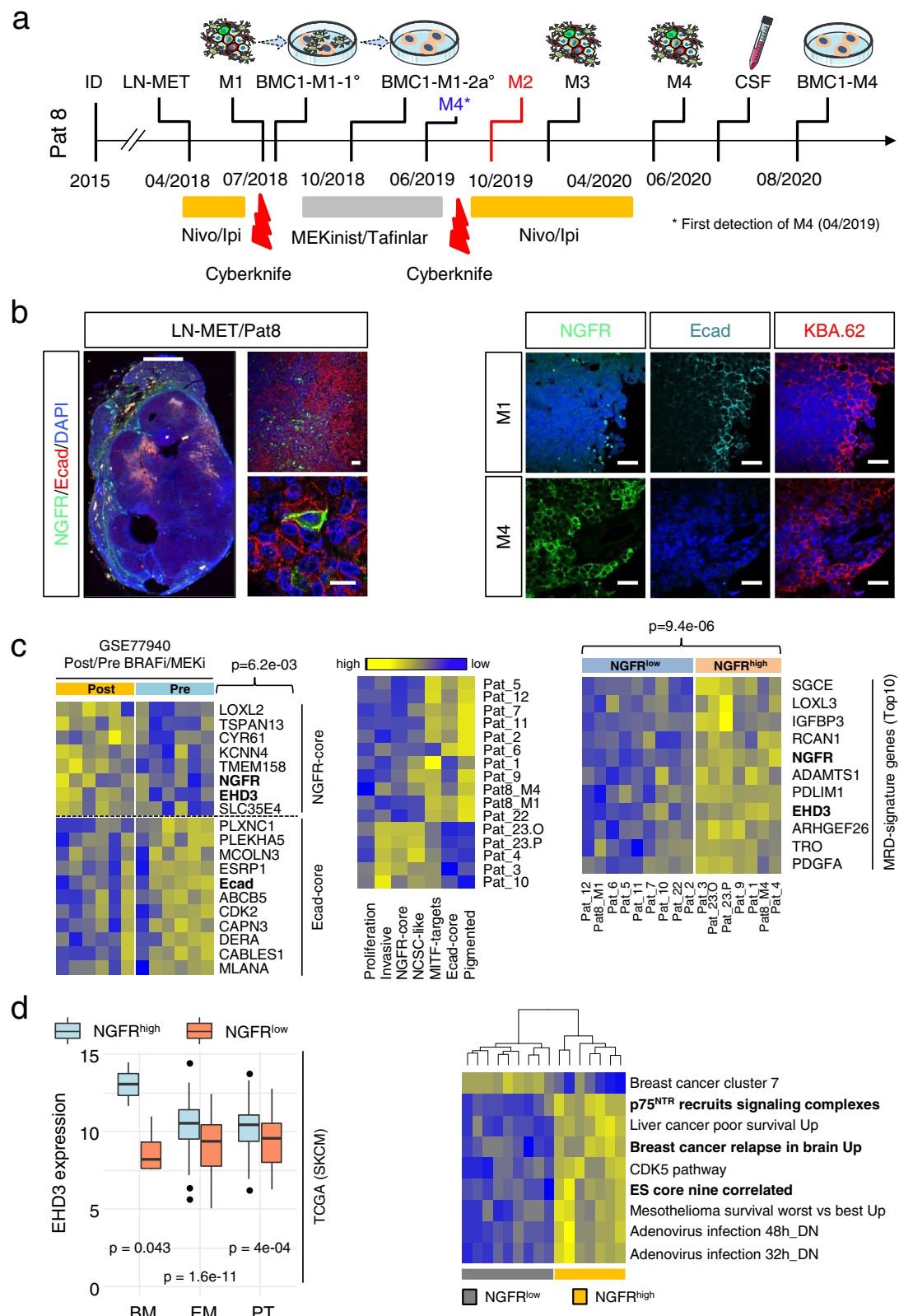

Intracranial progression and formation of multiple tumors likely includes the three-dimensional invasion and migration of single cells into the surrounding microenvironment that is build up by stromal cells and stromal cell-secreted extracellular matrix (ECM) proteins, forming the metastatic niche[73]. The capability of in vitro spheroid formation of melanoma cells is associated with stemness and a progressive phenotype[74,75]. To ascertain the spheroid-forming capacity of

BMCs, we seeded $2.5 \times 10^3$ BMC1-M1 and BMC1-M4 cells onto a layer of matrigel, which enabled spheroid formation following invasion of the collagen and laminin-rich layer. BMC1-M4 cells formed loosely connected three-dimensional (3D) spheres and satellite colonies (Fig. 6d, upper panel) whereas BMC1-M1 established colonies featured a non-scattered phenotype (Fig. 6d, lower panel). This assay enabled an interaction of tumor cells with ECM, however, it lacked in secreted

**Fig. 4 | Intracranial progression accompanied an Ecad⁺ to NGFR⁺ phenotype switch. a** Timeline indicating the therapeutic history and time points of surgical removal of tumors, cerebrospinal fluid (CSF) and establishment of tumor-derived cell lines of Pat8, initially diagnosed (ID) with primary melanoma in 2015. **b** Left panels: IF of a M1 concordant lymph node metastasis (LN-MET) for Ecad and NGFR, indicating unique and co-expression of both in a whole-tissue scan and by confocal microscopy. Right panels: Comparative imaging of M1 and M4 for levels of cell surface expression of Ecad and total levels of NGFR. KBA.62 served as marker of stem-like and non-stem like human melanoma cells. Scales indicate 50 μm. **c** Left panel: Enrichment of NGFR-core genes and simultaneous downregulation of Ecad-core genes in pre- and post-BRAFi/MEKi melanoma. Only NGFR-signature genes *TSPAN13* ($p = 4.19e−02$); *TMEM158* ($p = 8.06e−03$); *EHD3* ($p = 4.18e−02$) and *NGFR* ($p = 4.10e−03$) and Ecad-signature gene *PLXNC1* ($p = 3.98e−03$) and combined NGFR-signature genes ($p = 6.2e−03$) significantly discriminate pre/post melanoma. Significance was determined by a paired two-tailed, *t*-test. Center panel:

Deconvolution of MBM based on signature genes indicating displayed subgroups. Right panel: Investigation of MBM for presence of MRD-related genes demonstrates a correlation of MRD-status and expression of *NGFR*; top10 NGFR[high]-enriched genes *SGCE* ($p = 2.88e−03$); *RCAN1* ($p = 3.95e−03$); *NGFR* ($p = 4.25e−06$); *ADAMTS1* ($p = 2.66e−03$); *PDLIM1* ($p = 1.91e−03$); *EHD3* ($p = 6.73e−04$) and *ARHGEF26* ($p = 4.34e−04$) are shown. Significance was determined by a unpaired two-tailed, *t*-test. **d** Left panel: Box plots depicting expression levels of *EHD3* (EH Domain Containing 3) in NGFR[high] and NGFR[low] subsets of primary tumors (PT), extracranial (EM) and brain metastases (BM) of the TCGA-SKCM data set. Right panel: Single-sample GSEA of NGFR[high] and NGFR[low] MBM ($n = 16$) depicting nine discriminating gene signatures, among them NGFR/p75[NTR] signaling, embryonic stemness and breast cancer-related brain metastasis. In **d** box and whisker plots show median (center line), the upper and lower quartiles (the box), and the range of the data (the whiskers), including outliers. Significance was determined by a two-way anova. Source data are provided as a Source Data file.

factors that are potentially supplied by BTE cells. The 3D satellite-growth pattern of BMC1-M4 cells suggested a higher progressive phenotype as compared to BMC1-M1 cells. We discovered a proliferative phenotype of the progressive tumor M4 ($61.0 ± 11.0\%$ Ki67⁺ cells) and a lower level of proliferative cells in M1 ($17.1 ± 12.3\%$ Ki67⁺ cells, Supplementary Fig. 9a, left panel). This finding was also reflected by Ki67 levels (BMC1-M1, $48.1 ± 7.0\%$; BMC1-M4, $52.4 ± 19.4\%$), and BrdU incorporation (BMC1-M1, $35.0 ± 9.9\%$; BMC1-M4, $45.5 ± 8.5\%$; BMC4, $25.5 ± 6.2\%$) of the corresponding BMCs (Supplementary Fig. 9a, center panel). In addition, we validated the proliferative capacity of BMCs (Supplementary Fig. 9a, right panel) that was well reflected by doubling times (BMC1-M1: $80.0 ± 25.2$ h; BMC1-M4: $34.6 ± 2.2$ h; BMC2: $33.0 ± 3.0$ h; BMC3: >96 h; BMC4: $40.2 ± 15.1$ h).

Cellular properties of BMCs might have changed in response to in vitro conditions and growth patterns of in situ MBM are only accessible via MRI. To explore the in vivo growth properties of BMCs we established orthotopic models. We assumed that the transfer of BMCs back into a versant, non-inflamed environment might equalize the properties of BMCs. Accordingly, we injected $2.5 × 10^4$ cells of three BMCs; BMC1-M1, BMC1-M4 and BMC2 into the right hemispheres of CD-1 nude mice ($n = 3$ per group) and weekly tracked tumor formation via MRI for a period of 49 days (Fig. 6e, upper left scheme). BMC1-M4 cells established detectable tumors (median volume of $10.64 ± 7.04$ mm³ ($n = 3$; range: 3–17 mm³) 14 days post injection (range 7–14d) that reached a median volume of $25.59 ± 4.37$ mm³ (range 22–30 mm³) after 21d. BMC2 cells harbored mutations in BRAF (BRAF[N581Y]) and NRAS (NRAS[G12C]) and reached a median volume of $14.00 ± 0.27$ mm³ (range 13–14 mm³) after 28d. In contrast, BMC1-M1, like BMC1-M4 cells harbored mutations in BRAF (BRAF[V600E]) and *RAC1* (RAC1[P29S]) but indeed featured a less progressive phenotype and established small size tumors (maximum volume: 2.705 mm³; $n = 2$) 35d after injection (Supplementary Fig. 9b).

We observed that the growth properties were not affected by the BTE and proliferative properties of BMC1-M1/M4 cells were maintained in vivo. BMC2 cells that harbored *BRAF*[N581Y] and *NRAS*[G12C] mutations, equally responded, suggesting a superior role of yet not defined intrinsic factors. However, the capability of tumor formation of BMCs was maintained in vitro, probably suggesting the presence of different tumorigenic phenotypes.

**NGFR cooperates with a network of progressive-genes to mediate cell migration and invasion**

The BRAFi/MEKi-mediated selection of NGFR⁺ cells suggests higher survival properties of this cellular subset. To unravel *NGFR*-driven properties of MBM, we used a customized shRNA targeting the exon 3 of *NGFR* to perform a stable, doxycycline (DOX)-inducible knockdown (KD) in conventional melanoma cells (A375, WM35) and BMCs ~7–14 days after DOX-treatment (Fig. 7a, upper panel). Expression profiling of BMC1-M1 with a KD of NGFR, identified 339 (FClog2 ≤ −1.14;

$p ≤ 0.05$) down- and 193 (FClog2 ≥ 1.13; $p ≤ 0.05$) up-regulated genes among them 33 genes that were commonly downregulated in BMC1-M1 cells and in a BRAF/NRAS[wt] LN-MET-derived cell line (T2002, GSE52456, Supplementary Fig. 10a). Our survey identified *SOX4*, a master-regulator of EMT[76,77] and *PTPRZ1*, a mediator of stemness in glioblastoma[78] (Fig. 7a, lower panel) among the most significantly downregulated genes (Supplementary Data 8).

The expression of *NGFR* was correlated with a suspension phenotype of melanoma cells[39] and frequently observed in BMCs. Suspension cells (SCs) were viable and serial transplantation established adherently growing NGFR⁺ cells (Supplementary Fig. 10b, upper panel). The phenotype was more evident in BMC1-M4 ($10.3 ± 0.7\%$) than in BMC1-M1 ($3.6 ± 2.2\%$) cells (Supplementary Fig. 10b, lower panel) and was significantly reduced ($3.3 ± 1.8\%$, $p = 0.013$) by the KD of *NGFR* (Supplementary Fig. 10b, lower panel). To assess whether the modulation of NGFR levels affected invasion and 3D- spheroid formation, $2.5 × 10^3$ of either control (-DOX) or cells with a validated KD of NGFR (+DOX, NGFR[KD] cells) were seeded on a matrigel layer (Fig. 7b, left scheme). Control cells invaded the matrigel layer and formed spheroids after ~7 days that in turn shed cells into the matrigel and a minority of cells that likely featured a highly invasive phenotype completely crossed the matrigel layer and attached to the vessel´s bottom, not featured by NGFR[KD] cells (Fig. 7b, center panels). The concomitant expression of GFP enabled the tracing of cells upon DOX induction (Fig. 7b, center panels; inlaid and Supplementary Fig. 10c, left panels). Downregulation of *NGFR* expression significantly ($p = 3.4e−12$) reduced the diameter of spheroids (Fig. 7b, right panel, Supplementary Fig. 10c, center panel). Finally, we assessed the migratory capacity of BMCs with and without *NGFR* KD in a live cell imaging-based scratch-wound assay. We determined the relative wound density, reflecting a value which is normalized for potential changes in cell density caused by proliferation. We observed a rapid wound closure and increase in the relative wound density (RWD) by BMC2 ($69.9 ± 11.1\%$), BMC1-M4 ($45.3 ± 0.5\%$) and BMC4 ($42.5 ± 5,9\%$) but a reduced migratory capacity of BMC3 ($38.0 ± 3.1\%$) and BMC1-M1 cells ($26.4 ± 1.0\%$) 24 h after wounding (Fig. 7c, first panel).

Nevertheless, the migratory phenotype of BMC1-M1 cells, was significantly ($p = 3.0e−03$) reduced upon *NGFR* downregulation and decreased in BMC2 cells upon knockdown but without statistical significance (Fig. 7c, center panel, Supplementary Fig. 10c, right panel). As we could not fully exclude that a high proliferative phenotype affected wound closure we blocked cell proliferation by mitomycinC (MMC). Although we observed a decreased migratory capacity of BMC1-M1 cells at a later time point in the presence of MMC, we found this effect was caused by toxicity of the inhibitor (Supplementary Fig. 10d). MMC was ineffective after short time incubation (1 h).

The knockdown of NGFR revealed reduced levels of drivers of EMT cell migration such as *SOX4*[79], *MET* receptor (MET)[80], and *TCF19*[81] and surprisingly increased the level of *AXL* (Fig. 7c, right panel) in

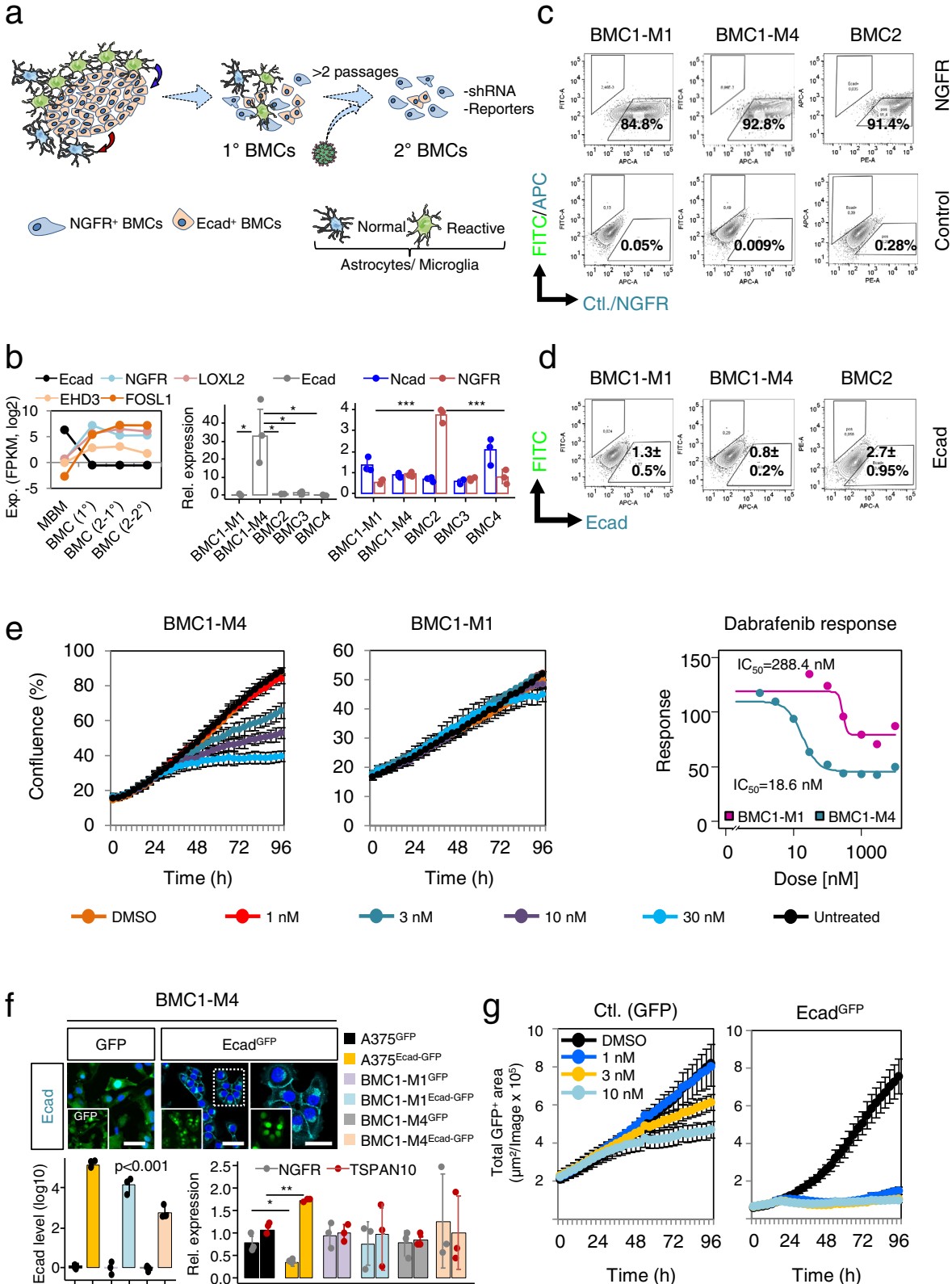

*NGFR*^KD cells. *SOX4* served as pan-MBM marker, hence was observed in all MBM investigated (FClog2 = 4.44 ± 0.97; range: 2.6–6.3) but at lower levels in BC (FClog2 = 0.45 ± 1.64; range: −1.04−2.21; *p* = 0.044) (Supplementary Fig. 10e, left panel) and was sustained in BMCs (Fig. 7d, upper panel). The ubiquitous expression in melanoma metastases suggests a general role of *SOX4* for cell maintenance in vivo and in vitro. To dissect the role of the transcriptional regulator, we

performed a KD of *SOX4* in BMCs (BMC1-M4 and BMC4) and tracked changes in proliferation for 5 days (Fig. 7d, scheme). We observed that levels of *NGFR* and *FOSL1* expression were not affected after 10d of DOX treatment in *SOX4*^KD cells (Fig. 7e, left panel), however cells exhibited a significant decrease in proliferation; BMC1-M4 (*p* = 9.8e −09; *p* = 5.7e−03) and BMC4 (*p* = 1.1e−06; *p* = 1.1e−04) (Fig. 7e, center and right panels; Supplementary Fig. 10e center and right panels),

**Fig. 5 | The brain microenvironment maintains an Ecad⁺ tumor cell phenotype.**
**a** Proposed in vivo-to-in vitro phenotype-switch. In vivo: MBM communicate with surrounding normal and reactive astrocytes/microglia via supplied soluble factors (astrocytes/microglia; blue arrow) or tumor cells (red arrow). **b** Left panel: Expression levels of indicated genes in Pat8/M1 (MBM) and stable, early (1°, +admixed stromal cells) or late (2°, without admixed cells) passage BMC1-M1 (BMC, MBM-derived cell line) cells. Center and right panels: qPCR of indicated BMCs for levels of Ecad, Ncad and *NGFR*. Mean ± SD expression levels of $n = 3$ biological replicates, normalized to β-actin (*ACTB*) are shown. Ecad and *NGFR* are most significantly expressed in BMC1-M4 ($p = 1.77e{-}02$) and BMC2 ($p = 9.62e{-}05$).
**c**, **d** Analysis of cell surface levels of NGFR and Ecad of indicated BMCs by flow cytometry, compared with non-stained control cells; 50,000 cells were recorded. A representative of $n = 3$ experiments is shown. **e** Left and center panels: Assessment of sensitivity of BMC1-M4 and BMC1-M1 cells towards increasing doses (1–30 nM) of dabrafenib. Data are presented as mean values +/−SD of $n = 8$ technical replicates

depicting confluence (%). A representative of $n = 3$ independent biological experiments is shown. Right panel: Dabrafenib $IC_{50}$ values of indicated BMCs. **f** Upper panels: Confocal imaging of BMC1-M4 cells indicating proper cell surface expression or absence of Ecad (turquoise) in Ecad-GFP or GFP transduced cells. DAPI served as nuclear dye. Scale bar, 50 μm. Lower panels: qPCR of indicated cell lines with overexpression of Ecad (Ecad^GFP) or GFP showing significantly different levels of Ecad ($p = 6.75e{-}05$, A375; $p = 6.38e{-}03$ BMC1-M1; $p = 6.84e{-}04$ BMC1-M4), *NGFR* ($p = 1.78e{-}02$) and *TSPAN10* ($p = 3.13e{-}04$). Mean ± SD values of $n = 3$ biological replicates and $n = 3$ independent cell lines are shown. **g** Assessment of dabrafenib sensitivity of BMC1-M4 cells either expressing GFP (Ctl.) or Ecad^GFP. Values represent mean ± SD of eight technical replicates; a representative out of $n = 3$ independent experiments of $n = 2$ independent cell lines is shown. In **b** and **f** statistically significant differences were tested by unpaired two-tailed *t*-test. Source data are provided as a Source Data file.

suggesting that *SOX4* acts downstream of *NGFR* and controls properties of brain metastatic melanoma cells.

## Global methylome profiling uncovered differentially methylated CpGs discriminating BRAF^mut and BRAF^wt MBM

Global methylome profiling serves as a prognostic tool for the comprehensive molecular classification of several primary brain tumors[82,83]. We applied 850k methylome profiling to unravel the epigenetic landscapes of MBM ($n = 20$) and to assess potential molecular subgroups. To exclude that the clustering was affected by admixed brain-derived stromal cells, we calculated the tumor cell content based on the expression levels of *PRAME* (Preferentially Expressed Antigen in Melanoma) observed in melanoma cell lines. *PRAME* was comprehensively expressed in MBM (median: 79.8%, range: 119.4–72.8%) with one exception (Pat10) that showed no *PRAME* expression and was classified as primary central nervous system (CNS) melanoma. *PRAME* was not expressed in brain-derived stromal cells and several MBM exhibited even higher levels than melanoma cell lines (Supplementary Fig. 11a). Next, we queried whether the subsets Ecad^high vs low, TIL^high vs low and BRAF^mut vs wt may be further classified by a set of differentially methylated genes (DMGs). We identified a significant (FDR-adjusted $p$-value < 0.05) difference in 46 CpG positions in promoters, islands, shelfs and shores of 35 genes that were hypomethylated in BRAF^mut tumors (Fig. 8a, left panel, Supplementary Data 11). Clustering of MBM regarding β-values of each CpG area revealed a highly significant ($p < 2.2e{-}16$) separation of BRAF^mut and BRAF^wt/NRAS^mut tumors (Fig. 8a, right panel). We queried the EWAS data hub (https://ngdc.cncb.ac.cn/ewas/datahub/index) and ranked the identified candidates in terms of their association with melanoma survival. Primarily the differential methylation of a CpG island encompassing the region 53,197,471–53,197,983 within the *ITGB7* (integrin subunit beta 7) gene (Supplementary Fig. 11b) was associated with survival of melanoma patients. We validated the differential methylation status of this particular CpG island using single nucleotide primer extension (SNuPE)[84], that revealed hypomethylation in BRAF^mut ($0.126 \pm 0.089$) but hypermethylation ($0.370 \pm 0.204$) in BRAF^wt/NRAS^mut MBM (Fig. 8b, $p = 0.053$ and Supplementary Fig. 11c). Next, we investigated the methylation status of *ITGB7* in the TCGA-SKCM data set that well represented our cohort of BRAF/NRAS mutated tumors. We observed hypomethylation of *ITGB7* and *SUSD3*, another potential mediator of cell-cell interaction, in BRAF^V600E/K mutated but hypermethylation in BRAF^wt/NRAS^mut tumors. Hence, we independently validated a potential association of the BRAF/NRAS mutation state and methylation status of *ITGB7* and *SUSD3* in a broader context (Fig. 8c). Concordantly, the expression of *ITGB7* but not *SUSD3* was moderate but significantly increased ($p = 0.023$; $p = 0.062$) (Supplementary Fig. 11d, left and center panels). However, we observed a significant difference in *ITGB7* expression only among EM but not PT. The matching of *ITGB7* expression in our MBM cohort with methylation data unexpectedly

revealed only a low association ($R = 0.23$, $p = 0.0843$) as *ITGB7* was primarily expressed in TIL^high MBM (Fig. 8d, left panel). This finding was validated in the TIL^high TCGA-SKCM melanoma cohort (Fig. 8d, right panel) and indicated a correlation with immune but not tumor cells in MBM. Moreover, T cell markers, *CD3E* (naïve) and *CD8A* (cytotoxic) significantly correlated with expression of *ITGB7* (Fig. 8e, left panel, Supplementary Fig. 11d, right panel). Finally, the expression of *ITGB7* but not *SUSD3* was associated with a favorable outcome and like the TIL^high state might serve as a potential prognostic marker in BRAF^V600 mutated MBM.

## Discussion

Despite ongoing progress, the long-lasting therapeutic control of brain metastasis remains challenging and the majority of patients exhibit a poor prognosis due to intracranial progression that is associated with neuroinflammation and the formation of multiple brain metastases (BM). In particular, molecular mechanisms controlling formation of the latter are poorly understood and might involve (i) the establishment of a founder clone that gives rise to multiple subclones or (ii) the transition of dormant micrometastases into actively proliferating macrometastases triggered by microenvironmental cues and/or therapeutic interventions[3,85,86].

Here, we performed a molecular, genetic and epigenetic profiling of MBM of stage IV melanoma patients (Fig. 9a) and uncovered at least two different molecularly distinct subgroups. Ecad^high MBM featured a proliferative and likely drug-naïve and/or drug-responsible phenotype and GSEA suggests that this subgroup but not Ecad^low/NGFR^high tumors probably depends on oxidative phosphorylation (OXPHOS) as previously delineated[87]. Our hypothesis is strengthened by the uncovering of molecular features of drug-naïve MBM by Rabbie et al.[88] and Biermann et al.[89], showing that this set of MBM expressed Ecad-associated genes such as *ABCB5* or pigmentation-related factors such as *MLANA*, *PMEL* or *DCT* found in either of the studies (Fig. 9b). Indeed the top-expressed genes as identified in the studies overlapped with the Ecad^high signature that was identified in this study, suggesting that drug-naïve MBM feature an Ecad⁺ rather than NGFR⁺ state. Surprisingly, gene signatures of both studies did not overlap, which might be a consequence of the different methods used (bulk vs. sc-RNAseq) or different developmental states of tumors of both cohorts. Moreover, Biermann et al. did not observe enrichment of *AXL*-associated genes in drug-naive MBM. Certainly, we found *AXL* among the top enriched genes in NGFR⁺ MBM in a previous study[23]. The molecular analysis of NGFR⁺ MBM revealed a downregulation of OXPHOS and strongly suggests a subclass of tumors that is distinct from Ecad^high MBM. Previous studies already proposed that NGFR⁺ melanoma feature a stem-like phenotype exhibiting cellular plasticity and drug-resistance. In line with this, we observed a higher expression of MRD-associated genes in NGFR^high than NGFR^low tumors, indicating that NGFR⁺ melanoma

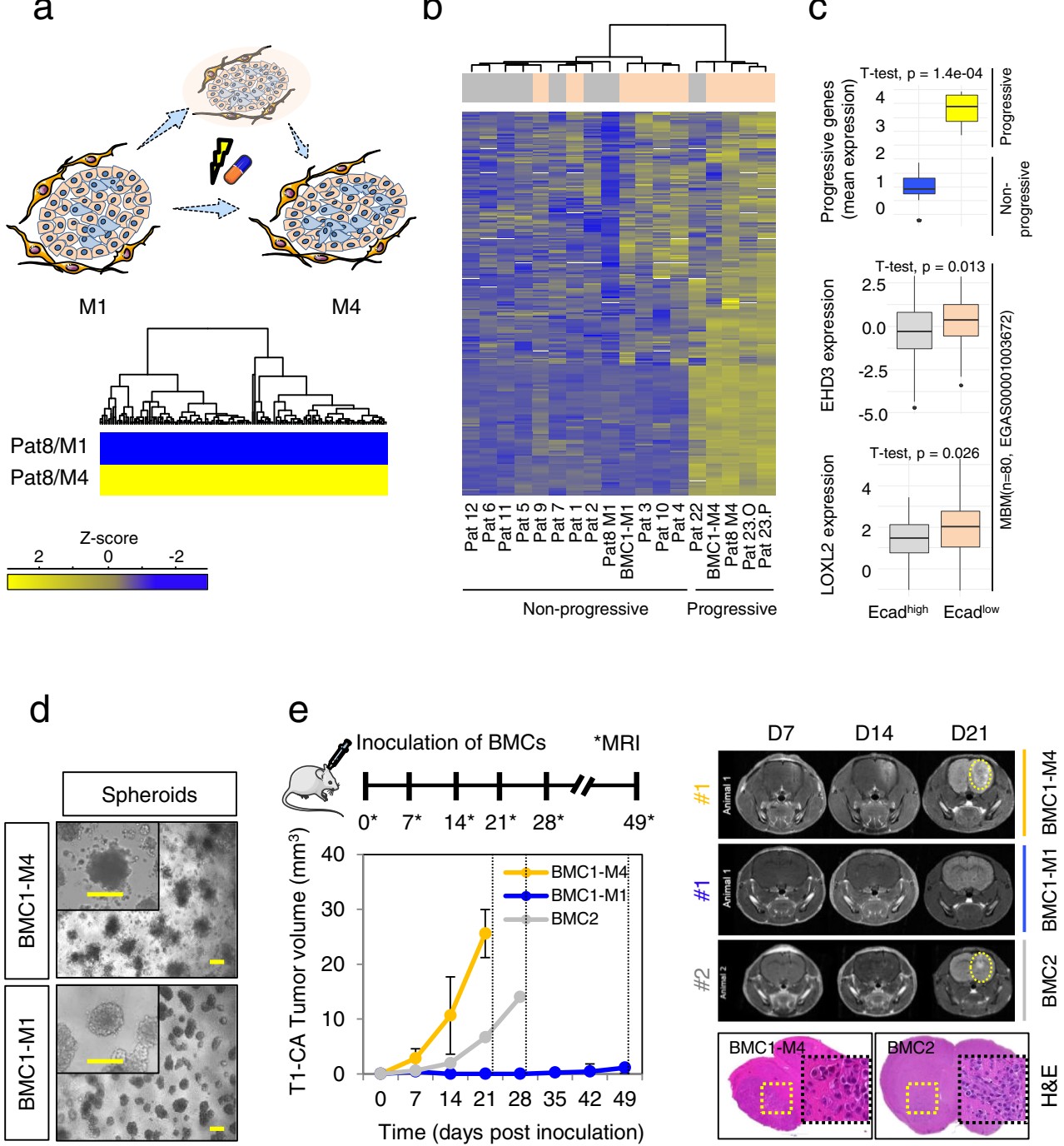

**Fig. 6 | A gene signature defines progressive stages of MBM. a** Pat 8/M1 and M4 tumors represent genetically concordant subclones that emerged sequentially or independent. M1 only received XRT/ICi but not BRAFi/MEKi therapy; M4 emerged under BRAFi/MEKi and progressed under ICi therapy, hence M1 and M4 are termed non/less-progressive and resistant/progressive tumors. Lower panel: The comparative profiling and analysis of both uncovered a signature 1063 DEGs and a core signature of 389 genes. **b** Clustering of MBM regarding the expression of core genes subdivides progressive and non-progressive tumors. **c** First panel: Box plot shows mean expression values of signature genes of progressive and non-progressive tumors. Center and bottom panels: Box plots showing expression of progressive genes *EHD3* and *LOXL2* in an independent set of Ecad^high and Ecad^low MBM (*n* = 80, study EGAS00001003672). **d** Embedding of indicated BMCs in matrigel established three-dimensional growing spheroids and demonstrated differences of invasive phenotypes. Bars indicate 50 μm. **e** Upper scheme:

Experimental set-up of in vivo experiments. CD-1 nude mice (*n* = 3/group) were stereotactically inoculated with $2.5 \times 10^4$ BMC1-M1, BMC1-M4 and BMC2 cells on day 0. Tumor volumes were determined over time for up to 21d by MRI in T1 (lower left panel). Data are presented as mean values +/−SD of *n* = 3 tumors per cell line indicating tumor volume ($mm^3$). Right panels: Representative MRI images taken 7, 14 and 21 days (D) after transplantation depict tumor formation (dotted yellow line) in BMC1-M4 and BMC2 but not BMC1-M1 inoculated and contrasted animals. The experiments were terminated once a tumor volume of at least 20 $mm^3$ was reached or at latest on day 49. Bottom panels: H&E staining of established tumors of BMC1-M4 and BMC2 cells after 21 and 28 days, inlays depict a distinct cellular morphology. In **c** box and whisker plots show median (center line), the upper and lower quartiles (the box), and the range of the data (the whiskers), including outliers and significance was determined by unpaired two-tailed *t*-test. Source data are provided as a Source Data file.

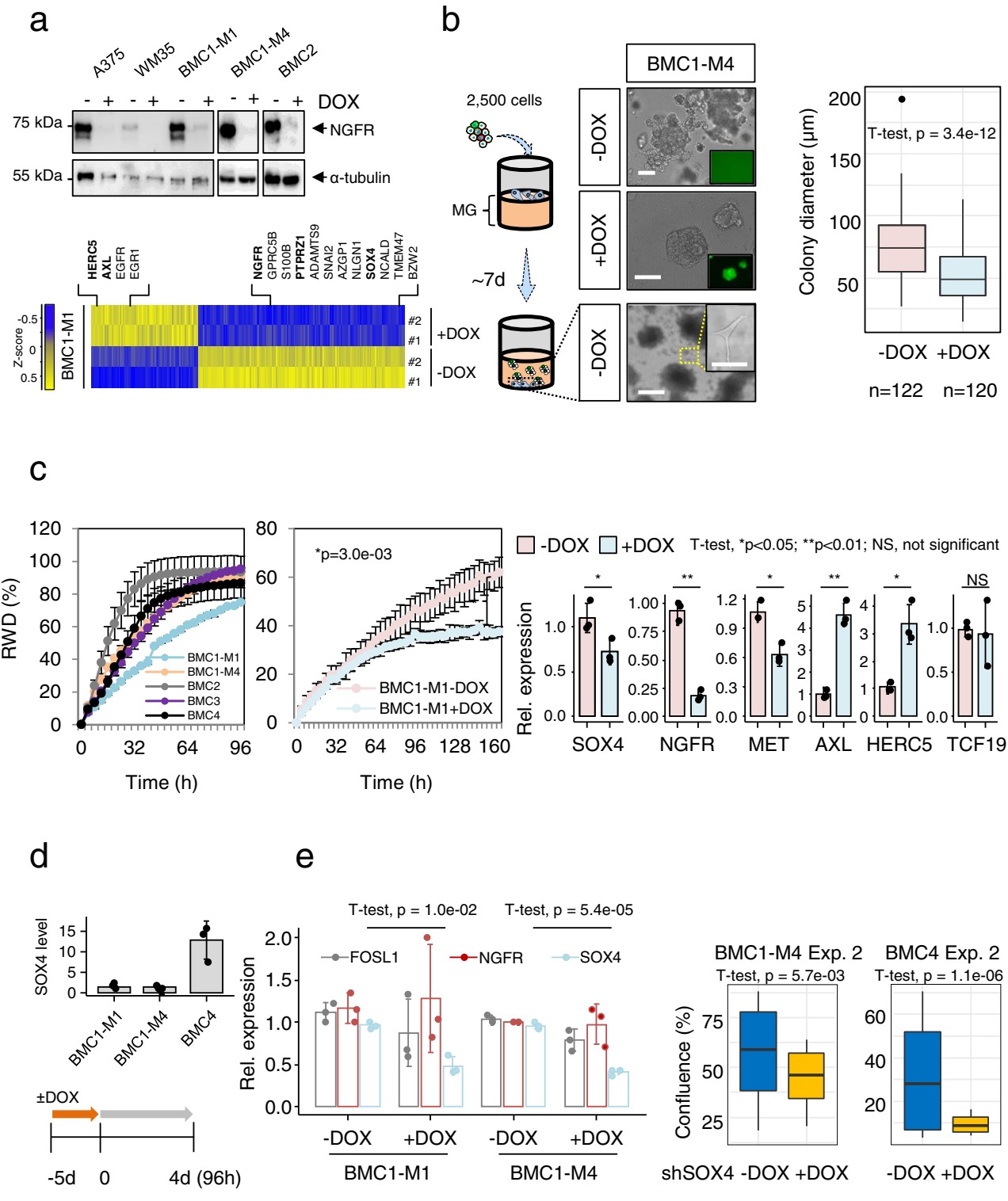

cells are indeed responsible for intracranial relapse and progression and might emerge under hypoxic conditions[38,90].

Although we found a mutually exclusive expression pattern of Ecad- and NGFR-correlated genes, a subset of MBM featured expression of genes of both signatures and immunofluorescence analyses and reporter-based assays suggest that Ecad and NGFR are connected via cellular plasticity. Therefore Ecad⁺ melanoma cells can turn into NGFR⁺ cells and vice versa. This transition likely includes an intermediate cell state. The mechanisms controlling the emergence, maintenance and drug-response of cells that participate in this phenotype switching process remain elusive. We observed that the Ecad-to-NGFR switch is probably fostered by therapeutic interventions, particularly BRAFi/MEKi. The analysis of pre- and post-BRAFi/MEKi and relapsed melanoma showed decreased levels of Ecad and associated genes and increase in *NGFR* and NGFR-associated genes. However, this analysis is limited due to the small sample size and number of patients that have been investigated.

**Fig. 7 | NGFR and SOX4 control migratory and proliferative properties. a** Upper panel: Immunoblotting of conventional: A375, WM35 and MBM-derived cell lines (BMCs) for levels of NGFR following DOX-treatment. α-tubulin served as loading control. Lower panel: Heat map representation of DEGs in BMC1-M1 cells (±DOX). Up- and down regulated genes are indicated, only significantly regulated genes (Bonferroni corrected, $p < 0.05$) are shown. Left scheme: The seeding of 2500 BMC1-M4 cells (±DOX) on a solidified layer of matrigel enabled the investigation of cell invasion. Center panels: NGFR-dependent phenotypical changes of spheroids. Bottom panels: Invasive (−DOX) cells crossed the matrigel layer and attached to the bottom of wells. Insets show GFP expression upon DOX treatment, bars indicate 50 μm. Right panel: Box plots depicting the diameter (μm) of spheroids. **c** Left panel: Changes in the relative wound density (RWD, %) depict distinct migratory phenotypes of indicated BMCs. Center panel: Migration assay of BMC1-M1 cells ±DOX. Values represent mean ± SD of $n = 3$ independent biological replicates. Right panel: qPCR of BMC1-M4 cells with *NGFR* KD: levels of *SOX4* ($p = 3.40e-02$); *MET*

($p = 5.18e-02$) and *NGFR* ($p = 9.16e-03$); *AXL* ($p = 8.89e-03$) and *HERC5* ($p = 4.29e-02$) and *TCF19* (NS) are shown. **d** Upper panel: qPCR illustrates levels of *SOX4* in indicated BMCs. In **c, d** values represent mean ± SD of $n = 3$ independent biological replicates of independently established cell lines. Lower panel: experimental scheme of DOX treatment of BMC1-M4 and BMC4 cells prior to live cell imaging-based assessment of proliferation for 4d. **e** Left panel: qPCR indicating significantly reduced levels of *SOX4* but not *NGFR* and *FOSL1* in *SOX4*[KD] cells (BMC1-M1; $p = 1.0e-02$ and BMC1-M4; $p = 5.4e-05$). Right panels: proliferation of BMC1-M4 ($p = 5.7e-03$, center panel) and BMC4 cells ($p = 1.1e-06$, right panel) with and without *SOX4* KD. Relative expression is presented as mean ± SD of $n = 3$ independent biological replicates, normalized to *ACTB* and related to −DOX controls. In **b, e** box and whisker plots show median (center line), the upper and lower quartiles (the box), and the range of the data (the whiskers), including outliers. In **b, c, e** significance was determined by unpaired two-tailed *t*-test. Source data are provided as a Source Data file.

The comparison of metachronous MBM that faced different therapeutic interventions and even developed under therapy provides insights into the mechanisms that could potentially foster therapy resistance and establishment of MRD. Although we cannot determine whether the tumor M4 that developed after BRAFi/MEKi and during ICi therapy directly derived from M1, TargetSeq demonstrated a common genetic background. However, both tumors might have acquired additional mutations that have not been picked up by TargetSeq due to limitations of the amplicon panel. The comparative expression profiling uncovered a gene signature that was likely associated with intracranial progression. This process was at least partly resembled by the derivation of BMC1-M1 cells from M1. Both processes featured a loss of Ecad expression and gain in expression of particularly *NGFR* and *EHD3*; hence genes that were found increased in relapsed tumors and in addition were correlated with MRD that developed in a xenograft model[24].

Tumor derived cell lines BMC1-M1 and BMC1-M4 exhibited distinct proliferative and migratory properties, probably the consequence of the loss of microenvironmental cues such as growth factors and cytokines or adhesive connections or modulation of levels of NGFR expression[39]. Thus, we assumed that the transfer of BMCs back into a versant environment might equalize the properties of both. Nevertheless, we observed that the cellular properties were not affected by murine BTE cells, or presence of a non-BRAF[V600] mutation, suggesting a potential role for as yet undefined intrinsic factors. However, as BMCs were injected in a non-inflamed environment, our experiment did not provide information on the role of interacting reactive astrocytes or microglia and secreted factors that likely contribute to the patient tumor microenvironment.

Our findings so far suggest that melanoma cells can re-acquire an Ecad⁺ phenotype that is either maintained during metastasis or lost but re-established at distant organ sites such as the brain. Recently, Ecad expression was demonstrated to be required for breast cancer metastasis[91], suggesting that the features of Ecad[high], Ecad[low] or NGFR⁺ tumors, might be controlled by different molecular programs. MBM of both subsets showed different molecular features and might differentially respond to therapeutic interventions as suggested by the overexpression of Ecad in an established MBM-derived cell line. Indeed, Ecad-OE cells were more sensitive to BRAFi, hence, Ecad⁺ tumors likely precede an NGFR⁺ state of MBM. Whether NGFR⁺ primary tumors can adopt an Ecad⁺ phenotype once cells have entered the brain remains unknown. Considering that the expression of NGFR in melanoma cells accompanies a network of associated genes such as mediators of a NCSC-phenotype[34,92,93] - which was recently identified as a major driver of MRD[24] - we propose that the upregulation of NGFR and associated genes may indicate the fate of relapse in the brain and intracranial progression.

It is likely that the interaction of tumor cells with reactive astrocytes, microglia and TILs can determine the timing and fate of

intracranial progression. Generally, a high TIL status of MBM is associated with favored survival. We analyzed expression levels of T cell markers and observed that the expression of CD3D sufficiently separated MBM into TIL[high] and TIL[low] tumors. Moreover, TIL[high] and progressive tumors were clearly separated, again validating the favorable effect of infiltrated lymphocytes. However, high levels of NGFR were associated with an immune suppressive phenotype[94]; but we observed high expression of NGFR even in TIL[high] tumors, suggesting no direct correlation at least in MBM. Although, we investigated only a small cohort, our findings suggest a T cell-triggered process of de-differentiation mediated via secreted TNFα[20,95] in TIL[high]/NGFR⁺ MBM. As we also observed expression of Ecad in TIL[high] MBM, the T cell mediated dedifferentiation likely presents a spatiotemporal process that also depends on the activity status of microglia and astrocytes. However, the latter mechanism is poorly understood and requires further investigation. Besides the molecular separation regarding the levels of Ecad, NGFR or TILs, we observed that mutations in BRAF and NRAS genetically and epigenetically separated MBM. Our analysis identified differentially methylated sites in tumor or immune cell expressed genes, particularly *ITGB7* and suggest that *NRAS* mutated MBM might feature a more progressive phenotype as observed before in primary melanoma[96]. We found a correlation of *ITGB7* with expression of T cell markers and although the role of *ITGB7* is poorly understood, increased levels of ITGB7⁺ leukocytes correlated with favored survival in a colorectal cancer mouse model due to blocking of tumorigenesis and progression[97]. Hence, *ITGB7* expression might serve as a prognostic marker in melanoma and potentially help to predict MBM progression states.

In summary, our study provides evidence that Ecad and NGFR-controlled programs molecularly subdivide MBM, suggesting the presence of at least two different cell states that may uniquely respond to therapeutic interventions. The Ecad-to-NGFR switch probably presents a hallmark in the progression of MBM and temporally subdivides tumors into progressive and non-progressive. However, our study is limited by the relative small number of tumors due to the limited access to MBM. Hence additional studies are needed to fully unravel the molecular programs that drive the emergence and progression of solitary and multiple brain metastases in melanoma and other cancers showing a high incidence for brain metastasis such as lung cancer and breast cancer.

## Methods
### Patient cohorts
All procedures performed in this study were in accordance with the ethical standards of the respective institutional research committees and with the 1964 Helsinki declaration and its later amendments or comparable ethical standards. All patients gave written informed consent for the collection and scientific use of tumor material which

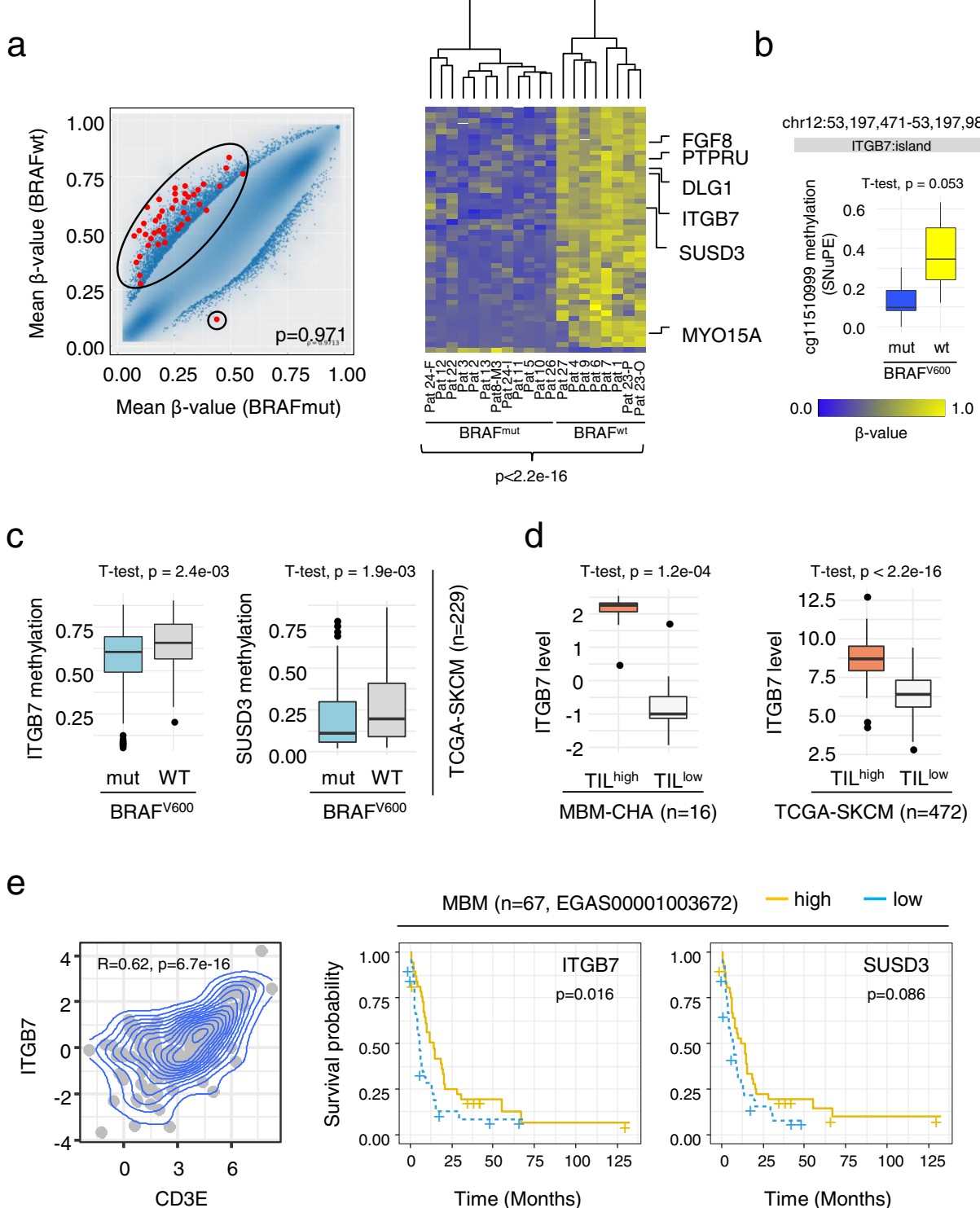

was collected at the Biobank of the Charité – Comprehensive Cancer Center (CCCC) following ethics approval by the Ethics Committee of the Charité (EA1/152/10; EA1/107/17; EA4/028/18). In addition, thirty-two MBM archived at the Department of Neuropathology, Charité-Universitätsmedizin Berlin, Germany, were included in the study and analyzed. The usage of archived (FFPE) melanoma and central nervous system-derived control samples (pons, cortex, cerebellum) has been reviewed and approved by the local ethics committee (EA1/107/17 and EA1/075/19). Intraoperative brain metastases of sixteen

patients with diagnosed stage IV melanoma (MBM) were surgically removed at the Department of Neurosurgery, Charité – Universitätsmedizin Berlin, Germany. Tumor pieces were split into parts of equal size and i) snap-frozen in liquid nitrogen and stored at −80 °C for later isolation of RNA and DNA, and ii) formalin-fixed and paraffin-embedded (FFPE) for (neuro-)pathological work-up. If additional sufficient material was available, it was used for the establishment of MBM-derived cell lines (BMCs). CSF of patient 8 was collected by lumbar puncture.

**Fig. 8 | Global methylome analysis uncovered CpGs discriminating BRAF$^{V600E/K}$ and NRAS$^{mut}$ MBM. a** Left panel: Scatter plot representation of β-values of BRAF$^{V600E/K}$ vs BRAF$^{wt}$ MBM, identified 46 differentially methylated CpGs in islands, shelfs and shores as indicated (red dots). Significance was determined by a unpaired two-tailed $t$-test and correction for multiple testing (Bonferroni adjusted $p < 0.05$). Right panel: Heat map representation of β-values of identified CpGs of 36 genes demonstrates a distinct ($p < 2.2e−16$) separation of BRAF$^{V600E/K}$ and BRAF$^{wt}$ MBM. All wt tumors harbored NRAS mutations. The level of methylation is color coded (blue, yellow) and genes showing a correlation of methylation and survival are indicated. **b** SNuPE (single nucleotide primer extension) revealed a significant ($p = 0.053$) difference in the methylation level within a CpG island covering exons 4–5 of the ITGB7 gene (probe cg11510999) of BRAF$^{V600E/K}$ ($n = 9$) and BRAFwt ($n = 4$) tumors. In total $n = 14$ biologically independent tumors were investigated. **c** Box plot representation of β-values of *ITGB7* and *SUSD3* of BRAF$^{V600}$ and BRAF$^{wt}$/NRAS$^{mut}$ tumors (TCGA-SKCM, $n = 229$) showing differential methylation in *ITGB7* ($p = 2.4e$

−03) and *SUSD3* ($p = 1.9e−03$). **d** Left panel: Expression levels of *ITGB7* in TIL$^{high}$ and TIL$^{low}$ MBM (left panel, $n = 16$) and TCGA melanoma showing increased expression of ITGB7 in TIL$^{high}$ subsets, MBM-CHA ($p = 1.2e−04$) and TCGA-SKCM ($p < 2.2e−16$). **e** Left panel: Dot plot showing a significant ($p = 6.7e−16$) correlation ($P = 0.62$) of expression of *ITGB7* and of the marker of naïve T cells *CD3D* of melanoma (primary, metastases; TCGA-SKCM set). Center and right panels: Survival probabilities of MBM of an independent data set ($n = 67$, study EGAS00001003672) with regards to levels of *ITGB7* and *SUSD3*. High levels of *ITGB7* but not *SUSD3* are associated with favorable survival. Expression levels (high, low) are color coded. Significance was determined by log-rank test, p-values are not corrected for multiple testing. In **b**–**d** box and whisker plots show median (center line), the upper and lower quartiles (the box), and the range of the data (the whiskers), including outliers and significance was determined by unpaired two-tailed $t$-test. Source data are provided as a Source Data file.

## Cell culture

**Conventional melanoma cell lines.** Conventional melanoma cell lines A375 (CRL-1619), WM35 (CRL-2807, Discontinued) and MeWo (HTB-65) were purchased from ATCC and kept at 37 °C/ 5% CO$_2$ and 95% humidity in cell culture medium (DMEM, 4.5 g/L glucose, stabilized glutamine/GlutaMax, pyruvate, Gibco/ThermoFisher) supplemented with 10% fetal bovine (FBS, Gibco) serum and 1% penicillin/streptomycin (P/S) (Gibco/ThermoFisher). T2002 cells were established from a intraoperative lymph-node metastasis as previously reported[33].

**MBM-derived cell lines (BMCs).** Intraoperative tumors were surgically resected during routine craniotomy and processed to establish BMCs as following: Tumor pieces were stored in physiological saline, 0.9% on ice until further processing. Following mincing using scalpels, the mechanically dissociated tissue was transferred to a 15 ml falcon tube containing trypsin/EDTA (0.05%) and incubated at 37 °C in a water bath for up to 20 minutes. In addition, the tissue was mechanically dissociated by usage of a Pellet Mixer (VWR International). The cell suspension was applied to a 70 μm cell strainer to remove undigested tissue fragments and cells in the flow-through were collected by centrifugation at 330 g for 5 min. Collected cells were resuspended in cell culture medium (DMEM, 4.5 g/L glucose, stabilized (GlutaMax) or conventional glutamine, pyruvate, 10% FBS, 1% P/S) and seeded on appropriate cell culture dishes. Cells were maintained for at least three days without medium change to achieve optimal recovering and attachment of tumor cells. BMCs were kept at low passages (2–20) and split according to their proliferative capacity (1:2–1:10) at a confluence of ~80%.

## BrdU labeling

For labeling, cells were maintained for 2 h in medium containing BrdU (Becton&Dickinson) at a final concentration of 2 mM. Subsequently, cells were washed with phosphate buffered saline (PBS) and fixed with freshly prepared paraformaldehyde (4% in PBS) for 10 min at room temperature and washed and permeabilized by Triton-X100 (0.1 %/PBS). Cells were then treated with hydrochloric acid (2 M) for 10 min and washed twice with PBS. For BrdU detection, labeled cells were incubated with anti-BrdU-AlexaFluor488 for 1 h at room temperature or overnight at 4 °C and washed with PBS-Tween20 (PBST; 0.1%/PBS). Images were taken with Leica fluorescence microscope (Zeiss Axioskop 2) and edited with Adobe Photoshop 2020 using the gradation curve and picture size function. Images were adjusted to a resolution of 600 dpi (RGB). BrdU-positive cells were quantified by counting and related to the total number of cells.

## Flow cytometry/Fluorescence-activated cell sorting (FACS)

After removal of medium, cells were washed with PBS and harvested by Trypsin (0.05% Trypsin/EDTA). Following addition of cell culture medium, cells were collected by centrifugation at 330 g at room temperature for 3 min and resuspended in 100 μl of ice cold buffer (PBS/0.5% bovine serum/2 mM EDTA) and stored on ice. Cells were incubated with fluorescently labeled primary antibodies against NGFR (anti-CD271-AlexaFluor647, BioLegend, catalog number: 3345114; clone: ME20.4; LOT#:B266105, mouse IgG1κ, dilution: 1:80); NGFR (anti-CD271-PE, BioLegend, catalog number: 3345106; clone: ME20.4; mouse IgG1κ, dilution: 1:80); Ecad/CDH1 (anti-CD324-AlexaFluor647 [recognizing the N-terminal domain of E-cadherin], BioLegend, catalog number:147308; clone: DECMA-1; rat IgG1κ, dilution: 1:80) and non-labeled antibodies against AXL (anti-AXL, Atlas Antibodies, catalog number: HPA037423, rabbit, dilution: 1:100) and MET (anti-MET, Cell signaling, catalog number: CST #8741; clone L6E7, mouse, dilution: 1:100) diluted in buffer according to the manufacturer's specifications and stored at 4 °C for 10 min to achieve proper labeling. Cells were washed by addition of buffer, collected by centrifugation and resuspended in 100 μl of buffer that contained secondary antibodies (AlexaFluor-488/594/647) and/or DAPI, diluted according to the manufacturer's specifications. After incubation for 10 min at 4 °C and washing, cells were resuspended in 500 μl PBS and analyzed by flow cytometry (Canto II) or fractioned by FACS using a FACSAria™III cell sorter (Becton&Dickinson, BD). FACS-isolated cells were collected in cell culture medium and seeded on appropriate vessels following centrifugation. Data analysis was performed with FlowJo (Ver 10.7.1).

## Immunophenotyping

**Immunofluorescence (IF) - sections.** Two micrometer sections of FFPE tumors were dewaxed and subjected to antigen retrieval with citrate buffer (10 mM, ph = 6.0) and heating for 20 min in a steamer. Cooled sections were blocked with blocking buffer (2% BSA/PBS) to reduce unspecific binding. Primary antibodies against NGFR (anti-p75$^{NTR}$, Cell signaling, catalog number: CST #8238, clone D4B3, LOT#:2, rabbit, dilution: 1:100); Ecad/CDH1 (anti-CD324; C-term, Cell signaling, catalog number: CST #3195, clone 24E10, LOT#:13, rabbit, dilution: 1:200); Ecad/CDH1 (anti-CD324, Santa Cruz, catalog number: sc-8426, clone G10, mouse, dilution: 1:50); Ecad/CDH1 (anti-CD324-AlexaFluor647, BioLegend, catalog number: 147308, clone DECMA-1, rat IgG1κ, dilution: 1:80); Melanoma Associated Antigen (anti-KBA.62, NovusBiologicals, catalog number: NBP2-45285, clone: KBA.62, LOT#:MSM1-895P180523, mouse IgG1κ, dilution: 1:100); GFAP (anti-GFAP-AlexaFluor594: BioLegend, catalog number: 644708, clone 2E1.E9, mouse IgG2b, dilution: 1:200); BrdU (anti-BrdU-AlexaFluor488, BioLegend; catalog number: 364105; clone 3D4; mouse IgG1κ, dilution: 1:20;) and MKI67 (anti-Ki67, Cell signaling, catalog number: CST #9449, clone 8D5, mouse IgG1, dilution: 1:200) were diluted in blocking buffer and incubated for 2 h at room temperature or overnight at 4 °C. After washing with PBST, secondary antibodies and DAPI (Merck, 508741, 2 mg/ml), all diluted to 1:500 in blocking buffer were applied to sections and incubated at room temperature for 1 h. Following

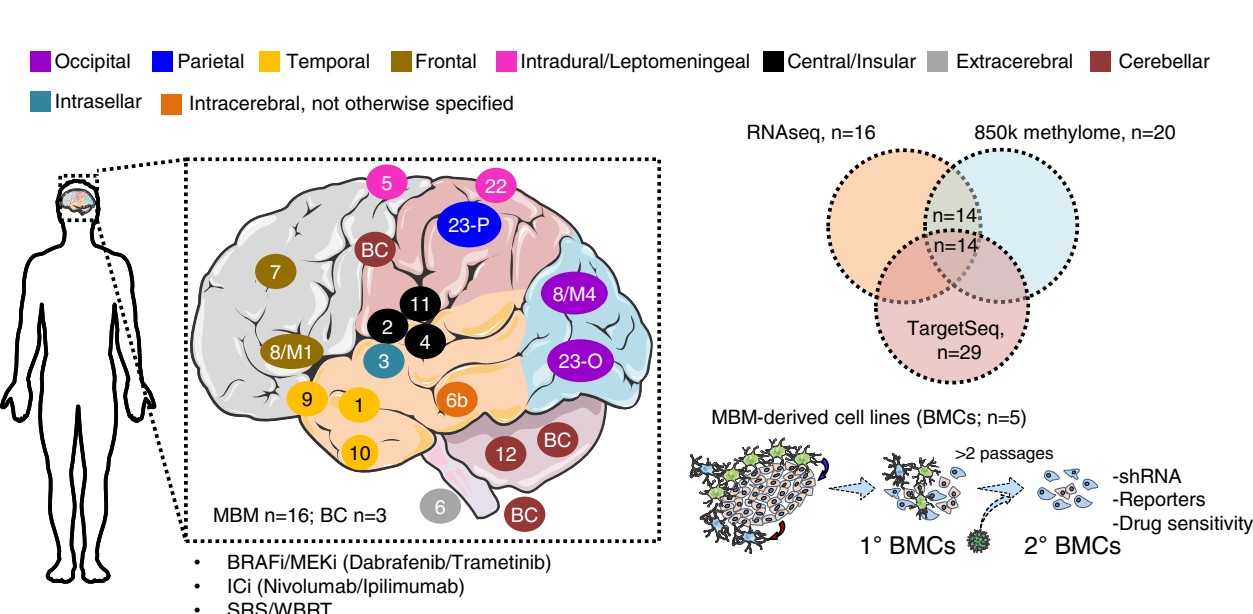

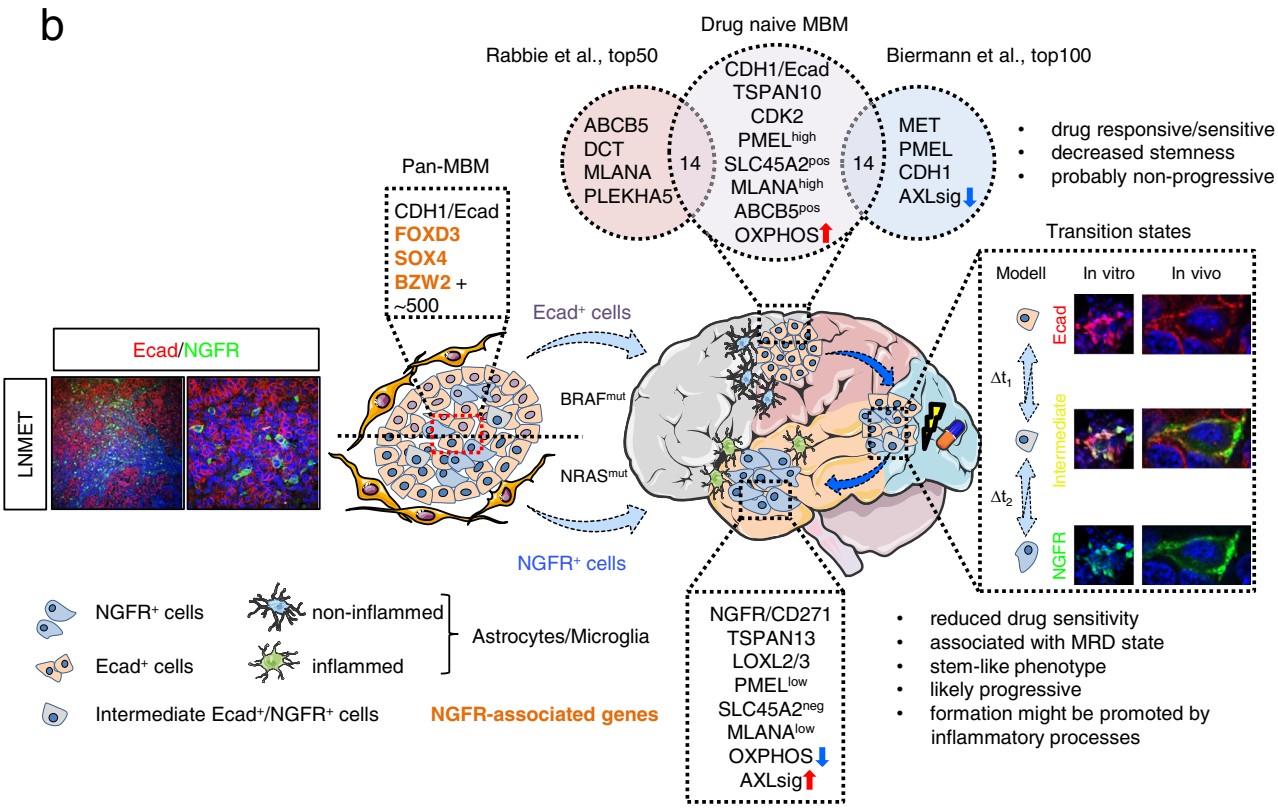

washing, sections were covered with mounting medium and cover slips and stored at 4 °C until fluorescence microscopy-based imaging.

**Immunofluorescence (IF) – cell culture.** Cells were seeded onto glass 8-chamber slides to a density of 5000–10,000 cells per chamber in cell culture medium and fixed in PFA (4% in PBS) at room temperature for 10 min, 24–48 h after seeding. After fixation, cells were washed once with PBS and blocked with bovine serum albumin (BSA, 2% in PBS) for 30 min at room temperature. Following,

primary antibodies against NGFR (anti-p75$^{NTR}$, Cell signaling, catalog number: CST #8238, clone D4B3, rabbit, dilution: 1:100); Ecad/CDH1 (anti-CD324; C-term, Cell signaling, catalog number: CST #3195, clone 24E10, rabbit, dilution: 1:200) and AlexaFluor555-Phalloidin (ThermoFisher, catalog number: A34055, dilution: 1:1000) were added in BSA, in above mentioned dilutions and incubated overnight at 4 °C. Next day, cells were washed twice with PBST and incubated with secondary antibodies and DAPI for 1 h at room temperature. Following washing, cells were covered with

**Fig. 9 | Expression of Ecad and NGFR probably subdivides drug-naïve and drug-resistant MBM. a** Schematic representation of our study cohort. We collected snap frozen and archived (FFPE) MBM specimen from different intracranial sites of late-stage melanoma patients who developed progressive intracranial disease under treatment such as BRAFi/MEKi/ICi and/or XRT (SRS or WBRT) and performed RNAseq (n = 16), global 850k methylome analysis (n = 20) and TargetSeq (n = 29). A set of n = 14 MBM was investigated by all approaches. Moreover, we established BMCs from n = 5 intraoperative MBM presenting in vitro models that enabled the analysis of molecular programs potentially controlling cellular dependencies by shRNA/drug-based perturbance and reporter-based tracking. **b** Melanoma cells exist in Ecad⁺, NGFR⁺ and intermediate states in extracranial tumors such as lymph node metastases (LN-MET) and probably primary tumors. Transcriptome profiling of MBM and BMCs with knockdown of NGFR revealed "Pan-MBM" markers, not expressed by BTE cells but generally found in melanoma and associated with NGFR mediating cell survival (orange, bold). Likely, Ecad⁺ and NGFR⁺ subsets are capable

of establishing MBM while retaining their initial phenotypes. The latter are potentially determined by the BRAF/NRAS mutation state, suggesting that Ecad⁺ or NGFR⁺ MBM are associated with a BRAF$^{V600}$ or NRAS$^{Q61}$ genotype. Intracranially, Ecad⁺ and NGFR⁺ MBM show different molecular features. A comparative analysis of MBM for representation of very recently identified gene signatures suggests a drug-naïve phenotype of Ecad⁺ tumors, featuring expression of particularly pigmentation/melanocyte-related genes (Venn diagram) and absence of AXL-signature (AXLsig) genes. Cellular plasticity probably controls the switching of Ecad⁺ into NGFR⁺ cells that likely feature a stem-like and drug-resistant/MRD state and show a low level of pigmentation/melanocyte-related genes. The Ecad-to-NGFR transition is likely triggered by neuroinflammation that in turn is mediated by reactive astrocytes and microglia and might be promoted by therapeutic interventions. Moreover, Ecad⁺ and NGFR⁺ may depend on different metabolic programs. OXPHOS may be the preferred source of energy in Ecad⁺ MBM (red arrow) but was minor represented (blue arrow) in NGFR⁺ tumors.

mounting medium (Vectashield, Avantor) and stored at 4 °C until imaging.

**Immunohistochemistry (IHC).** Automated histological staining was performed on the BenchMark Ultra platform (Ventana) or autostainer (Agilent) using primary antibodies against NGFR (anti-p75$^{NTR}$, Cell signaling, catalog number: CST #8238, clone D4B3, rabbit, dilution: 1:100); CD3 (anti-CD3ε, Agilent, catalog number: #A045201-2, rabbit, dilution: 1:100) and MKI67 (anti-Ki67, DAKO, catalog number: M7240, clone Mib-1, dilution: 1:100). Tumors were scored regarding the quantity of positive cells as determined by counting of 50 visual fields at 200x magnification and by relative levels of NGFR expression.

**RNA isolation and sequencing**
Isolation of total RNA from snap frozen tumors was performed with the RNAeasy extraction kit (Quiagen) according to the manufacturer's instructions. RNA integrity was determined by automated electrophoresis (4200 TapeStation system, Agilent). The library preparation of 100 ng total RNA was performed with TruSeq Stranded total RNA Sample Preparation-Kit and Ribo-Zero Gold Kit (Illumina). Paired-end (2x100 bp) whole transcriptome profiling of RNA with integrity numbers (RIN) ≥ 7 was performed at Cegat GmbH, Tuebingen (Germany) and sequenced on NovaSeq6000 platform. Illumina bcl2fastq (2.19) was used for demultiplexing of sequenced reads and adapter trimming was performed with Skewer (version 0.2.2)[98]. The information on FASTQ files was obtained using the FastQC program (version 0.11.5-cegat) read out. Raw sequencing data (fastq files) were quality controlled using fastqc (version 0.11.7 - Bioinformatics Group at the Babraham Institute) and further preprocessed with fastp[99]. Reads were aligned to the GRCh38 version of the human genome using TopHat[100] and counts per gene were calculated by the featureCount-algorithm from the Rsubread package[101]. All further steps of the analysis were done in R. Raw counts of protein-coding genes were normalized using the DESeq2 (https://bioconductor.org/packages/release/bioc/html/DESeq2.html) package[102]. Differential expression of genes between groups was determined after fitting models of negative binomial distributions to the raw counts. Raw p-values were FDR (false discovery rate)-adjusted for multiple testing and a value below 0.05 for the adjusted *p*-values were used to determine significant differentially expressed genes. Functional annotation of genes, over representation and gene set enrichment analysis were done using the clusterProfiler package[103]. For visualization of differentially expressed genes and molecular subgroups we used ComplexHeatmap[104] https://www.bioconductor.org/packages/release/bioc/html/ComplexHeatmap.html).

**Genetic profiling**
For amplicon-based targeted DNA-sequencing, 10–40 ng of DNA was isolated from stored snap frozen and archived FFPE tumor

tissue or from cell lines using the DNeasy Blood & Tissue or the QIAamp DNA FFPE Tissue Kit and used for library preparation. Sequencing of cancer hotspot (CHP2v, ThermoFisher) or TruSight Oncology 500 panel (Illumina) libraries was performed with benchtop sequencers IonProton (Thermo fisher) or NextSeq2000 (Illumina) with 775X mean coverage (Supplementary Data 6). Sequencing results were analyzed with VariantCaller software and validated using databases such as Varsome, COSMIC, and the 1000Genomes project[105]. Data of the latter enabled the separation of single nucleotide polymorphisms (SNPs) from mutations (single nucleotide variants, SNVs).

**Production of retroviral and lentiviral particles**
For production of lentiviral particles, LentiX cells were seeded to 1x10⁴ cells on a 10 cm dish and transfected after 24 h with 4 µg of plasmids either expressing the DOX (doxycycline)-inducible, SOX4 targeting shRNA (GEPIR Sox4.2137, Addgene #101119), wildtype E-cadherin (pHAGE-CDH1, Addgene #116722), Ecad-reporter (pHAGE-E-cadherin-RFP, Addgene #79603), NGFR 3'-UTR/miRNA reporter (ABM Inc. 3180008) or GFP/EGFP (Amsbio, IK-VB160109-10005) and 2 µg of pMD2.G (Addgene # 12259, VSV-G envelope) and 1 µg of psPAX2 (#12260) packaging plasmids using 20 µl/1 ml Polyethylenimine (PEI, Sigma-Aldrich). Medium was changed after 24 h and viral supernatant was harvested after additional 24 h. Viral supernatants were filtered through a 0.45 µm filter and applied to target cells for 24–48 h. The knockdown of NGFR was performed with a DOX-inducible shRNA (SMARTvector, Dharmacon, clone ID V3SVHS02_8785341). Transgene expression was induced and maintained with DOX at a final concentration of 2–4 µg/ml. Virally transduced cells were selected for puromycine (Puro) resistance using a final concentration of 10 µg/ml Puro. Stable selection was achieved after passaging and growth of cells in presence of Puro for ~3 passages.

**Methylome profiling**
For global methylome analysis, DNA of snap frozen or FFPE MBM was isolated according to standard procedures using the DNeasy blood and tissue DNAextraction kit (Qiagen, Hilden, Germany). 500 ng genomic DNA were subjected to bisulfite conversion using the EZ DNA Methylation-Gold Kit (Zymo Research) according to the manufacturer´s protocol. Subsequently, samples were analyzed on the Infinium MethylationEPIC Kit (Illumina) according to the manufacturer´s recommendations to obtain genome-wide data from 850,000 CpG positions. Raw data from Illumina Epic arrays were preprocessed and analyzed in the standard workflow of the packages RnBeads[106] and watermelon[107]. Differential methylation analysis was conducted on site and region level according to the sample groups regarding their levels of E-cadherin or NGFR expression or level of immune cell infiltration (TIL status) or mutation status of BRAF. For statistical analysis, *p*-values on the site level were

computed using the limma method. I.e. hierarchical linear models from the limma package were employed and fitted using an empirical Bayes approach on derived M-values.

## Validation of cg11510999 methylation by SNuPE and IP-RP-HPLC (SIRPH analysis)

Ten microliters of bisulfite converted DNA was used as the template in a 50-μL reaction in the presence of 1xHotStarTaq PCR buffer, 2.5 mM of MgCl2, 0.06 mM of each dNTP, 1.5 U HotStarTaq DNA polymerase (Qiagen), and 167 nM of primers (F: 5′-tttataaggtagtataggtttat-3′, R: 5′-ttactaaacaaaacttcacc-3′). PCRs were performed at 95 °C for 15 min followed by 47 cycles at 95 °C/60 s, 51 °C/45 s, 72 °C/45 s, and a final extension 72 °C/5 min. Ten μL of the PCR reaction was treated with 2U of ExoCIAP (mixture of Exonuclease I (Jena Bioscience) and Calf Intestine Alkaline Phosphatase (Calbiochem) for 30 min at 37 °C. To inactivate the ExoCIAP enzymes, the reaction was incubated for 15 min at 80 °C. Afterwards, 13 μL of primer extension mastermix (50 mM of Tris−HCl, pH 9.5, 2.5 mM of MgCl2, 0.05 mM of ddATP and ddGTP, resp., 1.6 μM of SNuPE primer (5′-atctaaactaac-3′), and 2.5 U of Termipol DNA polymerase [Solis BioDyne]) were added. Primer extension reactions were performed at 96 °C for 2 min, followed by 50 cycles at 96 °C/30 s, 50 °C/30 s, and 60 °C/20 s. Separation of products was conducted on an XBridge OST C18 2.5 μm 4.6 mm × 50 mm column (Waters) at 0.9 mL/min at 50 °C by continuously mixing buffer B (0.1 M TEAA, 25% acetonitril) with buffer A (0.1 M TEAA) over 8 min: 30−37%. The methylation index was calculated after measuring the heights (h) of the peaks applying the formula:

$$\%Meth = \frac{h(meth)}{h(meth) + h(unmeth)} \quad (1)$$

Meth = methylation index; h(meth) = peak height, methylated; h(unmeth) = peak height, unmethylated.

## Gene-set enrichment GSEA/Single-sample GSEA

GSEA was performed using the most current BROAD javaGSEA standalone version (http://www.broadinstitute.org/gsea/downloads.jsp) and gene signatures of the molecular signature database MsigDB[108,109], 7.4 (Hallmark, C2) as well as published signatures specifying different phenotypic states of melanoma such as "Melanoma aggressiveness"[110], "Proliferation", "Invasion"[58], parts of the IPRES signatures ("MAPKi-induced EMT"[111]) and MITF-target gene signature ("MITF_Targets_TCGA"[112]). The NGFR^high gene signature that defines anti-PD-1 therapy resistance was kindly provided by Oscar Krijgsman and Daniel S. Peeper[93] and overlaps with our identified set of NGFR-associated genes[32,33]. Gene signatures defining the undifferentiated neural-crest cell state were taken from Tsoi et al.[113]. Analyses of single signatures were run using 10,000 permutations; analyses of signature collections were run using 1000 permutations. Genes were ranked based on the Signal2Noise metric. Ecad- and NGFR-associated gene signatures were defined by the comparative analysis of Ecad^high or NGFR^high vs low MBM. Other signatures used in the study are summarized in Supplementary Data 10.

## Confocal microscopy

High-resolution immunofluorescence imaging of tumor sections and cell lines was performed with an LSM880 airyscan confocal microscope (Zeiss) and appropriate software (Zen black, ver. 2.3 SP1). Images were taken with objectives 10x, 20x and 63x/1.40 plan-apochromat, oil dic M27) at a resolution of 2048x2048 pixels/cm, 8 bit, scan speed 6, averaging 4. Imersol 518F was used for oil microscopy. Stacked multichannel image files (czi) were separated and background adjusted with Adobe Photoshop 2020 and stored as merged tiff files at a resolution of 600 dpi. Z-stacks were converted into three-dimensional

images using the arivis tool of ZEN2 software. The modifications to the image did not alter results in any way.

## 3D-invasion assays

Briefly, 50 μl of ice cold matrigel (Corning, 734−0270) were plated per well of a cooled 96-well plate and incubated for 10 min in a standard cell culture incubator at 37 °C. After matrigel polymerization, 2500 cells of BMCs were plated on top of the matrigel layer in 100 μl medium. Images were taken every 3 days for tracking of spheroid formation.

## Live cell imaging-based assays

**Migration assay.** The migratory capacity of unmodified or modified shRNA or reporter expressing BMCs was assessed using the Incucyte® Zoom live-cell imaging system. Briefly, $3 \times 10^4$ cells/well of each cell line were seeded on 96-well plates 24 h before, yielding a dense cell layer. Reproducible scratches were performed using the Incucyte® Wound-Maker tool (EssenBioscience/ Sartorius) and floating cells were removed by gentle washing of wells with medium. After wounding, the 96-well plate was placed into the live-cell imaging system and cell migration was monitored every 4 h for 7 days, using a 10× objective. Serial pictures were stacked for movie preparation using the ImageJ software (https://imagej.nih.gov/ij/). Statistical analysis was performed by using a two-tailed, paired $t$-test (simple migration assay) and unpaired $t$-test for assessing the migration of knockdown cells. To assess the migratory properties of BMCs, we determined the relative wound density that is determined as the percentage of spatial cell density in the wound area relative to the spatial cell density outside of the wound area at each time point[114]. This metric reflects a value which is normalized for changes in cell density caused by proliferation and is defined as:

$$RWD(\%) = \frac{100^*[w(t) - w(0)]}{[c(t) - w(0)]} \quad (2)$$

RWD = relative wound density; w(t) = Density of wound region at time, (t); c(t) = Density of cell region at time, (t).

**Proliferation assay.** Briefly, $2.5-5.0 \times 10^3$ cells/well of each cell line were seeded on 96-well plates and treated with dabrafenib or treatment control (DMSO) or left untreated, 24 h after plating. Following, the growth of cells was monitored for 72−96 h and images were recorded every 3 h. Cell growth over time was determined by changes in the cellular density as determined by a confluence mask and are given as "Phase Object Confluence (%)".

## Immunoblotting

Whole protein was isolated from frozen cell pellets using RIPA buffer and protein concentration of lysates was determined by Bradford assays (Pierce™ Coomassie Plus Assay Reagent, Thermo). 25−40 μg of total protein lysates were separated on 12% SDS-PAGE gels and transferred to PVDF membranes (Merck) by using the turbo semi-dry blotting system (BioRad). Membranes were blocked with 5% BSA solution and incubated with primary antibodies (NGFR, clone D4B3, rabbit; GAPDH, clone D16H11, rabbit (#5174) and β-Tubulin, clone 9F3 all from Cell Signaling Technology, Germany; all diluted 1:1000) overnight at 4 °C. For signal detection membranes were washed twice with PBS-Tween20 (0.1%) and incubated with a horse radish peroxidase (HRP)-coupled secondary antibody (goat anti-rabbit IgG, Cell signaling) for 1 h at RT and analyzed with an automated imaging system (Vilber).

## Animal experiments

All experiments with animals were performed in accordance with the German Animal Protection Law under the permission number G0130/20 obtained via the Berlin Ministry of Health and Social Affairs

(LaGeSo). ARRIVE 2.0 Guidelines were strictly followed. The maximal tumor size of 45 mm³ as permitted by the ethics committee was not exceeded in any of the experiments. Female CD-1 nude mice (8–9 weeks of age, 24–26 g, Charles River Laboratories) were stereotactically inoculated with $2.5 \times 10^4$ BMC1-M1, BMC1-M4 and BMC2 cells using a 1 µl Hamilton syringe and a stereotactic frame as described previously[115]. Briefly, the bur hole was placed 2 mm lateral (right) and 1 mm rostral from the bregma. The cells were administered at a depth of 3 mm. The number of cells used for the inoculation was determined in accordance with previous literature with the established human melanoma cell line M14[116]. For the procedure, the animals received anesthesia (9 mg Ketamine-Hydrochloride (CP-Pharma Handelsgesellschaft mbH, Burgdorf, Germany) +1 mg Xylazine (CP-Pharma Handelsgesellschaft mbH, Burgdorf, Germany) per 100 g) intraperitoneally as well as subcutaneous prophylaxis against infection (10'000 I.E, benzylpenicillin potassium, InfectoPharm Arzneimittel und Consilium GmbH, Heppenheim, Germany) and analgesia (100 mg/kg Paracetamol (B. Braun Deutschland GmbH & Co. KG, Melsungen, Germany), Lidocaine (Aspen Germany GmbH, Munich, Germany). Additionally, analgesia (300 mg/kg*d Paracetamol, bene-Arzneimittel GmbH, Munich, Germany) was administered via the drinking water for the first two postoperative days. Following the procedure, MRI scans were performed every 7 days until either the tumor volume was above 20 mm³ or at latest on the 49th day after implantation. Animals were sacrificed by perfusion with 4% PFA in deep anesthesia. The animals were kept in a 12 h light-dark cycle, at temperatures of 22 °C (+/−2 °C) with 55% (+/−10%) humidity, and had *ad libitum* access to water and food. Examinations for general and neurological symptoms were performed once a day and on the first two postoperative days twice a day. The weight was measured postoperatively once a day followed by regular weight monitoring once a week.

### Magnetic resonance imaging (MRI)
The MRI scans were performed using a 7 Tesla small animal MRI (BioSpec 70/20USR or PharmaScan 70/AS, Bruker Biospin, Ettlingen, Germany and ParaVision 6.0.1 or 5.1 software). During the scans, the mice received inhalation anesthesia (1.0–1.5% Isoflurane (CP-Pharma Handelsgesellschaft mbH, Burgdorf, Germany) in a mixture of 30% oxygen and 70% nitrous oxide). The depth of the anesthesia was monitored using the respiratory frequency (70–120 breaths per minute). T1 weighted sequences (TR = 1000 ms, TE = 10 ms, RARE factor = 2, 3 averages for BioSpec; TR = 975 ms, TE = 11.5 ms, RARE factor = 2, 4 averages for PharmaScan) after intraperitoneal administration of gadolinium-based contrast agent (12,09 mg per mouse in a solution with 180 µl 0,9% NaCl, Gadovist, Bayer AG, Leverkusen) and T2 (TR = 4200 ms, TE = 36 ms, RARE factor = 8, 3 averages for Biospec and TR = 4200 ms, TE = 36 ms, RARE factor = 8, 4 averages for PharmaScan) were measured. The tumor volume was measured using ITK-SNAP 3.8.0 Software[117] (Paul A. Yushevich, Guido Gerig, www.itksnap.org).

### Quantitative real-time PCR
RNA isolation from frozen cell pellets was performed with the RNeasy Mini Kit (Qiagen, Germany), following the manufacturers protocol. Reverse transcription of 500 ng–2.5 µg RNA was performed with SuperScript VILO cDNA synthesis kit (Invitrogen, Germany) and diluted to a final volume of 50 µl. qRT-PCR was carried out on a Step one plus PCR cycler (Applied Biosystems, Germany) for 30–40 cycles. Primers were designed for 55–60 °C annealing temperatures. Relative expression levels were calculated with the ΔΔCT method[118], normalized to β-actin. Primer sequences are shown in Supplementary Data 9.

### Drug sensitivity assays
Drug treatments were performed 24 h after seeding of 2500–5000 cells/96-well in 100 µl medium. The response of BMCs and conventional melanoma cell lines to dabrafenib in a range of 1nM-10µM of

eight technical replicates was determined by live cell imaging. Images were taken every three hours using a 10× objective and the general label-free mode, two pictures of eight technical replicates per condition were taken. Drug response was assessed by changes in the cellular density over time. The cell density was determined by a confluence mask tool as part of the IncucyteS3 software. IC50 values were calculated by curve-fitting (https://search.r-project.org/CRAN/refmans/REAT/html/curvefit.html) based on confluence measurements at day 3.

### Statistics & reproducibility
No statistical methods were used to predetermine sample sizes. We included all individuals with MBM where sufficient material was available as specified in the description of study design. No data were excluded from the analyses. Statistical details for each analysis are mentioned in each figure legend or in the respective part of the text. RNA-sequencing, quantitative real-time PCR and immunohistochemically staining were performed in a blinded fashion. Histological diagnosis of melanoma samples was performed by at least two consultants of (neuro)pathology with agreement. Histological stainings were replicated at least once with the appropriate positive and negative controls. Each replication was successful. Immunohistochemistry/-fluorescent analyses were technically replicated at least once. Each replication was successful. SNuPE-based validation of methylation of a CpG island within the ITGB7 gene was performed in $n = 14$ MBM patients. All cell-based in vitro experiments were performed in six to eight technical and three biological replicates. Each replication was successful. RT-qPCR was performed in two to three technical and three biological replicates. Each replication was successful. The reliability of TargetSeq was demonstrated by comparative analyses of concordant sets of MBM and/or associated sets of MBM and BMCs. In addition, overlapping analyses of two independent amplicon-panels and prior results from routine diagnostic (at least BRAF status) successfully confirmed the reliability of DNA sequencing. Knockdown of NGFR was additionally validated by immunoblotting which was replicated in five cell lines. Each replication was successful. The representative images shown were adjusted in brightness and contrast to different degrees (depending on the need resulting from the range of brightness and contrast of the raw images) in Adobe Photoshop 2020; Version: v.21.1.0.106x64; however the modifications to the image did not alter results in any way. The experiments were not randomized. All images were created by the authors none of them were adapted from previous works. All schemes provided in Figs. 1a, 2a, 3a, 4a, 5a, 6a, e, 7b, 9 and Supplementary Fig. 1a were designed by the authors. However, parts of the figures were drawn by using pictures from Servier Medical Art. Servier Medical Art by Servier is licensed under a Creative Commons Attribution 3.0 Unported License (https://creativecommons.org/licenses/by/3.0/). This information was taken from https://smart.servier.com/how-to-cite-servier-medical-art/.

### Reporting summary
Further information on research design is available in the Nature Portfolio Reporting Summary linked to this article.

## Data availability
Whole transcriptome and methylome data were deposited in the European Genome-Phenome Archive (EGA), under accession numbers EGAS00001005976 and EGAS00001005975. The data are available under controlled access due to patient consent. Access can be obtained by contacting the appropriate Data Access Committee listed in the study. Access will be granted to commercial and non-commercial parties according to patient consent forms and data transfer agreements. A response to requests for data access can be expected within 14 days. After access has been granted, the data is available for two years. Supplementary tables have been deposited at Zenodo (https://zenodo.org/record/7013097 and https://doi.org/10.

5281/zenodo.7249214). Kaplan-Meier survival data were derived from study EGAS00001003672 shown in Fig. 8e and Supplementary Fig. 4b are provided in the Source Data file with this paper. Raw image files of histological stainings, immunofluorescence, and immunohistochemistry shown in the Figures have been deposited publicly at Zenodo (https://doi.org/10.5281/zenodo.7249214). Targeted-sequencing (TargetSeq) results (SNVs) are given in Supplementary Data 6. SYBR Green quantitative real-time PCR (qPCR) results of SOX4, NGFR, AXL, MET, HERC5, TCF19 and FOSL1 are given in Fig. 7c (right). Raw real-time PCR data (original xls file containing CT/RQ values) are given in the Source Data file with this paper. Live cell imaging-based raw measurement files of drug response and proliferation assays are given in the Source Data file with this paper. The remaining data are available within the article, Supplementary Information or Source Data file. Source data are provided with this paper.

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

## Acknowledgements

This research was supported using resources of the VetCore Facility (VetImaging | VetBioBank) of the University of Veterinary Medicine Vienna, Austria. We gratefully thank Mascha Osang for careful proof-reading of the manuscript. We are indebted to Petra Matylewski for excellent technical assistance. We thank Dr. Gilles Gasparoni for excellent technical assistance conducting the EPIC arrays. J.R. is a participant in the BIH-Charité Clinical Scientist Program funded by the Charité – Universitätsmedizin Berlin and the Berlin Institute of Health. We thank the German Cancer Consortium (DKTK), Partner site Berlin for technical support. A.V. and T.R. received funding by the Deutsche Forschungsgemeinschaft DFG (Grant number: 392534127). A.V. is funded by the Doctoral Program of the Austrian Academy of Sciences (Grant number: DOC/26523).

## Author contributions

J.R. performed experiments and wrote the manuscript; E.S. performed experiments and wrote the manuscript; R.K. performed experiments; A.V. performed experiments, T.R. performed experiments and wrote the manuscript; J.O. performed craniotomy and provided melanoma brain metastases (MBM); G.A. performed intracranial injections and MRI; B.B. performed intracranial injections and MRI; C.S. provided murine models; S.T. performed methylation profiling; M.M. performed DNA sequencing; P.V. performed craniotomy and provided melanoma brain metastases; S.H. provided immunohistological expertise; P.K. performed immunophenotyping; D.W. provided BMC53 cells, sections and expertise; F.W. provided expertise; F.H. provided archived tumor samples and infrastructure; S.K.R. provided funding and infrastructure; F.G. provided funding and infrastructure; K.J. performed initial data analyses and provided expertise; T.R. performed data analysis. All authors have read the manuscript, provided intellectual input and agreed to the published version of the manuscript.

## Funding

Open Access funding for this article was provided by the University of Veterinary Medicine Vienna (Vetmeduni Vienna).

## Competing interests

The authors declare no competing interests.

## Additional information

[1]Department of Pathology, University Medicine Greifswald, Greifswald, Germany. [2]Berlin Institute of Health (BIH), Berlin, Germany. [3]German Cancer Consortium (DKTK), Partner Site Berlin, CCCC (Campus Mitte), Berlin, Germany. [4]Department of Neuropathology, Charité-Universitätsmedizin Berlin, corporate Member of Freie Universität Berlin and Humboldt-Universität zu Berlin, Berlin, Germany. [5]Department of Neurosurgery, Charité-Universitätsmedizin Berlin, corporate member of Freie Universität Berlin, Humboldt-Universität zu Berlin and Berlin Institute of Health, Berlin, Germany. [6]Charité CyberKnife Center, Department of Radiation Oncology, Charité Universitätsmedizin Berlin, corporate member of Freie Universität Berlin, Humboldt-Universität zu Berlin and Berlin Institute of Health, Berlin, Germany. [7]Department of Genetics/Epigenetics, Faculty NT, Saarland University, Saarbrücken, Germany. [8]Department of Pathology, Charité-Universitätsmedizin Berlin, corporate member of Freie Universität Berlin, Humboldt-Universität zu Berlin and Berlin Institute of Health, Berlin, Germany. [9]Institute for Medical Biochemistry, University of Veterinary Medicine Vienna, Vienna, Austria. [10]Institute of Pathology, Unit of Laboratory Animal Pathology, University of Veterinary Medicine Vienna, Vienna, Austria. [11]Department of Dermatology, University Hospital Carl Gustav Carus at TU Dresden, Dresden, Germany. [12]National Center for Tumor Diseases (NCT), Dresden, Germany. [13]Helmholtz-Zentrum Dresden-Rossendorf (HZDR), Dresden, Germany. [14]German Cancer Research Center (DKFZ), Faculty of Medicine, Heidelberg, Germany. [15]Skin Cancer Center at the University Cancer Center Dresden, University Hospital Carl Gustav Carus at TU Dresden, Dresden, Germany. [16]Nutrigenomics Unit, Institute of Animal Nutrition and Functional Plant Compounds, University of Veterinary Medicine Vienna, Vienna, Austria. [17]Berlin Institute of Health at Charité – Universitätsmedizin Berlin, Center for Regenerative Therapies (BCRT), Berlin, Germany. [18]These authors contributed equally: Josefine Radke, Elisa Schumann. ✉e-mail: josefine.radke@med.uni-greifswald.de; torben.redmer@vetmeduni.ac.at

