## [Peer Review File · Nature Communications]

Decoding molecular programs in melanoma brain metastasesEditorial Note: Parts of this Peer Review File have been redacted as indicated to maintain the confidentiality of unpublished data.

REVIEWER COMMENTS

Reviewer #1 (Remarks to the Author): Expert in brain metastasis

this is an excellently-written and very thoughtful paper addressing selectd aspects of the biology of brain metastases from melanoma that might explain a somewhat bipolar behavior of aggressiveness and resistance or sensitivity to currently-used systemic therapies.

this reviewer is a clinical investigator working only indirectly with basic scientists in the field and thus not well-equipped to critique the techniques, hypotheses, results or their interpretation.

a few minor comments to the authors follow:

while most if not all abbreviations are spelled out the first time used, some are repeated and others are used often in the manuscript, which is so long and complicated that it becomes hard to remember the abbreviation by the time one gets to the last 1/2 of the paper. maybe those abbreviations could at least be repeated in the figure legends.

the terms chosen to denote the single most important molecule in the paper, which is the NGFR, are heterogeneous, and unless it is necessary to use different terms in different places, it would be preferable to see the same term over and over--either CD271 or NGFR or NGFR/p75 or whatever is most correct and pertinent to this set of experiments.

would it be feasible to test these melanoma brain metastasis clones for their sensitivity as targets for immune effectors such as CD8 cells from patients treated with immune checkpoint blockade (or a more artificial system such as maximally stimulated effectors against these brain metastasis clones?)

Reviewer #2 (Remarks to the Author): Expert in melanoma cell biology

The authors here perform a comprehensive characterization of melanoma brain metastases using transcriptomic and methylomic approaches. The authors propose key roles for Ecad, NGFR and TIL status for stratifying tumors into the dogmatic proliferative or invasive categories. The work is critically important and well timed. The study is weakened by the relatively small number of patients samples used in certain figures, which should be balanced with the reality that attaining these rare samples is difficult. Nonetheless, concerns of how robust these findings may be would be addressed with additional data from mouse models of brain metastases. Below are recommendations to increase the robustness of this study:

- Line 471 (Figure 1B right panel), please show lines that demonstrate the pre/post pairs within that dataset. Also confirm that there were only 6 total pairs in that dataset.
- “The latter synchronous metastases of patient 23 that developed within 493 the occipital and parietal lobes or MBM of patient 24 that emerged within the frontal lobe and lobus insularis, like metastases of Pat8 (M2-M4) featured a therapy-resistant phenotype (Fig. 2a).”
 - o Not clear how Figure 2A shows evidence that justifies this claim above
- The authors show evidence in Figure 1D that NGFRhi cells invade to make proximal metastases near the primary. In Figure 2 the authors are now differentiating between NGFRhi cells and invasive cells. It is unclear whether the authors believe NGFRhi cells are invasive or not.
- Figure 2B needs to be better visualized, it is taken quite a bit of effort for me to understand what I am looking at, while going back and forth between the color key and the data.
 - o Designating a few key genes in the data will help the reader appreciate the significance of any differences observed
- Figure 3 is based entirely on a single patient. Clearly the material is difficult to detect, but it makes assessing how robust these findings are very difficult. This work would be great enhanced by assessing melanoma brain metastases in mice for the same markers to demonstrate how robust

these findings are.

- Figure 4C needs to be better visualized, with a subset of genes labeled in the heat map for the reader to appreciate the data. It is very difficult to distill meaning from the way these data are currently organized
 - o What is meant by BRAF status? What specific BRAF mutations would be defined as BRAF mutant? Any? Just V600E/K?
- It seems much of key results are put in the supplemental and much anecdotal data from 1-3 patients is included in the main figures. The authors should find a way to incorporate more of the Supplemental into the formal figures.
- Figure 5E: It is unclear whether the authors injected mice with BMC1-M1 and BMC1-M4 cell lines? And which color is which cell line?
 - o What were the growth dynamics and viability of both these cell lines? Why not use a spontaneous metastasis approach rather than injecting right into the brain?
- Figure 5 D/E: Why not knockdown NGFR and/or manipulate Ecad to prove whether they are playing roles in brain metastases as most of the manuscript proposes? I am not convinced that showing a relative difference in brain met formation between these 2 cell lines (M4 versus M1) strengthens the central hypothesis of this manuscript.
- The title of Figure 5 concerns “distinct migratory phenotypes” whereas the data focuses on a different cell surface marker profile and ability to grow in the brain when cells are injected directly into the brain. The evidence does not support the title of Figure 5
- Figure 8B should show WBs or RT-PCR proof of knockdown of NGFR in the formal paper figures.
- The authors should show the consequence of manipulating Ecad expression and its consequence on invasion/migration/proliferation since Ecad is mentioned in the abstract as a key part of this story.
- The authors performed methylome profiling but it does not seem that any of their findings were validated using the cell lines, in terms of having a consequence on cellular proliferation/invasion/migration. Validating an aspect of the methylome analysis would greatly strengthen this story

Reviewer #3 (Remarks to the Author): Expert in RNA-seq

In this paper, Radke et al. describe the molecular characteristics of melanoma brain metastases. First they analyzed the transcriptome, methylome and mutational profile of a heterogeneous mixtures of intracranial metastases (pre or post therapy). They identified different subpopulations that can be clustered based on their Ecad and CD271 expression. They validated these findings using immunohistochemistry. They relied on pre-defined genesets to determine the 'invasive' subpopulation. They then complemented this work with in-vitro work to validate the causative role of CD271 loss (and SOX4) in inducing migration. While the biology is not completely novel its demonstration in melanoma brain metastases is.

Strengths of this study include:

Leveraging very valuable clinical specimens (fresh melanoma brain metastases)

Methods are described in details.

Complementing the genetic and immunohistochemical analysis with functional validation

Concerns.

While the authors leverage valuable clinical specimens, and the overall number of specimens appears adequate, some of the analysis focuses on samples obtained from one patient. While these provide useful information to trace the clones that have re-emerged after therapy, it would have been much stronger if they had multiple cases pre and post treatment. I realize that this may not be feasible.

The authors should be careful not to draw pre-mature conclusions. They rely on predefined genesets, some of which were derived from different tumors (i.e. to define invasiveness) to assess tumor's invasiveness. While in general these genesets can serve as a correlate to increased 'invasiveness'

of the tumor, the authors should not establish causation solely based on GSEA association. For example the title of figure 1 . 'MAPKi treatment increases invasiveness of melanoma' is definitely not supported by the data and can be misleading. This also applies to their interpretation throughout the results section.

The authors derive cell lines from tumors and perform transcriptome analysis along with established cell lines. The authors claim that these cell lines that they derived better recapitulate the in vivo tumor. Given that these cell lines were analyzed at low passages (1-4) compared to the established cell lines, this is to be expected. I am not sure what is the point of this comparison.

What is more relevant is that they should compare the transcriptome of these derived cell lines (and tumors) to cells with CD271 knockdown.

The authors should list all the genesets that they have used (including the referenced ones) in a separate supplementary document.

While the authors described the technical details, the paper may benefit from a 'big picture' perspective in the introduction. For example, the authors do not explain the rationale of using CD271 as a marker of aggressive melanoma (Figure 1). The authors should also reference relevant papers in the introduction. The authors should also reference and discuss the work by (Restivo, Nature Communications) in the introduction given that this paper was among the first to discuss the role of CD271 in inducing a phenotypic switch in melanoma.

Also, Reference 84 is the correction and not the original publication.

The wound scratch assay lasts for 128 hours. This is a long time. The authors should also demonstrate that there is no increased proliferation. Alternatively, they can use mitomycin to inhibit cell proliferation and assess migration only.

Sequencing was done using Illumina Platform and IonProton (Thermo). Were some tumors analyzed by one modality, or were all analyzed by both platforms. Can the authors ensure adequate and comparable coverage among all tumors? This is important since they are using this information to infer the persistence of specific subclones. Figure 6.

A schematic diagram at the end summarizing the main findings could be helpful.

Reviewer #4 (Remarks to the Author): Computational expert

In this manuscript, Radke et al investigate through transcriptome and methylome profiling a set of melanoma brain metastasis in order to identify molecular pathways, subgroups and potential therapeutic possibilities. I can appreciate that the authors have done an impressive amount of work, and I believe the experiments are in general carefully done and the figure are informative and aesthetically pleasant. However, I found the writing difficult to follow, as I think there are too much data that (in my humble opinion) perhaps confuse more than clarify their message.

Major points

1. Though I can appreciate the incredible amount of work that went into this manuscript, I am unsure of what the main message is, and I struggled several times to remember why certain experiments were being performed. Is the main message that melanoma brain metastases (MBMs) can be subdivided in invasive (NGFR+) and proliferative (Ecad+) and that these may be targeted by repurposed drugs? Is it the proposed switch from Ecad+ to NGFR+ as the metastatic process evolves? Is it that the genomic profile and other molecular characteristics are maintained in vitro when

cultivating these cells? I would recommend simplifying the manuscript and perhaps focus on one main, clear message.

2. Can it be stated clearly if the authors think that MBMs recapitulate what has been described for cutaneous MRD (For example, in the paper by Rambow et al, Cell 2018)? Do the authors anticipate that their proposed method of targeting MBMs would work in vivo?

3. In my opinion, some assumptions are perhaps not completely backed up by their data. For example:

a. Line 462-463: How does this level of staining compare to non-treated or non-resistant MBM? Is there a baseline with which to compare to back up the claim that CD271 expression in these metastases is high? Were the tumours in Figure 1B from brain metastases too? Also, later in that sentence authors say that this expression may be "promoted by reactive astrocyte secreted factors" but what is their evidence to say this? Did they stain for such factors?

b. Line 501 - "...identified a pan-gene signature potentially defining MBM showing enrichment of extracellular exosomes and genes that are associated with focal adhesion or defining extracellular matrix components (ECM) or melanosomes". How was this pan-gene signature defined? Are the authors suggesting that melanocytes within those BM are expressing these genes but not the surrounding cells? What is this assumption based on? The authors compare against a set of controls (supp figure 2) but how do they know that normal brain tissue is comparable to non-melanocyte cells within a tumour environment?

4. A line of logic I found confusing but I think is crucial for the message the authors want to deliver is in line 593: "Likely, brain metastatic tumor cells exhibit a rapid EMT-MET capacity and metastases re-acquire Ecad expression soon after every metastatic step." - As I understand, the authors have analyzed so far the gene expression of tumours before and after RAFi/MEKi therapy (Figures 1,3) and see an association of CD271 expression with more invasive characteristics. They suggest that an Ecad-to-CD271 switch marks a final step of MBM progression (line 560) but then they analyze more metastases (from other datasets including TCGA) but do not see much difference in expression. I think the assumption that metastases re-acquire Ecad after every metastatic step (what do they define as a metastatic step?) requires more backing. For example, was the TCGA data cleaned to keep melanocytes only (as the authors did previously with their dataset)? The methods don't seem to list how the authors analysed these data, but apologies if I have missed it.

Minor

There were several instances of sentences I found confusing and that I think it would be good to review:

*Introduction

Line 102 - "The time from initial diagnosis of primary tumors to the detection of BM ranges from 1-10 years, supporting the assumption of a slow evolutionary process of BM in 20 – 40% of melanoma patients". - Is this adjusted by stage at diagnosis? I wonder if the patients with detected BM 1 year after diagnosis are at a more advanced stage than those with BM detected 10 years after diagnosis?

Please clarify what MBM refers to (where it is first mentioned it's unclear what it is abbreviating). Related to this, I find the beginning of line 108 confusing - seems to imply extracranial metastases are melanoma brain metastases? Please clarify this point.

Line 118 - "responsible for a non-genomic cellular heterogeneity" - should this sentence drop the "a"?

Line 131 - "non-genetic emergence of minimal-residual disease" - what does this refer to? Do the authors want to say "non-genetic mechanisms of resistance in MRD" or something similar?

Line 132 - "Recent work has shown that melanoma cells secreted extracellular vesicles were enriched with CD271 that was taken up by lymphatic endothelial cells, therefore aiding lymph node

metastasis". - This reads confusing to me, would it be better to say "Extracellular vesicles secreted by melanoma cells were enriched with CD271...".

*Methods

Line 157 - Should this be "cell lines or iii) isolation of RNA and DNA or IV) formaldehyde-fixed and paraffin embedded (FFPE) and archived.? (Is "IV" missing?)

Line 200 - Should this be "and edited WITH Adobe Photoshop"? Also, please specify that the modifications to the image do not alter results in any way.

Line 210 - Please add antibody ID and if possible lot number.

Line 248 - "Isolation of total RNA from snap frozen or intraoperative tumors" - This line confuses me. Above it says that intraoperative tumours were snap-frozen among a number of possibilities. Does this mean "snap frozen or fresh..".?

Line 250 - Please split the sentence starting in this line "The library preparation...."

Line 282 - "Data of the [1000 genomes project] enabled the separation of single nucleotide polymorphisms (SNPs) from mutations (single nucleotide variants, SNVs)." - Was this the only database used? I would argue it is better to use the GnomAD sets as they contain many more individuals.

Line 385 - Can the number of cells used for the inoculation be specified? Were all mice inoculated with the same number of cells?

*Results

Line 488 - "environmental cells" - do the authors mean cells in the extracellular microenvironment?

Line 490 "we collected intraoperative and cryo-preserved MBM (n=16; Supplementary Table 1) from different intracranial sites" - Supplementary Table 1 lists more than 16 MBM, with different patient IDs, can it be clarified please which samples were used, or what these 16 are? There seem to be inconsistencies with Figure 2a (This figure lists patient 6 but the table does not?)

Line 492 - This sentence is hard to understand ("the latter synchronous metastases of patient 23 that.."), please clarify.

Line 512 - perhaps a good paper to cite here is Rabbie et al (2020), who have done an analysis of the mutational status of driver genes within melanoma brain metastasis. Following from that - do the authors not see KRAS mutations in these MBM?

Line 515 - Can the authors please specify which tumours underwent methylome profiling? I think it is important to ensure they all have the same origin (i.e. snap frozen or FFPE) or alternatively that this has no effect on the results described.

Line 552 - "suggesting that the emergence of CD271-driven subclones..." - how do the authors know that CD271 drives these sub clones and is not only a biomarker?

Line 618 - Should the "though" be changed to "however" so the sentence is completed?

Line 749 - Should "public" mutations be "established", or "known"?

*Figures

Supp Figure 3B has several gene names that say "NA", please correct this (if the CpG is not associated with a gene, can the location be marked?) Also, this figure is referenced in the statement

"The matching with transcriptome data revealed 14 MBM expressed genes that featured high methylation of promoters in BRAFwt tumors and identified integrin b7 (ITGB7) as a potential predictor of favorable survival". Can the authors please specify what these 14 genes are?

Figure 3a - Do the vertical lines crossing the timeline represent all tumours removed from P8, but only M1-4 were used in the study? What does BMC stand for?

Figure 3D - Can the label DECMA1 be changed for Ecad? (As in the left panel)

Supp Figure 5A. There doesn't seem to be much difference in E-cadherin expression among the TCGA tumours, the p value is probably driven by a few outliers.

Point-by-point responses to the reviewers' comments:

We have used different font colours to differentiate between the reviewers' comments (**black font colour**), our responses (**blue font colour**), and revised manuscript passages (**orange colour**). We uploaded a manuscript version, where all changes are highlighted in yellow.

1. Reviewer #1 (Remarks to the Author): Expert in brain metastasis

This is an excellently-written and very thoughtful paper addressing selected aspects of the biology of brain metastases from melanoma that might explain a somewhat bipolar behavior of aggressiveness and resistance or sensitivity to currently-used systemic therapies.

This reviewer is a clinical investigator working only indirectly with basic scientists in the field and thus not well-equipped to critique the techniques, hypotheses, results or their interpretation.

A few minor comments to the authors follow:

C 1.1. while most if not all abbreviations are spelled out the first time used, some are repeated and others are used often in the manuscript, which is so long and complicated that it becomes hard to remember the abbreviation by the time one gets to the last 1/2 of the paper. maybe those abbreviations could at least be repeated in the figure legends.

Response: We thank reviewer #1 for his/her overall positive evaluation of our study as well as for the valuable comments to improve our manuscript. As suggested by the reviewer, we have repeated important abbreviations (such as MBM and BMC etc.) in the figures legends to make this clearer.

C 1.2. the terms chosen to denote the single most important molecule in the paper, which is the NGFR, are heterogeneous, and unless it is necessary to use different terms in different places, it would be preferable to see the same term over and over--either CD271 or NGFR or NGFR/p75 or whatever is most correct and pertinent to this set of experiments.

Response: We fully agree with the reviewer that this might be confusing for the reader. Following the HUGO Gene Nomenclature Committee, we are now consistently using "NGFR".

C 1.3. would it be feasible to test these melanoma brain metastasis clones for their sensitivity as targets for immune effectors such as CD8 cells from patients treated with immune checkpoint blockade (or a more artificial system such as maximally stimulated effectors against these brain metastasis clones?)

Response: We thank the reviewer for this interesting idea. The testing of our cell lines or orthotopic models for sensitivity to CD8+ T cells needs to be established. The *in vitro* testing would need the establishment of a co-culture system and *in vivo* testing would need a humanized mouse model or T cell transfer model, which is out of our expertise. However, such a project would be an important follow-up research project.

2. Reviewer #2 (Remarks to the Author): Expert in melanoma cell biology

The authors here perform a comprehensive characterization of melanoma brain metastases using transcriptomic and methylomic approaches. The authors propose key roles for Ecad, NGFR and TIL status for stratifying tumors into the dogmatic proliferative or invasive categories. The work is critically important and well timed. The study is weakened by the relatively small number of patients samples used in certain figures, which should be balanced with the reality that attaining these rare samples is difficult. Nonetheless, concerns of how robust these findings may be would be addressed with additional data from mouse models of brain metastases. Below are recommendations to increase the robustness of this study:

Response: We thank reviewer #2 for his/her remarks and valuable comments that have helped to substantially improve our manuscript. Basically, we restructured large parts of the manuscript particularly the results and figures 2-8 and discussion sections underwent considerable changes. Figures 8 and 9 were newly added.

C 2.1. Line 471 (Figure 1B right panel), please show lines that demonstrate the pre/post pairs within that dataset. Also confirm that there were only 6 total pairs in that dataset.

Response: We agree and changed Figure 1B accordingly. Yes, indeed this data set comprises matched pairs of pre- and post-BRAFi/MEKi melanoma from only six patients. TCGA provides additional patients. However, this data set does not provide transcriptome data.

C 2.2. “The latter synchronous metastases of patient 23 that developed within 493 the occipital and parietal lobes or MBM of patient 24 that emerged within the frontal lobe and lobus insularis, like metastases of Pat8 (M2-M4) featured a therapy-resistant phenotype (Fig. 2a).”

Not clear how Figure 2A shows evidence that justifies this claim above

Response: We agree and removed this statement. Moreover, we removed the statement about the comparison of Pat23 and Pat24 tumors.

C 2.3. The authors show evidence in Figure 1D that NGFRhi cells invade to make proximal metastases near the primary. In Figure 2 the authors are now differentiating between NGFRhi cells and invasive cells. It is unclear whether the authors believe NGFRhi cells are invasive or not.

Response: The results of our previous studies as well as studies by Boiko et al. (DOI: 10.1038/nature09161)¹ and older studies by Dario Marchetti (DOI: 10.3892/ijo.7.1.87)² strongly suggest that NGFR^{high} cells feature an invasive phenotype. This is now better underpinned by additional data we added in Figs. 1c, 3a-c and 4c supporting this statement. We indicated expression of invasion-related genes such as Lysyl oxidase-like 2 (LOXL2) or LOXL3 and observed that both are related to NGFR expression. In addition, our present study and previous migration studies (Radke et al., 2017) suggest that the expression of NGFR modulates the migratory phenotype of conventional melanoma cells such as A375 and BMCs. Marchetti et al. demonstrated that brain metastases express NGFR and environmental cell such as astrocytes provide ligands such as NGF or BDNF. However, whether this is a preferred mechanism of MBM invasion remains to be investigated.

C 2.4. Figure 2B needs to be better visualized, it is taken quite a bit of effort for me to understand what I am looking at, while going back and forth between the color key and the data.

Designating a few key genes in the data will help the reader appreciate the significance of any differences observed

Response: To make this easier to understand, we removed the “Classifiers” and added a more accessible annotation. The heat map in Fig. 2b shows unsupervised clustering of top1000 variant genes among all MBM compared with brain controls and demonstrates that MBM rather cluster by their molecular subgroup instead of their intracranial location. Likely, this is also true for patient-matched MBM such as Pat23 tumors. However, to investigate whether MBM might feature molecular programs that are defined by spatial-dependent cues, one would need at least three concordant MBM per region. Possibly, MBM show a clustering regarding their location in the absence of interpatient-heterogeneity. However, this needs to be investigated.

We added genes specifying astrocytes (GJB6), microglia (TMEM119) and neurons (ROBO2) and indicated expression of Ecad and NGFR. Remarkably, even this initial clustering reveals a clear separation of MBM.

C 2.5. Figure 3 is based entirely on a single patient. Clearly the material is difficult to detect, but it makes assessing how robust these findings are very difficult. This work would be great enhanced by assessing melanoma brain metastases in mice for the same markers to demonstrate how robust these findings are.

Response: We fully agree and included data from the pre-/post-treatment melanoma data set, demonstrating a switch of not only Ecad into NGFR⁺ phenotypes but also showing a change in expression of Ecad- and NGFR-associated signature genes, suggesting a switch of molecular programs and not of only two genes, shown in Figure 4c, left panel. In addition, we performed a deconvolution of gene signatures specifying invasive, stem-like, Ecad- or NGFR-driven programs and observed a clear separation into low invasive/pigmented/Ecad⁺ and stem-like/invasive/NGFR⁺ subtypes shown in Figure 4c, center panel and Reviewer-only Figure 1. In a third analysis we assessed whether the NGFR⁺ phenotype of MBM was associated with presence of minimal-residual disease as previously suggested by Rambow et al., in melanoma. Indeed, we observed that NGFR⁺ MBM featured expression of MRD-associated genes, at least partly; shown in Figure 4c, right panel. Very recently, Biermann et al., provided insights into drug-naïve melanoma brain metastases, however mentioned that AXL-associated genes were not higher expressed than in extracranial metastases and concluded that “the AXL-high program is not a defining feature of MBM”³. However, we observed AXL-signature genes indeed enriched in NGFR^{high} MBM that are not primarily found among drug-naïve MBM. As AXL-expression in melanoma was associated with increased invasiveness, we strongly suggest that 1.) Molecular programs of AXL and NGFR overlap; AXL and NGFR might act in concert for controlling invasive properties. 2.) We observed that AXL and Ecad like NGFR show a mutually exclusive expression pattern whereas MITF-signature genes were found overlapping with Ecad-signature genes (identified in this study). Nevertheless, we observed intermediate tumor cell states showing expression of both Ecad and NGFR as well as Ecad and AXL.

[REDACTED]

[FIGURE REDACTED]

C 2.6. Figure 4C needs to be better visualized, with a subset of genes labeled in the heat map for the reader to appreciate the data. It is very difficult to distill meaning from the way these data are currently organized

Response: As suggested by the reviewer, we added some representative genes of the Ecad- and NGFR signatures to the figure to make this easier to interpret. Figure 4C was revised and moved and is now Fig 3A.

C 2.7. What is meant by BRAF status? What specific BRAF mutations would be defined as BRAF mutant? Any? Just V600E/K?

Response: To be more precise, we now added information about the type and variant frequencies of all BRAF and NRAS mutations detected in the MBMs that underwent transcriptome or methylome profiling or TargetSeq to Figure 2A. Additional information on the BRAF/NRAS mutation state is collected in Supplementary table 1.

C 2.8. It seems much of key results are put in the supplemental and much anecdotal data from 1-3 patients is included in the main figures. The authors should find a way to incorporate more of the Supplemental into the formal figures.

Response: We agree and re-structured the main and supplementary figures.

C 2.9. Figure 5E: It is unclear whether the authors injected mice with BMC1-M1 and BMC1-M4 cell lines? And which color is which cell line?

Response: We apologize that this was unclear. We revised the Figure accordingly. As the figures were reorganized, Fig. 5E is now displayed with regard to the analysis of progressive phenotypes in Fig. 6E. In addition to the initial two cell lines, we injected a third cell line, representing another patient (Pat 35, Supplementary table 1). These BMC2 cells harbored a BRAF^{N581Y} and NRAS^{G12C} mutation.

C 2.10. What were the growth dynamics and viability of both these cell lines? Why not use a spontaneous metastasis approach rather than injecting right into the brain?

Response:

a.) Why not use a spontaneous metastasis approach rather than injecting right into the brain

We understand the reviewer's comment concerning our metastasis approach. However, current spontaneous melanoma models rely either on overexpression of the RET-tyrosine kinase receptor or are based on B16-F10 cells. Although both model systems might provide insight into the mechanisms of metastasis and potentially brain metastasis, the basic questions remain unsolved: what factors drive brain metastasis. RET was neither expressed nor mutated in MBM and even the comparison of gene signatures from brain metastatic A375Br cells with signatures gathered from expression profiling of MBM did not overlapped.

In the present manuscript we mostly focused on intracranial metastasis and are not claiming that Ecad or NGFR expression alone is sufficient for promoting brain metastasis. This is indeed unlikely. We gained insights from the latest reports by Rabbie et al. and Biermann et al., both investigated drug-naïve melanoma brain metastases (MBM) and both observed expression of

melanocyte/pigmentation-related genes and at least Biermann et al. observed E-cadherin (Ecad, CDH1) expressed in their tumors, although they used different methods for transcriptome profiling of these tumors (bulk vs. scRNAseq) and independent tumor cohorts.

We compared the top-expressed genes as provided by publications and although both signatures (top50 genes Rabbie et al., and top100 genes Biermann et al.) did not overlap with each other, both signatures overlapped with the Ecad^{high} signature. This suggests that Ecad^{high} MBM indeed may represent a set of either drug-naïve/drug-responsive or less treated tumors. To resolve whether NGFR or Ecad might indeed characterize tumors that have developed by a certain therapeutic intervention one would need a set of tumors that have been solely treated with or without BRAFi/MEKi or ICi or radiation therapy. However, access to these samples is almost impossible, considering the already small number of accessible brain metastases.

Although tail vein injection of melanoma cells/brain metastasis-derived cells (BMCs) might provide insight into their in vivo metastatic capacity we are worried about such an assay for several reasons. 1.) If not using humanized mouse models, the standard assay for a systemic distribution of melanoma cells requires immune compromised mice. As the role of the immune system is highly important because cellular subclones might be selected by the interaction with immune cells, this system might not adequately reflect the processes in the human patient. 2.) Even in the spontaneous RET-melanoma model, the development of metastases is observed in a range of 20-70 days after injection and mice likely suffer from a plethora of symptoms stemming from metastases in other organs but not from brain metastases. Hence the spatiotemporal development of MBM is not reflected by such a mouse model. 3.) Although recent profiling studies of human and mouse astrocytes and microglia have shown differences not only in the heterogeneity of cellular subsets but also in the transcriptome, the intracranial injection of tumor cells can provide insights into early steps of intracranial tumor development and metastasis that are likely triggered by the interaction of tumor cells and astrocytes/microglia. Although this assay also leaves the identification of molecular mechanisms driving metastasis to the brain unresolved as well. In human brain metastases we observed a strong activation of astrocytes. Such reactive astrocytes show a different morphology than non-reactive astrocytes, however whether and to which extent astrocytes and microglia are activated after cells resided within the brain for only ~30 days is currently outside the frame of this manuscript. Considering the high number of micrometastases that are observed in melanoma patients post mortem and the time from initial diagnosis of a primary tumor to detection of brain metastases, it's very likely that only a small number of tumor cells that entered the brain indeed develop symptomatic brain metastases. The microenvironment/metastatic niche probably controls this process.

b.) What were the growth dynamics and viability of both these cell lines

We performed live-cell imaging based proliferation assays of our BMCs and indeed observed distinct proliferative properties. BMC1-M1 and BMC3 exhibited lowest doubling times (80.0 ± 25.2 h) and >96 h) and BMC1-M4, BMC2 and BMC4 showed highest doubling times (34.6 ± 2.2 h, 33.0 ± 3.0 h and 40.2 ± 15.1 h). Hence, BMC1-M1 and BMC1-M4 indeed showed less and higher progressive characters and even reflected the in vivo (Patient tumors level of Ki67) growth properties. These data are shown in Supplementary fig. 9a. As we hypothesized that an interaction with non-inflamed environmental brain cells might affect the proliferative properties, we performed intracranial injection of three different BMCs. However, we recapitulated the different growth properties and suggest that intrinsic cues might control the proliferative capacities of BMCs likely in concert with extracellular/environmental cues.

We made a new Figure (Figure 6) that investigates a potential progressive subset of MBM via a gene signature that has been deduced from a analysis of tumors M1 and M4. Although we cannot provide evidence proving that the drug-resistant subclone M4 directly emerged from M1, we can assume that M1 presents an early and more naïve state as M4. The latter has developed and progressed under BRAFi/MEKi therapy and did not respond to a subsequent ICI therapy. We investigated whether this “progressive” signature was observed in additional tumors of our cohort and indeed found signature genes expressed in additional MBM, particularly in a set of highly aggressive tumors of patient 23 (23-O and 23P). The identification of progressive genes was performed without any relationship to NGFR expression; however we found NGFR⁺ tumors among the progressive state MBM with one exception Pat22). Pat22 presents a leptomeningeal metastasis. As this type of tumors is per se associated with increased aggressiveness and even worse survival than cerebral metastases, we assume that these tumors might feature always an progressive phenotype.

C 2.11. Figure 5 D/E: Why not knockdown NGFR and/or manipulate Ecad to prove whether they are playing roles in brain metastases as most of the manuscript proposes? I am not convinced that showing a relative difference in brain met formation between these 2 cell lines (M4 versus M1) strengthens the central hypothesis of this manuscript.

Response: The knockdown of NGFR has been investigated in a lymph-node metastasis derived cell line (T2002) or a melanoma patient. This cell line featured a high tumorigenic capacity that was entirely diminished in NGFR knockdown cells⁴. Moreover, our previous results also suggest a critical role of NGFR for cell migration. Melanoma cells with overexpression or knockdown of NGFR clearly featured higher or lower migration and cells with knockdown of NGFR showed a reduced survival⁴⁻⁶. In line with previous finding, we observed a reduced migratory capacity of BMC1-M1 and BMC2 cells with knockdown of NGFR.

Our not yet published preliminary data suggested a higher metastatic capacity of A375 cells with overexpression of NGFR. The latter cells but not control cells with endogenously expressed NGFR formed metastases in the intestine and kidney after tail vein injection. However such an assay does

not necessarily provide much insight into the basic mechanism and more experiments are needed to investigate the programs fostering an NGFR-driven metastasis and brain metastasis. In line, Boiko et al. and Ngo et al. demonstrated that NGFR⁺ cells metastasize to lymph nodes, which was prevented by a NGFR-neutralizing antibody^{1,7}.

Generally, the simple overexpression of NGFR alone might not be sufficient for driving melanoma cells to the brain, considering that brain metastasis is mostly seen in patients with late stage disease and additional factors such as the activation of immune cells (as mentioned above) or a pre-activation of astrocytes and/or microglia might be required. A mouse model that is based on the establishment of metastases via tail-vein injection not necessarily recapitulates all these steps and the old model system that is based on the injection of tumor cells into the carotid artery easily leading to brain metastases excludes all the previous steps.

The intracranial injections are now shown in the context of a progressive phenotype. We determined a gene signature from the comparison of M1 and M4 tumors and investigated the entire set of MBM and associated BMCs for representation of the signature. We found that besides M4, signature genes were clearly expressed in Pat23 tumors and separated BMC1-M1 and BMC1-M4 cells. As previously suggested, the latter cell line was termed “progressive”. As the progressive and non-progressive states might have been established during *in vitro* cultivation we asked whether properties of intracranially injected cells might be affected by the non-inflamed brain parenchyma. However, we observed that growth properties of both cell lines were maintained and the proliferation of BMC1-M1 cells was not affected by the potentially supportive function of the brain microenvironment. As both cell lines harbored a same set of mutations and comparable levels of NGFR (and completely lack expression of Ecad protein that is crucial for establishing adherence junctions), other/additional cues might determine the different phenotypes.

C 2.12. The title of Figure 5 concerns “distinct migratory phenotypes” whereas the data focuses on a different cell surface marker profile and ability to grow in the brain when cells are injected directly into the brain. The evidence does not support the title of Figure 5

Response: We absolutely agree. As suggested by the reviewer, we restructured the content of all figures. The original Figure 5 does not exist anymore and the figure titles were changed

C 2.13. Figure 8B should show WBs or RT-PCR proof of knockdown of NGFR in the formal paper figures.

Response: As suggested by the reviewer, we now show the WB and a heat map representing differentially regulated genes as determined by RNAseq of BMC1-M1 cells with stable knockdown of NGFR in Fig. 7A. qRT-PCR results for NGFR knockdown are shown with regards to changes in

the migratory capacity of NGFR knockdown cells and additional investigated genes such as SOX4 in the right panel of Figure 7C.

C 2.14. The authors should show the consequence of manipulating Ecad expression and its consequence on invasion/migration/proliferation since Ecad is mentioned in the abstract as a key part of this story.

Response: We thank the reviewer for this comment. We now included data of BMC1-M4 and A375 cells that overexpress Ecad. Surprisingly, we observed that in contrast to all other cell lines, BMC1-M4 exhibited mRNA expression of Ecad at least at a low level, however Ecad expression was absent in all other cell lines as determined by qPCR and flow cytometry. As BMC1-M4 were more sensitive to dabrafenib treatment than BMC1-M1 cells, we asked, whether the Ecad expression might have been responsible for this finding. In Figure 5 we show that BMC1-M4 cells with overexpression of Ecad indeed featured a higher sensitivity to dabrafenib in a range of 1-30 nM than GFP expressing control cells. Although tumor M4 developed 2019 under BRAFi/MEKi therapy, the patient's therapy switched to nivolumab/ipilimumab right after M4 was detected. Hence, the tumor was probably on "drug holiday" and might have developed a sensitive phenotype, or as discussed in Fig. 6A was not directly correlated with tumor M1 or has developed dabrafenib sensitivity *in vitro*. However, as we observed Ecad⁺ areas in NGFR^{high} MBM, the switch of Ecad into NGFR expressing cells is likely reversible and probably represents cellular plasticity.

In addition, we investigated the proliferative capacity of BMC1-M4 cells with overexpression of GFP or Ecad^{GFP} and observed no statistical differences in proliferation. We included the following figure (Supplementary figure 8b):

C 2.15. The authors performed methylome profiling but it does not seem that any of their findings were validated using the cell lines, in terms of having a consequence on cellular proliferation/invasion/migration. Validating an aspect of the methylome analysis would greatly strengthen this story

Response: As suggested by the reviewer, we performed single-nucleotide primer extension (SNUPE) to validate an interesting, differentially methylated candidate. Our survey identified 46 differentially methylated CpGs but only some were significantly associated with survival. The most promising island is located in the ITGB7 gene, hypermethylated in NRAS mutant but hypomethylated in BRAF mutant MBM. We correlated expression and methylation of ITGB7 but found a low correlation and observed that ITGB7 was predominantly expressed in TIL^{high} MBM. We performed a survival analysis with expression data of the only currently available data set of MBM and observed a favorable outcome in MBM featuring a high level of ITGB7, hence tumors with a low methylation of ITGB7. In line with previous reports, NRAS mutated melanoma feature a TIL^{low} and more aggressive phenotype. We investigated expression levels of ITGB7 and CD3E, CD8A and CD79 (not shown) and observed a high correlation, indeed suggesting an immune cell rather than tumor cell-related expression of ITGB7. Studies in multiple myeloma suggested that ITGB7 regulates cell adhesion, migration and invasion and might suggest a role during immune cell infiltration of the brain and/or the tumors. Moreover, very recently Zhang et al. reported that increased levels of ITGB7+ leucocytes blocked the progression of colorectal cancer in a mouse model (DOI: 10.1158/2326-6066.CIR-20-0879)⁸. All results are shown in a new Figure (Figure 8).

3. Reviewer #3 (Remarks to the Author): Expert in RNA-seq

In this paper, Radke et al. describe the molecular characteristics of melanoma brain metastases. First they analyzed the transcriptome, methylome and mutational profile of a heterogeneous mixtures of intracranial metastases (pre or post therapy). They identified different subpopulations that can be clustered based on their Ecad and CD271 expression. They validated these findings using immunohistochemistry. They relied on pre-defined genesets to determine the 'invasive' subpopulation. They then complemented this work with in-vitro work to validate the causative role of CD271 loss (and SOX4) in inducing migration. While the biology is not completely novel its demonstration in melanoma brain metastases is.

Strengths of this study include:

Leveraging very valuable clinical specimens (fresh melanoma brain metastases)

Methods are described in details.

Complementing the genetic and immunohistochemical analysis with functional validation

We thank reviewer #3 for his/her remarks and valuable comments that have helped to substantially improve our manuscript. Basically, we restructured large parts of the manuscript particularly the results and figures 2-8 and discussion sections underwent considerable changes. Figures 8 and 9 were newly added.

C 3.1. While the authors leverage valuable clinical specimens, and the overall number of specimens appears adequate, some of the analysis focuses on samples obtained from one patient. While these provide useful information to trace the clones that have re-emerged after therapy, it would have been much stronger if they had multiple cases pre and post treatment. I realize that this may not be feasible.

Response: The reviewer is absolutely right. More samples and multiple cases would be nice to have and this is a limitation of this study. We revised the manuscript and mentioned this as possible limitation in the discussion part. Nevertheless, the material is very limited, especially from patients with different therapeutic intervention, post treatment, and post-mortem. Only in certain cases, a neurosurgical intervention is performed, e.g. to secure the diagnosis. Therefore, including more samples will - unfortunately - not be possible for this study. Moreover, publicly available expression data of therapy-naïve and MBM that developed under treatment *de facto* do not exist.

C 3.2. The authors should be careful not to draw pre-mature conclusions. They rely on predefined genesets, some of which were derived from different tumors (i.e. to define invasiveness) to assess tumor's invasiveness. While in general these genesets can serve as a correlate to increased 'invasiveness' of the tumor, the authors should not establish causation solely based on GSEA association. For example the title of figure 1. 'MAPKi treatment increases invasiveness of melanoma' is definitely not supported by the data and can be misleading. This also applies to their interpretation throughout the results section.

Response: We understand the reviewer's concern and modified Fig. 1 and changed the title accordingly. However, a selection of drug resistant and metastatic stem-like cells has already been demonstrated for glioblastoma and was clinically observed in melanoma^{9,10} among other cancer types. In addition, our previous study with A375 cells that expressed an NGFR-reporter demonstrated that dabrafenib subsequently enriched for NGFR⁺ cells¹¹.

C 3.3. The authors derive cell lines from tumors and perform transcriptome analysis along with established cell lines. The authors claim that these cell lines that they derived better recapitulate the in vivo tumor. Given that these cell lines were analyzed at low passages (1-4) compared to the established cell lines, this is to be expected. I am not sure what is the point of this comparison.

Response: We fully agree and removed this analysis. However, the analysis was meant to show whether BMCs cluster with conventional, long-term established or among each other and their respective tumors. Nevertheless, the clustering of BMCs among each other and their cognate tumors but not with A375 cells is indeed not surprising but was rather intended to show this difference.

C 3.4. What is more relevant is that they should compare the transcriptome of these derived cell lines (and tumors) to cells with CD271 knockdown.

Response: The knockdown (KD) of NGFR has been done in BMC1-M1 cells and a comparative analysis of NGFR KD in BMC1-M1 and in a lymph node-metastasis derived cell lines revealed 33 overlapping genes. This analysis is shown in Fig. 7a and Supplementary fig. 10a and suggests a direct or indirect association of these genes with expression of NGFR. Investigation of these genes in cBioPortal revealed a correlation of expression of ADAMTS9, ALDH1A3, DHRS3, CADM1 and TMTC2 with NGFR.

C 3.5. The authors should list all the genesets that they have used (including the referenced ones) in a separate supplementary document.

Response: We thank the reviewer for this comment and added a Supplementary file including all gene sets that have been used (please see Supplementary file 10).

C 3.6. While the authors described the technical details, the paper may benefit from a 'big picture' perspective in the introduction. For example, the authors do not explain the rationale of using CD271 as a marker of aggressive melanoma (Figure 1).

Response: As suggested by the reviewer, we revised the introduction accordingly, explaining the current knowledge on CD271 and cited more literature on CD271 and melanoma metastasis.

C 3.7. The authors should also reference relevant papers in the introduction. The authors should also reference and discuss the work by (Restivo, Nature Communications) in the introduction given that this paper was among the first to discuss the role of CD271 in inducing a phenotypic switch in melanoma.

Response: As suggested, we revised the introduction now providing additional information on NGFR and included the following sentence in line 104-108, p. 3-4: "Particularly, the latter non-genetic process, enabling the switching of melanoma cells within different phenotypical states controls growth and invasiveness via modification of levels of NGFR expression¹². Likely, phenotype switching is strongly effected by environmental cues such as inflammatory processes that fostered dedifferentiation and enrichment of NGFR⁺ melanoma cells"^{5,13}.

The following studies were cited:

- 1 Bao, S. *et al.* Glioma stem cells promote radioresistance by preferential activation of the DNA damage response. *Nature* **444**, 756-760, doi:10.1038/nature05236 (2006).
- 2 Haueis, S. A. *et al.* Does the distribution pattern of brain metastases during BRAF inhibitor therapy reflect phenotype switching? *Melanoma Res* **27**, 231-237, doi:10.1097/CMR.000000000000338 (2017).

- 3 Vidal, A. & Redmer, T. Tracking of Melanoma Cell Plasticity by Transcriptional Reporters. *International Journal of Molecular Sciences* **23**, 1199 (2022).
- 4 Boiko, A. D. *et al.* Human melanoma-initiating cells express neural crest nerve growth factor receptor CD271. *Nature* **466**, 133-137, doi:10.1038/nature09161 (2010).
- 5 Ngo, M. *et al.* Antibody Therapy Targeting CD47 and CD271 Effectively Suppresses Melanoma Metastasis in Patient-Derived Xenografts. *Cell Rep* **16**, 1701-1716, doi:10.1016/j.celrep.2016.07.004 (2016).
- 6 Radke, J., Rossner, F. & Redmer, T. CD271 determines migratory properties of melanoma cells. *Sci Rep* **7**, 9834, doi:10.1038/s41598-017-10129-z (2017).
- 7 Redmer, T. *et al.* The role of the cancer stem cell marker CD271 in DNA damage response and drug resistance of melanoma cells. *Oncogenesis* **6**, e291, doi:10.1038/oncsis.2016.88 (2017).
- 8 Redmer, T. *et al.* The nerve growth factor receptor CD271 is crucial to maintain tumorigenicity and stem-like properties of melanoma cells. *PLoS One* **9**, e92596, doi:10.1371/journal.pone.0092596 (2014).
- 9 Restivo, G. *et al.* low neurotrophin receptor CD271 regulates phenotype switching in melanoma. *Nat Commun* **8**, 1988, doi:10.1038/s41467-017-01573-6 (2017).
- 10 Guo, R. *et al.* Increased expression of melanoma stem cell marker CD271 in metastatic melanoma to the brain. *Int J Clin Exp Pathol* **7**, 8947-8951 (2014).

C 3.8. Also, Reference 84 is the correction and not the original publication.

Response: We substituted the reference by the original publication.

C 3.9. The wound scratch assay lasts for 128 hours. This is a long time. The authors should also demonstrate that there is no increased proliferation. Alternatively, they can use mitomycin to inhibit cell proliferation and assess migration only.

Response: We agree, investigated the proliferation of BMCs (Supplementary fig. 9a, right panel) and indeed observed distinct proliferative capacities with BMC1-M4 showing the highest level of proliferation. To exclude or reduce proliferation effects we determined the “relative wound density (RWD)” instead of the “wound width”. As the RWD reflects a value which is normalized for changes in cell density caused by proliferation, we used this value to assess the migratory properties of BMCs. The following sentence was added to the Material&Methods section.

To assess the migratory properties of BMCs, we determined the relative wound density that is determined as the percentage of spatial cell density in the wound area relative to the spatial cell density outside of the wound area at each time point¹⁴. This metric reflects a value which is normalized for changes in cell density caused by proliferation and is defined as:

$$\text{RWD}(\%) = \frac{100 * [w(t) - w(0)]}{[c(t) - w(0)]}$$

RWD = relative wound density; w(t)=Density of wound region at time, (t) ; c(t)=Density of cell region at time, (t).

In addition, we tracked wound closure of BMC1-M1 cells in the presence of mitomycinC (10, 30 µg/ml) and indeed observed a reduced cell migration, however this effect was due to the toxicity of mitomycin but showed no effect at early time points of the assay. We added the following diagram to Supplementary figure 10d:

Moreover, as we now determined RWD, we added the following

text to the results section: “We determined the relative wound density, reflecting a value which is normalized for potential changes in cell density caused by proliferation. We observed a rapid wound closure and increase in the relative wound density

(RWD) by BMC2 (69.9±11.1%), BMC1-M4 (45.3±0.5%) and BMC4 (42,5±5,9%) but a reduced migratory capacity of BMC3 (38.0±3.1%) and BMC1-M1 cells (26.4±1.0%) 24h after wounding (Fig. 7c, first panel). Nevertheless, the migratory phenotype of BMC1-M1 cells, was significantly ($p=3.0e-03$) reduced upon NGFR downregulation and decreased in BMC2 cells upon knockdown but without statistical significance (Fig. 7c, center panel, Supplementary fig. 10c, right panel). As we could not fully exclude that a high proliferative phenotype affected wound closure we blocked cell proliferation by mitomycinC (MMC). Although we observed a decreased migratory capacity of BMC1-M1 cells at later time point in the presence of MMC, we found this effect caused by toxicity of the inhibitor (Supplementary fig. 10d).

As the repeated washing steps would have affected the cell layer, we decided to keep MMC on cells during the assay. However, this caused cellular toxicity that was evident by a decrease in migration. The proliferation assay suggested a MMC-mediated inhibition of proliferation already after 8-12h on MMC, however, migration was not affected within 0-20h.

Nevertheless, proliferation and migration/invasion are part of equilibrium and controlled by levels of certain proteins such as NGFR, hence always connected. Restivo et al., demonstrated that high levels of NGFR indeed promote migration and low levels foster cell proliferation, probably suggesting that cells might also exist in an intermediate state, enabling proliferation and migration. The knockdown of NGFR likely affected both proliferation and migration. Indeed we observed a strong decrease in proliferation of T2002 cells (a BRAF/NRASwt lymph-node metastasis-derived cell line established from a melanoma patient).

C 3.10. Sequencing was done using Illumina Platform and IonProton (Thermo). Were some tumors analyzed by one modality, or were all analyzed by both platforms. Can the authors ensure adequate and comparable coverage among all tumors? This is important since they are using this information to infer the persistence of specific subclones. Figure 6.

Response: The majority of tumor libraries was initially sequenced by using the IonProton platform with a mean coverage of 500x-1,000x. However, all Pat8 tumors and derived cell lines as well as tumors and cell lines of patients 27 and 35 that were used for investigating the potential persistence of subclones were sequenced on same platform (TSO500 libraries, NextSeq2000) with a mean coverage of 750x. Nevertheless, we modified the Fig. 6 and rather focused on the comparability of the genetic background. We observed a high recovery of public and private mutations (subclones), suggesting high reliability of the system.

C. 3.11. A schematic diagram at the end summarizing the main findings could be helpful.

Response: We fully agree and summarized our main findings in a summarizing main Figure (Figure 9):

Reviewer #4 (Remarks to the Author): Computational expert

In this manuscript, Radke et al investigate through transcriptome and methylome profiling a set of melanoma brain metastasis in order to identify molecular pathways, subgroups and potential therapeutic possibilities. I can appreciate that the authors have done an impressive amount of work, and I believe the experiments are in general carefully done and the figure are informative and aesthetically pleasant. However, I found the writing difficult to follow, as I think there are too much data that (in my humble opinion) perhaps confuse more than clarify their message.

We thank reviewer #4 for his/her remarks and valuable comments that have helped to substantially improve our manuscript.

C 4.1. Though I can appreciate the incredible amount of work that went into this manuscript, I am unsure of what the main message is, and I struggled several times to remember why certain experiments were being performed. Is the main message that melanoma brain metastases (MBMs) can be subdivided in invasive (NGFR+) and proliferative (Ecad+) and that these may be targeted by repurposed drugs? Is it? I would recommend simplifying the manuscript and perhaps focus on one main, clear message.

Response: We are sorry that our main messages might get lost in this large amount of data and we might have not been clear enough. The main purpose of our study was to uncover molecular subgroups of brain metastases that might explain the different therapeutic response in MBM from different patients but also different MBM in one patient. Our analysis identified at least two subgroups: Ecad⁺ MBM featuring a pigmented, rather proliferative than invasive phenotype and a subgroup of MBM exhibiting a stem-like, MRD/drug-resistant and invasive/migratory phenotype. As a high level of NGFR expression is thought to mediate a drug-resistant phenotype, we suggest that NGFR^{high} and Ecad^{high} MBM differentially respond to conventional, clinically used drugs.

Whether Ecad or NGFR expression is maintained or modified (increased, decreased) or completely lost during the metastatic process and simply re-acquired at distant sites such as the brain remains elusive. However, our previous analyses of levels of NGFR in primary melanoma and concordant metastases revealed that NGFR expressing cells were present or absent in all tumors, hence was not triggered by brain parenchymal cells. As for NGFR, Ecad⁺ MBM probably derived from Ecad⁺ primary tumors or extracranial metastases and the difference in Ecad levels of both tumors is marginal.

Therefore, we propose at least two subtypes of MBM, however suggest that NGFR⁺ MBM likely exhibit a more progressive phenotype than Ecad⁺ MBM. Inflammation and additional environmental cues such as therapeutic interventions might foster the transition of Ecad⁺ into NGFR⁺ tumors. This process might be reversible. Hence, Ecad and NGFR are likely interconnected and involved in a phenotype switching process. We have previously shown that plasticity can include different markers such as NGFR and CD133⁴. Genetically and epigenetically, the two subgroups might be associated with presence of mutations in BRAF or NRAS but are not associated with presence of immune cells.

We absolutely agree that the characterization of MBM-derived cell lines regarding the persistence of subclones is out of the focus. We restructured the manuscript and now focused on the most important points.

The processes that drive intracranial progression are quite complex and determined by the activation of astrocytes and microglia. The latter in turn secrete several factors, e.g. growth factors such as FGFs, NGF, chemokines, cytokines etc. that promote MBM progression. However, how this exactly proceeds is poorly understood. Our survey provides insights into the different subgroups and demonstrated that these subgroups emerge independently of the intracranial site. Consequentially, the therapeutic targeting of molecularly distinct subgroups of tumor cells fosters relapse.

C 4.2. Can it be stated clearly if the authors think that MBMs recapitulate what has been described for cutaneous MRD (For example, in the paper by Rambow et al, Cell 2018)? Do the authors anticipate that their proposed method of targeting MBMs would work in vivo?

Response: In the revised manuscript, we included an analysis showing that MRD-associated genes are indeed expressed in NGFR^{high} MBM (Fig. 4C, right panel) which is line with the assumptions of Rambow et al. that MRD is indeed controlled by a neural crest stem cell (NCSC) program. We have previously shown that the knockdown of NGFR was accompanied by a loss of NCSC-related genes (Redmer et al., 2014) and strongly diminished the tumorigenic capacity of melanoma cells. The targeting of MBMs needs to be adjusted to the subtype, although we removed the MET-receptor targeting related data, this might be an option. Our subsequent studies focus on the functional characterization of MBM-related genes and hopefully will identify promising and druggable key drivers.

In my opinion, some assumptions are perhaps not completely backed up by their data. For example:

C 4.3. Line 462-463: How does this level of staining compare to non-treated or non-resistant MBM? Is there a baseline with which to compare to back up the claim that CD271 expression in these metastases is high? Were the tumours in Figure 1B from brain metastases too? Also, later in

that sentence authors say that this expression may be "promoted by reactive astrocyte secreted factors" but what is their evidence to say this? Did they stain for such factors?

Response: We agree that this is difficult to follow. Tumors showing a high number >50% of CD271/NGFR⁺ cells were termed as high. Tumors were scored regarding the quantity of positive cells as determined by counting of 50 visual fields at 200x magnification and by relative levels of NGFR expression (has now been added to the Material&Methods section). The comparison of concordant non-treated or non-resistant MBM is difficult as these MBM are often not surgically removed and the material is very rare. The comparison of MBM M1 that only received radiation and immune checkpoint inhibitor (ICi) therapy but not BRAF inhibitor (BRAFi) therapy, (that very likely selects for resistant clones) with M4 that developed under BRAFi represents the best model one can achieve. Concordant, early MBM M1 and M2 featured a low NGFR and pigmented phenotype. M1 showed a high level of Ecad expression and M3 and M4 demonstrated increased levels of NGFR. A comparative analysis of transcriptome data of M1 and M4 clearly revealed the "Ecad-to-NGFR switch". Hence, we are mostly looking at MBM from late stages but sometimes have the opportunity of receiving an early stage MBM that can be compared with concordant, later stage tumors. Nevertheless, this material is very rare.

Yes indeed, the tumor in Fig. 1B shows a MBM surrounded by GFAP expressing astrocytes showing the typical flattened and not star-type morphology. These reactive astrocytes (of murine models) have been profiled and hence the factors that are expressed and potentially secreted are well known. However, the staining of these factors is quite challenging due to the lack of suitable antibodies. However, we used transcriptome data of tumors that contained GFAP⁺ astrocytes and found markers of reactive astrocytes and microglia such as interferons, CXCL10, IL6 as well. The tumor in Fig. 1B was used as a representative of a highly progressive and therapy-refractory tumor, the reactive state was proven by GFAP staining and presence of a typical morphology.

C 4.4. Line 501 - "...identified a pan-gene signature potentially defining MBM showing enrichment of extracellular exosomes and genes that are associated with focal adhesion or defining extracellular matrix components (ECM) or melanosomes". How was this pan-gene signature defined? Are the authors suggesting that melanocytes within those BM are expressing these genes but not the surrounding cells? What is this assumption based on? The authors compare against a set of controls (supp figure 2) but how do they know that normal brain tissue is comparable to non-melanocyte cells within a tumour environment?

Response: We decided to remove these classifier signatures as they are indeed misleading. Melanocytes and melanoma cells share a couple of genes that are maintained during the transition of melanocytes into melanoma cells. The Pan-MBM gene signature was derived from a comparative analysis of MBM and brain controls hence brain tissue from autopsied patients. Although this was

not concordant brain tissue (as this is impossible to acquire) we clearly observed a set of genes that were only expressed in MBM but not brain controls (BC). We moved the fig. to the main figure (now Figure 2c) and point out to some important genes. The set of MBM-associated genes very likely overlaps with genes that are expressed in primary melanoma and/or extracranial metastases, suggesting that these genes are potentially crucial to enable metastasis to the brain and needed for maintenance of MBM. We run this analysis because MBM are often surrounded by normal brain tissue and this “subtractive” method enabled us to clean the MBM gene set.

C 4.5. A line of logic I found confusing but I think is crucial for the message the authors want to deliver is in line 593: "Likely, brain metastatic tumor cells exhibit a rapid EMT-MET capacity and metastases re-acquire Ecad expression soon after every metastatic step." - As I understand, the authors have analyzed so far the gene expression of tumours before and after RAFi/MEKi therapy (Figures 1,3) and see an association of CD271 expression with more invasive characteristics. They suggest that an Ecad-to-CD271 switch marks a final step of MBM progression (line 560) but then they analyze more metastases (from other datasets including TCGA) but do not see much difference in expression. I think the assumption that metastases re-acquire Ecad after every metastatic step (what do they define as a metastatic step?) requires more backing. For example, was the TCGA data cleaned to keep melanocytes only (as the authors did previously with their dataset)? The methods don't seem to list how the authors analysed these data, but apologies if I have missed it.

Response: The Ecad-to-CD271/NGFR transition presents a special case of EMT as MBM and melanoma cells in general never acquire a “full” epithelial state. This state is seen in typical epithelial cells and murine embryonic stem or iPS cells, hence express Ecad but no mesenchymal marker such as vimentin. Melanoma show high levels of vimentin expression regardless of the tumor type (primary or metastasis) and can also express Ecad. The transition presents a phenotype-switching process and is therefore a consequence of cellular plasticity. TCGA data mostly contain primary melanoma and metastases. However, these data do not need a cleaning step as performed for MBM. Although melanoma might contain admixed cells such as fibroblasts or endothelial cells, they do not contain brain-parenchymal cells. We investigated expression levels of typical melanoma marker such as PRAME, PMEL or SOX10 and assumed the TCGA-melanoma as clean. In addition, the origin and tumor cell content of all melanoma specimens was verified.

As every tumor represents a spatiotemporal snapshot of evolution, they express Ecad or NGFR or both. As all tumors, primary and metastatic of the TCGA-SKCM data set comprises all these phenotypes the levels of Ecad are comparable. However, in NGFR^{high} vs low subsets, we observed a significant difference in Ecad/CDH1 levels (Reviewer-only figure, not included in the manuscript):

Nevertheless, we cannot explain why levels of Ecad expression are high in NGFR^{high} MBM. Maybe data suggest that double positive melanoma cells are more prone to brain metastasis, however due to the low number of samples (n=6) included in the TCGA these data are not representative.

There were several instances of sentences I found confusing and that I think it would be good to review:

*Introduction

C 4.6. Line 102 - "The time from initial diagnosis of primary tumors to the detection of BM ranges from 1-10 years, supporting the assumption of a slow evolutionary process of BM in 20 – 40% of melanoma patients". - Is this adjusted by stage at diagnosis? I wonder if the patients with detected BM 1 year after diagnosis are at a more advanced stage than those with BM detected 10 years after diagnosis?

Response: In melanoma patients, late recurrence or development of brain metastasis even 10 years after initial diagnosis is not unusual. These patients might have had a stage II or stage III melanoma that has been surgically removed and the patient was considered to be “cured”. Indeed, the reviewer is right, patients who are diagnosed with BM 1 year after initial diagnosis are at stage IV melanoma and 20-40-% of all melanoma patients develop BM during the course of disease.

C 4.7. Please clarify what MBM refers to (where it is first mentioned it's unclear what it is abbreviating). Related to this, I find the beginning of line 108 confusing - seems to imply extracranial metastases are melanoma brain metastases? Please clarify this point.

Response: We understand that this might be a little confusing because sometimes we refer to brain metastasis (BM) in general and sometimes specifically to melanoma brain metastasis (MBM). We revised this. Whenever we talk about the findings in melanoma brain metastasis, it now says MBM.

The abbreviation BM no longer exists in the text. Whenever we refer to brain metastasis in general, we wrote "brain metastasis". We hope this makes this more simple and clear

C 4.8. Line 118 - "responsible for a non-genomic cellular heterogeneity" - should this sentence drop the "a"?

Response: This sentence was deleted in the course of restructuring the manuscript text.

C 4.9. Line 131 - "non-genetic emergence of minimal-residual disease" - what does this refer to? Do the authors want to say "non-genetic mechanisms of resistance in MRD" or something similar?

Response: This sentence was deleted in the course of restructuring the manuscript text.

C 4.10. Line 132 - "Recent work has shown that melanoma cells secreted extracellular vesicles were enriched with CD271 that was taken up by lymphatic endothelial cells, therefore aiding lymph node metastasis". - This reads confusing to me, would it be better to say "Extracellular vesicles secreted by melanoma cells were enriched with CD271...".?

Response: We revised the manuscript text according to suggestions raised during the review process. The role of extracellular vesicles is not discussed anymore, so this sentence was removed.

*Methods

C. 4.11. Line 157 - Should this be "cell lines or iii) isolation of RNA and DNA or IV) formaldehyde-fixed and paraffin embedded (FFPE) and archived.? (Is "IV" missing?)

Response: We are sorry that this was confusing. We changed the sentence accordingly:

"Tumor pieces were split into parts of equal size and i) snap-frozen in liquid nitrogen and stored at -80°C for later isolation of RNA and DNA, and ii) formaldehyde-fixed and paraffin-embedded (FFPE) for (neuro-)pathological work-up. If additional sufficient material was available, it was used for the establishment of MBM-derived cell lines (BMCs)."

C 4.12. Line 200 - Should this be "and edited WITH Adobe Photoshop"? Also, please specify that the modifications to the image do not alter results in any way.

Response: We changes this and added the following sentence to the Statistics & Reproducibility section: "shown were adjusted in brightness and contrast to different degrees (depending on the need resulting from the range of brightness and contrast of the raw images) in Adobe Photoshop 2020; however the modifications to the image did not alter results in any way".

C 4.13. Line 210 - Please add antibody ID and if possible lot number.

Response: We added the antibody IDs and LOT numbers to the text.

C 4.14. Line 248 - "Isolation of total RNA from snap frozen or intraoperative tumors" - This line confuses me. Above it says that intraoperative tumours were snap-frozen among a number of possibilities. Does this mean "snap frozen or fresh.."?

Response: We are sorry about this confusion. RNA was isolated from snap frozen tumor specimen. We changed the sentence accordingly:

"Isolation of total RNA from snap frozen tumors was performed with the RNAeasy extraction kit (Qiagen) according to the manufacturer's instructions."

C 4.15. Line 250 - Please split the sentence starting in this line "The library preparation...."

Response: We split the sentence accordingly:

"The library preparation of 100 ng total RNA was performed with TruSeq Stranded total RNA Sample Preparation-Kit and Ribo-Zero Gold Kit (Illumina). Paired-end (2x100 bp) whole transcriptome profiling of RNA with integrity numbers (RIN) ≥ 7 was performed at Cegat GmbH, Tuebingen (Germany) and sequenced on NovaSeq6000 platform."

C. 4.16. Line 282 - "Data of the [1000 genomes project] enabled the separation of single nucleotide polymorphisms (SNPs) from mutations (single nucleotide variants, SNVs)." - Was this the only database used? I would argue it is better to use the GnomAD sets as they contain many more individuals.

Response: We agree that GenomAD covers more genomes than 1000Genomes Project. However, we also used the COSMIC database for validation providing information of >37,000 genomes.

C 4.17. Line 385 - Can the number of cells used for the inoculation be specified? Were all mice inoculated with the same number of cells?

Response: We used 2.5×10^4 cells and yes, all mice were inoculated with the same number of cells. The information is given in the Material and Method section (under animal experiences).

*Results

C 4.18. Line 488 - "envirnomenta cells" - do the authors mean cells in the extracellular microenvironment?

Response: This was changed to: [...] to environmental cues is likely determined by the cellular composition of the BTE." Now line 495.

C 4.20. Line 490 "we collected intraoperative and cryo-preserved MBM (n=16; Supplementary Table 1) from different intracranial sites" - Supplementary Table 1 lists more than 16 MBM, with different patient IDs, can it be clarified please which samples were used, or what these 16 are? There seem to be inconsistencies with Figure 2a (This figure lists patient 6 but the table does not?)

Response: We are sorry that this was misleading, the list contains Pat 6a. This was corrected. The Supplementary table 1 contains all patient tumors that have been investigated in our study. We now used a color code to indicate the different methods. Eighteen samples (n=16 MBM and n=2 BMC) have been investigated by RNAseq.

C 4.21. Line 492 - This sentence is hard to understand ("the latter synchronous metastases of patient 23 that..."), please clarify.

Response: We removed this sentence.

C 4.22. Line 512 - perhaps a good paper to cite here is Rabbie et al (2020), who have done an analysis of the mutational status of driver genes within melanoma brain metastasis. Following from that - do the authors not see KRAS mutations in these MBM?

Response: We thank the reviewer for bringing our attention to the paper by Rabbie et al., which we now included in the discussion as their concordant MBM (n=5) of a drug-naïve patient featured expression of markers of Ecad^{high} MBM. The following text has been added on p.32 (lines 868-877):

Our hypothesis is strengthened by the uncovering of molecular features of drug-naïve MBM by Rabbie et al.¹⁵ & Biermann et al.³, suggesting that this set of MBM likely features an Ecad+ rather than NGFR+ state. Indeed the top-expressed genes as identified in the studies overlapped with the signature of Ecadhigh tumors that was identified in this study. Surprisingly, gene signatures of both studies did not overlap which might be a consequence of the different methods used (bulk vs. sc-RNAseq) or tumor of both subsets featured different developmental states. However, we observed Ecad-associated genes such as ABCB5 or pigmentation-related factors such as MLANA, PMEL or DCT expressed in either of the studies (Figure 9b)¹⁵.

Regarding the presence of KRAS mutations: no, we did not observe tumor cell clones (even not low frequency clones) that harbored a KRAS mutation.

C 4.23. Line 515 - Can the authors please specify which tumours underwent methylome profiling? I think it is important to ensure they all have the same origin (i.e. snap frozen or FFPE) or alternatively that this has no effect on the results described.

Response: All tumors that underwent methylome profiling are now highlighted in Supplementary fig. 1. We did not observe a clustering regarding a possible batch effect by the material used (fresh, snap frozen, FFPE) for DNA isolation.

C 4.24. Line 552 - "suggesting that the emergence of CD271-driven subclones..." - how do the authors know that CD271 drives these sub clones and is not only a biomarker?

Response: Melanoma cells can acquire expression of CD271/NGFR under stress conditions e.g. mediated by inflammatory processes or therapeutic drugs and this is known to mediate increased survival, stem-like properties and modify the migratory properties. Our previous work suggested that melanoma cells expressing NGFR are enriched under dabrafenib treatment and show an MRD phenotype and the knockdown of NGFR was accompanied by a strong reduction in proliferation and loss of tumorigenicity. We agree, that we do not have evidence to prove that NGFR expression indeed drives subclonal evolution *in vivo*, however previous studies by us and others strongly suggest that NGFR⁺ cells feature higher survival and migratory properties^{2,5,16,17}.

C 4.25. Line 618 - Should the "though" be changed to "however" so the sentence is completed?

Response: This sentence was deleted in the course of restructuring the manuscript text.

C 4.26. Line 749 - Should "public" mutations be "established", or "known"?

Response: We changed this to: "Apart from shared, we detected private mutations which were exclusively found in either of the longitudinal tumors". Now in line 659-660.

*Figures

C 4.27. Supp Figure 3B has several gene names that say "NA", please correct this (if the CpG is not associated with a gene, can the location be marked?) Also, this figure is referenced in the statement "The matching with transcriptome data revealed 14 MBM expressed genes that featured high methylation of promoters in BRAFwt tumors and identified integrin b7 (ITGB7) as a potential predictor of favorable survival". Can the authors please specify what these 14 genes are?

Response: We agree and prepared a new Main figure 8 showing now a heat map that represents all identified CpGs/genes that were differentially methylated among BRAFmut and BRAFwt MBM. However, we cannot provide all gene symbols as they have not been assigned during analysis and are not clearly definable via EWAS database. We now included a Supplementary table 11 providing detailed information about all identified CpGs.

C 4.28. Figure 3a - Do the vertical lines crossing the timeline represent all tumours removed from P8, but only M1-4 were used in the study? What does BMC stand for?

Response: BMC is the abbreviation for MBM-derived cell lines (BMCs). To make this clear, we additionally introduced the abbreviation in each figure legend. As indicated in Supplementary table 1, tumors M1 and M4 were investigated by RNAseq and TargetSeq; Tumor M2 was investigated by methylome profiling and TargetSeq (50 gene panel); M3 was only investigated by TargetSeq (~500 gene panel).

C 4.29. Figure 3D - Can the label DECMA1 be changed for Ecad? (As in the left panel)

Response: We thank the reviewer for this notice and changed the label accordingly.

C 4.30. Supp Figure 5A. There doesn't seem to be much difference in E-cadherin expression among the TCGA tumours, the p value is probably driven by a few outliers.

Response: We agree, however the comparative analysis of EM (extracranial metastases)+BM (brain metastases) with primary tumors (PT) resulted in a comparable p-value and is now shown in Supplementary fig. 3C. As we are comparing a lot of samples, namely EM (n=321) and PT(n=151), we don't believe that the p-value is driven by some outliers.

References

- 1 Boiko, A. D. *et al.* Human melanoma-initiating cells express neural crest nerve growth factor receptor CD271. *Nature* **466**, 133-137, doi:10.1038/nature09161 (2010).
- 2 Marchetti, D., McCutcheon, I., Ross, M. & Nicolson, G. Inverse expression of neurotrophins and neurotrophin receptors at the invasion front of human-melanoma brain metastases. *Int J Oncol* **7**, 87-94, doi:10.3892/ijo.7.1.87 (1995).
- 3 Biermann, J. *et al.* Dissecting the treatment-naive ecosystem of human melanoma brain metastasis. *Cell* **185**, 2591-2608 e2530, doi:10.1016/j.cell.2022.06.007 (2022).
- 4 Redmer, T. *et al.* The nerve growth factor receptor CD271 is crucial to maintain tumorigenicity and stem-like properties of melanoma cells. *PLoS One* **9**, e92596, doi:10.1371/journal.pone.0092596 (2014).
- 5 Radke, J., Rossner, F. & Redmer, T. CD271 determines migratory properties of melanoma cells. *Sci Rep* **7**, 9834, doi:10.1038/s41598-017-10129-z (2017).
- 6 Redmer, T. *et al.* The role of the cancer stem cell marker CD271 in DNA damage response and drug resistance of melanoma cells. *Oncogenesis* **6**, e291, doi:10.1038/oncsis.2016.88 (2017).
- 7 Ngo, M. *et al.* Antibody Therapy Targeting CD47 and CD271 Effectively Suppresses Melanoma Metastasis in Patient-Derived Xenografts. *Cell Rep* **16**, 1701-1716, doi:10.1016/j.celrep.2016.07.004 (2016).
- 8 Zhang, Y. *et al.* Integrin beta7 Inhibits Colorectal Cancer Pathogenesis via Maintaining Antitumor Immunity. *Cancer Immunol Res* **9**, 967-980, doi:10.1158/2326-6066.CIR-20-0879 (2021).
- 9 Bao, S. *et al.* Glioma stem cells promote radioresistance by preferential activation of the DNA damage response. *Nature* **444**, 756-760, doi:10.1038/nature05236 (2006).

- 10 Haueis, S. A. *et al.* Does the distribution pattern of brain metastases during BRAF inhibitor therapy reflect phenotype switching? *Melanoma Res* **27**, 231-237, doi:10.1097/CMR.0000000000000338 (2017).
- 11 Vidal, A. & Redmer, T. Tracking of Melanoma Cell Plasticity by Transcriptional Reporters. *International Journal of Molecular Sciences* **23**, 1199 (2022).
- 12 Restivo, G. *et al.* low neurotrophin receptor CD271 regulates phenotype switching in melanoma. *Nat Commun* **8**, 1988, doi:10.1038/s41467-017-01573-6 (2017).
- 13 Guo, R. *et al.* Increased expression of melanoma stem cell marker CD271 in metastatic melanoma to the brain. *Int J Clin Exp Pathol* **7**, 8947-8951 (2014).
- 14 Johnston, S. T., Shah, E. T., Chopin, L. K., Sean McElwain, D. L. & Simpson, M. J. Estimating cell diffusivity and cell proliferation rate by interpreting IncuCyte ZOOM assay data using the Fisher-Kolmogorov model. *BMC Syst Biol* **9**, 38, doi:10.1186/s12918-015-0182-y (2015).
- 15 Rabbie, R. *et al.* Multi-site clonality analysis uncovers pervasive heterogeneity across melanoma metastases. *Nat Commun* **11**, 4306, doi:10.1038/s41467-020-18060-0 (2020).
- 16 Lehraiki, A. *et al.* Increased CD271 expression by the NF- κ B pathway promotes melanoma cell survival and drives acquired resistance to BRAF inhibitor vemurafenib. *Cell Discov* **1**, 15030, doi:10.1038/celldisc.2015.30 (2015).
- 17 Marchetti, D., Aucoin, R., Blust, J., Murry, B. & Greiter-Wilke, A. p75 neurotrophin receptor functions as a survival receptor in brain-metastatic melanoma cells. *J Cell Biochem* **91**, 206-215, doi:10.1002/jcb.10649 (2004).

REVIEWER COMMENTS

Reviewer #2 (Remarks to the Author):

The authors have adequately addressed my comments.

Reviewer #3 (Remarks to the Author):

in the revised manuscript the authors addressed most of my concerns. It is easier to follow the manuscript and the authors tried to highlight the main message.

Regarding the wound scratch assay, it seems from the description of the methods section that mitomycin was added for few hours/days and they observed significant toxicity. Mitomycin should be added for an hour prior to the wound scratch assays. The authors attempted to distinguish between relative wound density and confluence. I think this is more confusing (Lines 795-806) and results are harder to interpret. I suggest repeating the experiment by adding mitomycin 10ug/ml for just 1 hour before the scratch assay.

Reviewer #4 (Remarks to the Author):

I am really thankful to the authors for being so thorough in their responses and for helping me understand their work better. I think they have thoughtfully addressed all my comments, and I am satisfied with their manuscript.

Point-by-point responses to the comments of reviewer#3:

We have used different font colours to differentiate between the reviewers' comments (**black font colour**), our responses (**blue font colour**), and revised manuscript passages (**orange colour**). We uploaded a manuscript version, where all changes are highlighted in yellow.

Reviewer #3 (Remarks to the Author):

In the revised manuscript the authors addressed most of my concerns. It is easier to follow the manuscript and the authors tried to highlight the main message. Regarding the wound scratch assay, it seems from the description of the methods section that mitomycin was added for few hours/days and they observed significant toxicity. Mitomycin should be added for an hour prior to the wound scratch assays. The authors attempted to distinguish between relative wound density and confluence. I think this is more confusing (Lines 795-806) and results are harder to interpret. I suggest repeating the experiment by adding mitomycin 10ug/ml for just 1 hour before the scratch assay.

Response:

We fully understand the reviewer's concerns regarding the results of the mitomycin C (MMC) treatment. Before testing the effect of MMC on cell migration, we performed a proliferation assay to investigate whether the incubation for one hour with the standard concentration of 10µg/ml MMC might even inhibit proliferation. We did not include the results as we did not observe any differences in proliferation with this MMC concentration, neither for BMC1-M1 nor for BMC1-M4 cells, which showed a higher proliferative capacity (Reviewer only figure 1a). Hence, the repetition of the migration assay with the suggested concentration of 10µg/ml MMC would not answer the question of whether we can exclude that cell proliferation might have influenced/biased the results of the migration assay.

As also noticed for several other drugs, MMC is not effective in all cancer cell lines and shows highly variable responses as reflected by IC₅₀ values even among melanoma cell lines (please see Reviewer only figure 1b, red arrow). Therefore, we suggest that BMCs might individually respond to MMC.

Consequently, we performed proliferation and migration assays in presence of MMC with doses of 10 µg/ml and 30 µg/ml, expecting a toxicity effect after some time.

The calculation of the relative wound density, meaning the density of cells inside the wound relative to the density of all cells (outside and inside the wound) corrects for proliferative effects. Thus, it reflects mostly the migration effects. However, as suggested by Restivo et al. proliferation and migration/invasion of melanoma cells and probably of other cancer cell types are likely not strictly separated cellular properties. However, both rather reflect plastic states that are modifiable via levels of certain genes such as NGFR. Whether melanoma cells can migrate and proliferate at the same time or melanoma cell pools comprise of migrating or proliferating subsets is unknown. Whether and to which extent the proliferative subset can be blocked with MMC is not defined considering the different responses of different cells/cell lines.

Figure 1: A.) Initial testing of MMC in BMCs: BMC1-M1 and BMC1-M4 cells were seeded and incubated with MMC, 10 µg/ml for 1h. Following removal of MMC and gentle washing the proliferative capacity of cells was monitored for 44h. However, MMC showed no effects in both cell lines. B.) IC₅₀ values for MMC (µM) among cancer cell lines as retrieved from <https://www.cancerrxgene.org> suggesting a strong variability of cell's response to MMC and even among melanoma cell lines (red arrow). C.) Comparison of early time points of migration (left) and proliferation (right) of BMCs suggesting related proliferation but distinct migration of BMCs.

To verify or falsify that the migratory capacity of BMCs was not intensively affected by proliferation we compared early time points of migrating and proliferating BMCs. We observed that cells featured distinct migratory properties at early time points but showed comparable proliferation (Reviewer only figure 1c) which was more evident after normalization to same cell density at time point zero (Reviewer only figure 2). Our previous results that are included in the manuscript together with data showing in Figures 1 and 2 suggest that the migration assay was not considerably biased by the cell's proliferative capacity.

To clarify the terms “relative wound density” and confluence: we are using the term “relative wound density” (RWD) to present data from scratch assays as the confluence/density of cells within the wound comparing to the overall confluence/density cells. It is determined by confluence masks. Same masks are applied by the software to growing cells that are tracked for a certain time in standard proliferation assays. Therefore we are using the term “Confluence” when talking about cell proliferation.

We added the following text to the Materials&Methods section (page 13, lines 360-365:

Proliferation assay

Briefly, 2.5×10^3 A375 cells or 5.0×10^3 BMC cells per well were seeded on 96-well plates and treated with dabrafenib or treatment control (DMSO) or left untreated, 24 h after plating. Following, proliferation was monitored for 72h to 96h and was determined by change in the cellular density as determined by images captured every 3 hours and software based calculation of a confluence mask that was defined by cellular morphology. The counting of masked cells defined the “Phase Object Confluence (%)” per time.

We hope that the additional data we provide are convincing and clarifying. Additionally, we would like to mention that the migration assay only enables a categorization of BMCs into more proliferative or migratory. Likely, both properties rather reflect interconnected than stable cell states that are probably not strictly separable.

We leave it up to the reviewer to decide whether this additional data should be included in the manuscript.

REVIEWER COMMENTS

Reviewer #3 (Remarks to the Author):

I would like to thank the authors for thoroughly addressing this concern and highlighting the limitations. I do not have any additional suggestions.